FERMILAB-CONF-19-299-T

# Neutrino Non-Standard Interactions: A Status Report

**P. S. Bhupal Dev,**[1,2a,b] **K. S. Babu,**[1c,b] **Peter B. Denton,**[1d] **Pedro A. N. Machado,**[1b] **Carlos A. Argüelles,**[e] **Joshua L. Barrow,**[f,g] **Sabya Sachi Chatterjee,**[h] **Mu-Chun Chen,**[i] **André de Gouvêa,**[j] **Bhaskar Dutta,**[k] **Dorival Gonçalves,**[l] **Tao Han,**[l] **Matheus Hostert,**[h] **Sudip Jana,**[a,b] **Kevin J. Kelly,**[b] **Shirley Weishi Li,**[m] **Ivan Martinez-Soler,**[b,j,n] **Poonam Mehta,**[o] **Irina Mocioiu,**[p] **Yuber F. Perez-Gonzalez,**[b,j,n] **Jordi Salvado,**[q] **Ian M. Shoemaker,**[r] **Michele Tammaro,**[s] **Anil Thapa,**[a,b] **Jessica Turner,**[b] **Xun-Jie Xu**[t]

[a] *Department of Physics and McDonnell Center for the Space Sciences, Washington University, St. Louis, MO 63130, USA*

[b] *Theoretical Physics Department, Fermi National Accelerator Laboratory, P.O. Box 500, Batavia, IL 60510, USA*

[c] *Department of Physics, Oklahoma State University, Stillwater, OK, 74078, USA*

[d] *Department of Physics, Brookhaven National Laboratory, Upton, NY 11973, USA*

[e] *Massachusetts Institute of Technology, Cambridge, MA 02139, USA*

[f] *Fermi National Accelerator Laboratory, MS220, PO Box 500, Batavia, IL 60510, USA*

[g] *Department of Physics & Astronomy, The University of Tennessee, Knoxville, TN 37996, USA*

[h] *Institute for Particle Physics Phenomenology, Department of Physics, Durham University, South Road, Durham DH1 3LE, United Kingdom*

[i] *Department of Physics and Astronomy, University of California, Irvine, CA 92697, USA*

[j] *Department of Physics & Astronomy, Northwestern University, 2145 Sheridan Road, Evanston, IL 60208, USA*

[k] *Mitchell Institute for Fundamental Physics and Astronomy, Department of Physics and Astronomy, Texas A&M University, College Station, TX 77843, USA*

[l] *Pittsburgh Particle Physics Astrophysics and Cosmology Center (PITT PACC), Department of Physics and Astronomy, University of Pittsburgh, Pittsburgh, PA 15260, USA*

[m] *SLAC National Accelerator Laboratory, 2575 Sand Hill Road, Menlo Park, CA 94025, USA*

[n] *Colegio de Física Fundamental e Interdisciplinaria de las Américas (COFI), 254 Norzagaray street, San Juan, Puerto Rico 00901*

[o] *School of Physical Sciences, Jawaharlal Nehru University, New Delhi 110067, India*

[p] *Department of Physics, The Pennsylvania State University, University Park, PA 16802*

[q] *Departament de Física Quántica i Astrofísica and Institut de Ciències del Cosmos, Universitat de Barcelona, Diagonal 647, E-08028 Barcelona, Spain*

[r] *Center for Neutrino Physics, Department of Physics, Virginia Tech, Blacksburg, VA 24061, USA*

[s] *Department of Physics, University of Cincinnati, Cincinnati, OH 45221, USA*

[t] *Max-Planck-Institut für Kernphysik, Saupfercheckweg 1, D-69117 Heidelberg, Germany*

---

[1]Editors

[2]Corresponding Author (Email: bdev@wustl.edu)

ABSTRACT: This report summarizes the present status of neutrino non-standard interactions (NSI). After a brief overview, several aspects of NSIs are discussed, including connection to neutrino mass models, model-building and phenomenology of large NSI with both light and heavy mediators, NSI phenomenology in both short- and long-baseline neutrino oscillation experiments, neutrino cross-sections, complementarity of NSI with other low- and high-energy experiments, fits with neutrino oscillation and scattering data, DUNE sensitivity to NSI, effective field theory of NSI, as well as the relevance of NSI to dark matter and cosmology. We also discuss the open questions and interesting future directions that can be pursued by the community at large. This report is based on talks and discussions during the Neutrino Theory Network NSI workshop held at Washington University in St. Louis from May 29-31, 2019 (https://indico.cern.ch/event/812851/).

# Contents

# 1 Overview

Given the wide interest in the worldwide neutrino program, it is timely to reassess the state of the art topics related to non-standard neutrino interactions (NSIs). This document presents an overview of NSIs and a number of in depth modern analyses spanning numerous related topics presented at a recent workshop.

## 1.1 Introduction to Non-Standard Neutrino Interactions (Denton)

NSIs provide a general effective field theory (EFT) style framework to quantify new physics in the neutrino sector[1]. While the details of a specific model may vary, they typically all have the following forms for NC and CC NSI,

$$\mathcal{L}_{\mathrm{NC}} = -2\sqrt{2}G_F \sum_{f,P,\alpha,\beta} \varepsilon_{\alpha\beta}^{f,P} (\bar{\nu}_\alpha \gamma^\mu P_L \nu_\beta)(\bar{f}\gamma_\mu P f) \,, \tag{1.1}$$

$$\mathcal{L}_{\mathrm{CC}} = -2\sqrt{2}G_F \sum_{f,P,\alpha,\beta} \varepsilon_{\alpha\beta}^{f,P} (\bar{\nu}_\alpha \gamma^\mu P_L \ell_\beta)(\bar{f}\gamma_\mu P f') \,, \tag{1.2}$$

where $G_F$ is Fermi's constant and the $\varepsilon$ terms quantify the size of the new interaction relative to the weak scale. The sum is over matter fermions, typically $f, f' \in \{e, u, d\}$ and $P \in \{P_L, P_R\}$ are the chirality projection operators. These projection operators can also be reparameterized into vector and axial components of the interaction. NSIs were first introduced by Wolfenstein in 1978 in his landmark paper that also identified the conventional matter effect [1].

Such a new interaction leads to a rich phenomenology in both scattering experiments and neutrino oscillation experiments [2–4]. Since oscillation phenomenology is generally quite distinct from scattering phenomenology, the NSI framework provides a convenient way to relate new physics models to both cases. The $\varepsilon$ terms can be thought of in a simplified model framework as $\varepsilon \propto g_X^2/M_X^2$. In the case of scattering the denominator becomes $q^2 + M_X^2$ indicating that a scattering experiment is only sensitive to mediators heavier than the typical energy scale of the experiment. NuTeV and COHERENT have particularly strong NSI scattering constraints [5–7].

The vector component of NSIs affect oscillations by providing a new flavor dependent matter effect. The Hamiltonian for this is

$$H = \frac{1}{2E} \left[ U_{\mathrm{PMNS}} \begin{pmatrix} 0 & & \\ & \Delta m_{21}^2 & \\ & & \Delta m_{31}^2 \end{pmatrix} U_{\mathrm{PMNS}}^\dagger + a \begin{pmatrix} 1+\varepsilon_{ee} & \varepsilon_{e\mu} & \varepsilon_{e\tau} \\ \varepsilon_{e\mu}^* & \varepsilon_{\mu\mu} & \varepsilon_{\mu\tau} \\ \varepsilon_{e\tau}^* & \varepsilon_{\mu\tau}^* & \varepsilon_{\tau\tau} \end{pmatrix} \right] \,, \tag{1.3}$$

where $U_{\mathrm{PMNS}}$ is the standard lepton mixing matrix [8, 9], $a \equiv 2\sqrt{2}G_F N_e E$ is the Wolfenstein matter potential, $N_e$ is the electron number density, $E$ is the neutrino energy, and the 1 in the $1 + \varepsilon_{ee}$ term is due to the standard charged current matter potential. For useful reviews see e.g. refs. [10–13]. The diagonal NSI terms are known as non-universal since

---

[1]The fact that neutrinos have mass already guarantees new physics beyond the standard model. NSIs represent new physics beyond mass generation.

they provide a mechanism for breaking lepton flavor universality while the off-diagonal terms are known as flavor-changing. While the non-universal (diagonal) terms are real, the flavor-changing terms (off-diagonal) are, in general complex and can interfere with the standard CP-violating phase [14], and can be parameterized,

$$\varepsilon_{\alpha\beta} = |\varepsilon_{\alpha\beta}|e^{i\phi_{\alpha\beta}} . \tag{1.4}$$

Since the flavor-changing NSI terms can be complex, at the Hamiltonian level there are 9 new parameters, of which 8 are testable by oscillations. Including scattering or oscillations in different materials the number of parameters increases sharply. If we consider NSI with $f \in \{e, u, d\}$ and both left and right handed NSI, this leads to 54 parameters, which can be even larger if different Lorentz structures are taken to mediate the interaction, or if the EFT is elevated to a simplified model with a mass. In light of this, most fits to data make several assumptions about the nature of NSI to reduce the number of parameters to a more tractable number. For this reason, it is important to be very careful about directly comparing the results of separate analyses and global fits (Secs. 4, 23).

There is one free parameter along the diagonal that oscillation experiments are never sensitive to, in the same way that oscillation experiments cannot measure the absolute neutrino mass scale. Without loss of generality the $\varepsilon_{\mu\mu}$ can be subtracted out and the diagonal part of the matter potential component of the Hamiltonian can be written as $a \, \text{diag}(1 + \varepsilon_{ee} - \varepsilon_{\mu\mu}, 0, \varepsilon_{\tau\tau} - \varepsilon_{\mu\mu})$. This degeneracy can only be probed by scattering experiments which are directly sensitive to the individual diagonal terms. There is another interesting exact degeneracy wherein the standard picture with no NSIs is exactly equivalent to a picture with large NSIs ($\varepsilon_{ee} = -2$) and the opposite mass ordering [15, 16]. That is, switching the normal ordering and the inverted ordering. This is essentially only probable via scattering experiments [17–20]. Many other approximate degeneracies involving NSIs and standard parameters have been identified in the literature (Sec. 19).

It is also important to note that the $\varepsilon$ terms in Eq. (1.1) differ from those in Eq. (1.3) where the former are referred to as Lagrangian level NSI and the latter as Hamiltonian level NSI. The distinction is that at the Hamiltonian level the strength of the NSI is given relative to the electron number density which controls the standard matter effect. These two are related via,

$$\varepsilon_{\alpha\beta} = \sum_{f\in\{e,u,d\}} \left\langle \frac{N_f(x)}{N_e(x)} \right\rangle \varepsilon_{\alpha\beta}^{f,V} , \tag{1.5}$$

where $N_f(x)$ is the number density of fermion $f$ at position $x$, and the Lorentz structure is typically taken to mean vector in the context of oscillation experiments unless otherwise specified. For example, in the Earth, we typically have the quantity in brackets as roughly three for each up quarks and down quarks, with slight corrections since the neutron number density in the Earth is slightly larger than the electron number density.

In the context of notation, the two levels can be distinguished by the presence of the fermion superscript for the Lagrangian level, or the lack thereof for the Hamiltonian level where a matter density is usually clear by context. In contexts when both scattering and oscillation are considered together, such as in global fits, it is our recommendation to

either use the explicit Lagrangian level notation, or the Hamiltonian level notation under a specific model assumption, such as $\varepsilon^e = 0$, $\varepsilon^u = \varepsilon^d$. This leads to another potential issue in comparing results: results at the Lagrangian level and results at the Hamiltonian level often differ by a factor of $\sim 3$ assuming that NSI couples to quarks only since there are about three of each light quark per electron in the Earth.

Numerous other examples of NSI style phenomenology exist in the literature. One is Non-Standard neutrino Self Interactions (NSSIs) (Sec. 24) where the interaction is with other neutrinos such as in the early universe or in core-collapse supernovae. Another is charged-current NSI wherein the oscillation picture is also modified at production and detection, in addition to scattering experiments. Charged current NSI with electrons can be Fierzed to neutral current NSI. NSIs can also be probed in collider experiments. Finally, beyond just left or right handed interactions (or vector or axial), interactions can, in general, be any of $S$, $P$, $V$, $A$, or $T$ (Secs. 16 and 17).

The above overview of NSIs represents the conventional picture of NSIs. Throughout this document we show that not only is this picture still very vital with many interesting constraints coming from new experiments, but we also show that the picture has evolved beyond this considerably with new models, new phenomenology, and new tests. For other unconventional NSIs not covered here, see e.g. Refs. [21] (scalar NSI) and [22] (dark NSI).

## 1.2 Motivation for NSIs (Dev)

The neutrino oscillation program is entering a new era, where the known parameters are being measured with an ever-increasing accuracy. Next generation of long–baseline neutrino experiments like DUNE [23] are poised to resolve the sub-dominant effects in oscillation data sensitive to the currently unknown oscillation parameters, namely $\delta_{\rm CP}$ and sign of $\Delta m_{32}^2$. Of course, all these derivations are within the $3 \times 3$ neutrino mass and mixing scheme under the assumption that neutrinos interact with matter only through the Standard Model (SM) weak interactions. Allowing for NSI can in principle change the whole picture. NSIs are of two types: Neutral Current (NC) NSI [cf. Eq. (1.1)] and Charged Current (CC) NSI [cf. Eq. (1.2)]. While the CC NSI of neutrinos with the matter fields $(e, u, d)$ affects in general the production and detection of neutrinos, the NC NSI affects the neutrino propagation in matter. The effects of both types of NSI on neutrino experiments have been extensively studied in the literature; see e.g., Refs. [10–13] for reviews. The general, model-independent bounds from the combination of neutrino oscillation and detection or production experimental results have been summarized in Refs. [11, 13] (see also Sec. 4 and Refs. [24, 25] for recent global-fit constraints on NSI parameters).

On the other hand, NSI of neutrinos can crucially affect the interpretation of the experimental data in terms of the relevant $3 \times 3$ neutrino oscillation parameters. For instance, the presence of NSI in the neutrino propagation may give rise to a degeneracy in the measurement of the solar mixing angle [26]. Likewise, CC NSI at the production and detection of reactor antineutrinos can affect the very precise measurement of the reactor mixing angle $\theta_{13}$ [14, 27]. Moreover, NSI can cause degeneracies in deriving the CP–violating phase $\delta_{\rm CP}$ [28, 29], mass hierarchy [30], as well as the correct octant of the atmospheric mixing angle $\theta_{23}$ [31] at current and future long–baseline neutrino experiments (see Sec. 19).

There are possible ways to resolve the parameter degeneracies due to NSI, by exploiting the capabilities of some of the planned experiments such as the intermediate baseline reactor neutrino experiments JUNO [32] and RENO50 [33].

Most of the NSI analyses parameterize the new interactions in terms of the effective dimension-6 operator given in Eq. (1.1). If the effective coupling comes from integrating out a new state $(X)$ of mass $m_X$ and coupling $g_X$, we expect the strength of the NSI parameters to be given by $\varepsilon \sim g_X^2 m_W^2/m_X^2$. Thus, for the NSI to be experimentally observable $(\gtrsim 10^{-2})$, the new particle $X$ cannot be much heavier than the electroweak scale (see Sec. 7 for a concrete model realization of large NSI with $m_X \sim \mathcal{O}(100)$ GeV). The alternative approach is to take $m_X \ll m_W$ and $g_X \ll 1$ such that $g_X^2/m_X^2 \sim G_F$, while evading the low-energy experimental constraints (see Secs. 13 and 14 for examples of this scenario).

One may wonder whether it is possible to build viable renormalizable, UV-complete models of neutrino mass with NSIs large enough to be discernible at neutrino oscillation experiments. In general, since neutrinos are part of the $SU(2)_L$ doublet in the SM, respecting the SM gauge symmetry imposes stringent constraints on possible models of large NSI [34]. For instance, allowing the dimension-6 operator of the form given in Eq. (1.1) with $f = e$ typically leads to another dimension-6 operator involving four charged leptons: $(\bar{\ell}_\alpha \gamma^\mu P_L \ell_\beta)(\bar{e}\gamma_\mu Pe)/\Lambda^2$, which is severely constrained by the charged-lepton flavor violation (LFV) searches, such as $\mu \to eee$. In this case, the current limit on $\mathrm{BR}(\mu \to 3e) < 10^{-12}$ [35] translates into an upper limit on $\varepsilon_{e\mu} \lesssim 10^{-6}$. Similar stringent constraints can be derived from other rare LFV processes, such as $\ell \to \ell'\gamma$, as well as from universality in muon and tau decays, assuming CKM unitarity [10]. However, there exist a few explicit UV-complete models with observable NSI (mostly involving $\nu_\tau$), where all experimental constraints can be avoided by a clever choice of the flavor couplings (see Secs. 1.3 and 7 for more details.) In summary, NSI provides yet another way to probe new physics at scales below or close to the electroweak scale, which is complementary to various other low and high-energy probes of neutrino mass models (see Table 31).

## 1.3   UV Complete Models of NSI (Babu)

The effective dimension-6 operators of Eq. (1.1) generating nonstandard neutrino interactions should arise from some fundamental renormalizable theory for such NSI to be reliable. This is an important theoretical challenge, which is a topic of ongoing research. A major hurdle is that by making the operators of Eq. (1.1) $SU(2)_L$ gauge invariant, new interactions among charged leptons will be induced, which are highly constrained by LFV and universality violation. For example, the strong constraint on the radiative decay of the muon, $\mathrm{Br}(\mu \to e\gamma) \leq 4.2 \times 10^{-13}$ [36], would naively set a limit of order $10^{-6}$ on the flavor changing NSI $\varepsilon_{e\mu}$, which would be too small to be probed in future neutrino oscillation experiments. Such a strong correlation between neutrino NSI and LFV, however, can be evaded in a number of ways, as discussed below.

If the $d = 6$ operators of Eq. (1.1) arise from higher dimensional $d = 8$ operators with two Higgs fields, once the vacuum expectation value of the Higgs field is inserted, then it is possible that nonstandard interactions arise only in the neutrino sector, and not in the

charged lepton sector [37, 38]. An example is given by the operator

$$(\overline{L^i}H_i)\gamma_\mu(H^{i\dagger}L_i)(\overline{e}_R\gamma^\mu e_R) \tag{1.6}$$

where $L$ and $e$ stand for the lepton doublet and singlet, and $H$ is the Higgs doublet with $Y = +1/2$. However, it has been noted [34] that models of this type are highly constrained from other low energy processes – such as non-unitarity of the PMNS matrix, electroweak oblique corrections, etc. – and would lead to very small neutrino NSI. One possibiliy is to generate the $d = 6$ operator of Eq. (1.1) as well as the $d = 8$ operators of Eq. (1.6) with some cancellation between the two in the charged lepton sector. Such cancellations are not known to be protected by symmetry.

Even without such cancellation, reasonably large neutrino NSI has been shown to be present in standard model extensions with additional scalars. For example, in Ref. [39] (see also Ref. [40]), it has been shown that the exchange of a charged $SU(2)_L$ scalar which mixes with an $SU(2)_L$ doublet field can generate $\varepsilon_{e\tau} \sim 0.3$, without violating collider and low energy constraints. A systematic analysis of radiative models of neutrino masses, as discussed in Section 7, has shown that large NSI mediated by charged scalars or leptoquarks can be realized, consistent with LEP, LHC and low energy data.

An interesting possibility that has received much attention lately is to use a light mediator to generate the neutrino NSI. Effective description in terms of $d = 6$ operator of Eq. (1.1) then would become invalid. However, for neutrino propagation in matter, it is the $q^2 = 0$ component that contributes to coherent forward scattering, which will still be described by the operators of Eq. (1.1). With light mediators, typically a new $U(1)'$ gauge boson, $SU(2)_L$ invariance does not require LFV. For example, in Ref. [41], a $U(1)$ extension of the standard model is constructed, corresponding to $B-L$ for the third family. If the new gauge boson has a mass of order 100 MeV, and a gauge coupling of order $10^{-3}$, large and observable NSI of the type $\varepsilon_{\alpha\alpha}$ are possible. A flavor-dependent $B - L$ model with a light $Z'$ has been constructed in Ref. [42], which also yields large NSI without excessive LFV. Baryonic $Z'$ models with small and flavor-dependent leptonic couplings have been shown to generate consistently large NSI in Refs. [20, 43]. The phenomenology of light mediators for NSI is discussed in Section 13 and explicit models are discussed in Sections 14 and 15.

All in all, there exist interesting UV-complete models that generate neutrino NSI of interest to oscillation experiments without conflicts with LFV and universality. Further research in this area is desirable, which could lead to other interesting models of large neutrino NSI.

## 1.4 Outlook and Discussion (Machado)

Here we present a summary of the topics discussed and we pose certain questions or thoughts that emerged during the workshop. The global status of effective NSI operators in the context of neutrino oscillation was presented (Sec. 4), as well as an in-depth exploration of the impact of NSIs IceCube (Sec. 23), and future DUNE constraints on NSIs taking into account the possibility of having a high energy beam configuration (Sec. 22). Constraints on more general NSI operators from neutrino scattering processes were also

discussed (Secs. 16, 17). It would be beneficial to characterize which models can give rise to general NSI operators. Even accounting for the present constraints, NSIs can lead to considerable effects on the measurement of neutrino properties (Secs. 11, 19) and on dark matter experiments through the neutrino floor (Sec. 5). The effective low energy NSI framework, Eq. (1.1) is particularly useful, as it provides a model independent way of looking for a broad class of new physics scenarios in neutrino oscillation experiments. Nevertheless, it is only valid at scales well below electroweak symmetry breaking, and thus it is important to consider several aspects of the NSI framework, such as theoretical consistency, experimental synergies, and so on.

Recent studies have shown that it is possible to build ultraviolet complete models that can give rise to substantial effects in neutrino oscillation measurements (Secs. 7, 13). In such UV constructions, additional field content is called upon, possibly leading to appreciable phenomenology in neutrinoless double beta decay experiments, flavor observables, rare meson decays, and high energy colliders (Sec. 3). For instance, further studies on the complementarity and interplay between neutrino oscillation measurements and LHC observables in the context of effective NSIs, simplified models, and full models are needed (Sec. 6). Besides, additional work on UV complete scenarios in order to identify realistic models of NSIs and their corresponding phenomenology is still desired. Theories of flavor were also discussed with emphasis on the dynamical generation of the Dirac $CP$ phase (Sec. 2), as well as leptogenesis scenarios in which the matter-antimatter asymmetry comes entirely from $\delta_{CP}$ (Sec. 10). Among the questions raised in the workshop, we highlight particularly challenging ones: *How low can the scale of convincing flavor models and neutrino mass models be? How can discrete flavor models be tested? How can leptogenesis be disproved?*

Finally, some NSI models may address current experimental anomalies, and/or predict non-trivial experimental signatures like neutrino tridents (Secs. 8, 18, 24). It was shown that dedicated searches for certain scenarios involving BSM in the neutrino sector or light dark matter in neutrino experiments can be quite powerful (Secs. 9, 12). To take full advantage of these, a better understanding of neutrino-nucleus interactions is required (Sec. 20). The capabilities of liquid argon detectors in probing non-trivial experimental signatures is still underappreciated and additional collaboration between theorists, experimentalists and event generator developers is necessary (Sec. 21).

The slides of the talks can be found in https://indico.cern.ch/event/812851/.

## 2 Neutrino Mass Generation and Flavor Mixing (Chen)

The measurements of various neutrino parameters have entered from the discovery phase into precision phase. Current data posts two theoretical challenges: (i) Why neutrino masses are so much smaller compared to the charged fermion masses? (ii) Why neutrino mixing angles are large while quark mixing are small? A variety of approaches based on different new physics frameworks have been proposed to address these challenges. In addition to addressing the neutrino mass generation and flavor problem, these models can also afford solutions to other issues in particle physics as well as predictions that can be tested experimentally. Even though the models described in this section assumes three neutrinos, they may be generalized to incorporate sterile neutrinos as well as NSI.

### 2.1 Smallness of the Neutrino Masses

The scale of new physics at which neutrino mass generation occurs is still unknown. This scale can range from the electroweak scale all the way to the GUT scale. And depending on the new physics, it is possible to obtain naturally small neutrino masses both of the Majorana type and of the Dirac type.

The Weinberg operator is the lowest higher dimensional operator if one assumes that the SM is a low energy effective theory. Given that the Weinberg operator breaks the lepton number by two units, neutrinos are Majorana fermions. There are three possible ways to UV-complete the Weinberg operator depending on whether the portal particle is a SM gauge singlet fermion, a complex weak triplet scalar, or a weak triplet fermion. These are dubbed the Type-I [44], II [45], and III [46] seesaw mechanism, respectively. The Type-I seesaw mechanism can be naturally incorporated in a Grand Unified Theory, such as one based on SO(10) symmetry group, while keeping the Yukawa coupling constants of order unity. On the other hand, it has been shown that the portal particles in the Type-II and Type-II seesaw mechanisms are not easily obtainable in string-inspired models.

If one allows for neutrino Yukawa coupling constants to be as small as the electron Yukawa coupling, it is possible to lower the seesaw scale down to the TeV range, thus allowing for the testability of the seesaw mechanisms at the Collider experiments. Beyond the three types of seesaw mechanisms, small neutrino masses can also be generated radiatively [47, 48], or through the $R$-parity breaking $B$-term in MSSM [49, 50], in addition to the so-called inverse-seesaw mechanism [51]. Depending on the new particles and new interactions possessed by the new physics models, there are signatures through which the models can be tested at the collider and low energy precision experiments, as discussed in Sec. 3. For a review and references, see for example, [52].

For Dirac neutrinos, it is also possible to generate their small masses naturally. In particular, in many new physics models beyond the SM aiming to address the gauge hierarchy problem, suppression mechanisms for neutrino masses are naturally incorporated. These include warped extra dimension models [53], supersymmetric models [54], and more recently the clockwork models [55]. Even though in some of these models [56], neutrinos are Dirac fermions and all lepton number violating operators with $\Delta L = 2$ are absent to

all orders, having been protected by symmetries, there can exist lepton number violation by higher units, leading to new experimental signatures [57].

## 2.2 Flavor Mixing and CP Violation

Generally there are two approaches to address the flavor puzzle. One is the so-called "Anarchy" scenario [58, 59] which assumes that there is no parametrically small parameter, and the observed large mixing angles and mild hierarchy among the masses are consequences of statistics. Even though at low energy the anarchy scenario appears to be rather random, predictions from UV physics, such as warped extra dimension [53, 60] as well as heterotic string models where the existence of some $\mathcal{O}(100)$ right-handed neutrinos are predicted [61], very often can mimic the results of anarchy scenario [62].

An alternate approach is to assume that there is an underlying symmetry, whose dynamics governs the observed mixing pattern and mass hierarchy. The observed large values for the mixing angles have motivated models based on discrete non-Abelian flavor symmetries. Symmetries that have been utilized include $A_4$ [63], $A_5$ [64], $T'$ [65], $S_3$ [66], $S_4$ [67], $\Delta(27)$ [68], $Z_7 \ltimes Z_3$ [69] and $Q(6)$ [70]. In addition, these discrete (flavor) symmetries may originate from extra dimension compactification [71, 72].

Generically, the prediction for the PMNS mixing matrix arises due to the mismatch between the symmetry breaking pattern in the neutrino sector and that in the charged lepton sector. Due to the symmetries of the models, different relations among the physical parameters are predicted. Such correlations can be a robust way for distinguishing different classes of models. To test some of these correlations requires measurements of mixing parameters at a precision that is compatible to the precision in the measurements for quark mixing parameters.

In addition to provide an elegant understanding of the observed mixing patterns, non-Abelian discrete symmetries also affords a novel origin for CP violation. Specifically, CP violation can be entirely group theoretical in origin [73], due to the existence of complex Clebsch-Gordan coefficients in certain non-Abelian discrete symmetries [74], leading to a rather predictive framework, in addition to providing a deep possible connection between residual spacetime symmetry after the compactification and the flavor symmetries at low energy [75].

## 3 Testing Seesaw: From $0\nu\beta\beta$ to Colliders (Han)

Observing lepton number violation would imply the existence of Majorana mass for neutrinos [76–78], confirming the existence of a new mass scale associated with the neutrino mass generation would, in addition, verify the general concept of a seesaw mechanism. There have been on-going experimental efforts in several directions, most notably the neutrinoless double beta decay experiments ($0\nu\beta\beta$), both current [79–82] and upcoming [83–85], as well as proposed general purpose fixed-target facilities [86, 87]. Complementary to these are on-going searches for lepton number violating processes at collider experiments, which focus broadly on rare meson decays [88–90], heavy neutral fermions in Type I-like models [91–95], heavy bosons in Type II-like models [96–98], heavy charged leptons in Type III-like

| Models | $0\nu2\beta$ | $\mu$-$e$ conv. $\mu \to e\gamma$ etc. | rare decays $\tau, K, D, B$ | LLP | $e^+e^-$ | $pp$ | features | CPv/cosmo |
|---|---|---|---|---|---|---|---|---|
| Type-I | $\checkmark$ | $\checkmark$ | $\checkmark$ | $\checkmark$ | $\checkmark$ | $\checkmark$ | $N$ | $\checkmark$ |
| Type-II | $\checkmark$ | $\checkmark$ | $\checkmark$ | $\checkmark$ | $\checkmark$ | $\checkmark$ | $H^{\pm\pm}$, $W_R^{\pm}$ | $\checkmark$ |
| Type-III | $\checkmark$ | $\checkmark$ | ? | $\checkmark$ | $\checkmark$ | $\checkmark$ | $T^{\pm}, T^0$ | $\checkmark$ |
| Zee (1-loop) | ? | ? | ? | ? | ? | ? | $h^{\pm}$ | ? |
| Ma (1-loop) | ? | ? | ? | ? | ? | ? | scalars, DM | ? |
| Zee-Babu (2-loop) | ? | ? | ? | ? | ? | ? | $k^{\pm\pm}$, $h^{\pm}$ | ? |
| 3-loops | ? | $\checkmark$ | ? | ? | $\checkmark$ | $\checkmark$ | scalars, DM | ? |
| extra-dim | ? | ? | ? | ? | $\checkmark$ | $\checkmark$ | KK states | ? |
| RPV/Leptoquark | $\checkmark$ | $\checkmark$ | $\checkmark$ | ? | $\checkmark$ | $\checkmark$ | Leptoquark | $\checkmark$ |
| $\nu\nu\phi$ | $\checkmark$ | ? | $\checkmark$ | ? | ? | ? | $E_{\text{miss}}$ | ? |
| Inverse/linear | $\checkmark$ | $\checkmark$ | $\checkmark$ | $\checkmark$ | $\checkmark$ | $\checkmark$ | $N$ | $\checkmark$ |
| Pseudo Dirac | ? | ? | $\checkmark$ | ? | $\checkmark$ | $\checkmark$ | ? | ? |
| NSI (dim 6) | ? | ? | ? | ? | $\checkmark$ | $\checkmark$ | mediators | ? |
| higher-dim ops | ? | $\checkmark$ | ? | ? | $\checkmark$ | $\checkmark$ | $T^{\pm}$ | ? |

**Table 31:** Summary of neutrino mass models and testable features. $\checkmark$ means already studied and ? means more work needed.

models [99–101], and lepton number violating contact interactions [102, 103]. Furthermore, accurate measurements of the PMNS matrix elements and stringent limits on the neutrino masses themselves provide crucial information and knowledge of lepton flavor mixing that could shed light on the construction of the seesaw models. For complementary reviews on a variety of models, we refer readers to Refs. [52, 104–108] and references therein.

Along with the current bounds from the experiments at LEP, Belle, LHCb and AT-LAS/CMS at 8 and 13 TeV, some recent studies for the 13/14 TeV LHC, a future 100 TeV hadron collider, an $ep$ collider (LHeC), and a future high-energy $e^+e^-$ collider were summarized in a review [109]. There, a number of tree- and loop-level seesaw models were considered, including, as phenomenological benchmarks, the canonical Type I, II, and III seesaw mechanisms, their extensions and hybridizations, and radiative seesaw formulations in $pp$, $ep$, and $ee$ collisions. We note that the classification of collider signatures based on the canonical seesaws is actually highly suitable, as the same underlying extended and hybrid seesaw mechanism can be molded to produce wildly varying collider predictions. Searching for new signatures such as long-lived particles (LLP) with displaced tracks and disappearing tracks are being proposed [110–114]. It is noted that state-of-the-art computations, newly available Monte Carlo tools are being developed for further studies to expand the coverage of seesaw parameter spaces at current and future colliders.

While searches for seesaw model signatures are on-going, it is highly desirable to develop a systematic program for testing and probing all classes of well motivated models, including the tree-level seesaw and radiatively generated Majorana masses, over broad scope of experiments, from $0\nu\beta\beta$, rare meson decays to high energy colliders, with both lepton-number violating processes as well as the related lepton-flavor violating processes in the charged lepton sector. Table 31 summarizes many of the neutrino mass models, along with their main teastable features and signatures both low and high-energy experiments.

# 4 Present Bounds on Non-Standard Interaction from a Global Oscillation Analysis (Martinez-Soler)

In the 3 neutrino oscillation scenario, the neutrino evolution in vacuum is described by two mass parameters ($\Delta m_{21}^2$, $\Delta m_{31}^2$), three mixing angles ($\theta_{12}$, $\theta_{13}$, $\theta_{23}$) and a complex phase ($\delta_{CP}$). In matter, we also need to consider the coherent forward elastic scattering of the neutrinos with the medium described by the matter potential $V_{mat} = \sqrt{2}G_F N_e(x)\text{diag}(1,0,0)$, where $N_e$ is the electron density through the neutrino trajectory.

In the presence of Non-Standard Interactions, the production/detection, as well as the neutrino propagation, can be altered depending on whether NSI modify the CC (Eq. 1.1) or NC (Eq. 1.2), respectively. We have focused on the case where only the neutrino propagation is altered, NSI-NC. Those new interactions generalize the matter potential into a non-diagonal matrix as in Eq. (1.3), where $\varepsilon_{\alpha\beta}$ correspond to the strength of NSI, and includes all the possible new interaction with the different fermions present in the medium. Such matter potential introduces 9 additional parameters, but since the flavor oscillations are invariant under a global phase transformation, just 8 of them can be determined by oscillation experiments. By convention, we remove $\varepsilon_{\mu\mu}$ from the diagonal elements.

Ordinary matter is composed by protons, neutrons and electrons. In this work [24], we have considered only NSI with quarks. In this way, $\varepsilon_{\alpha\beta}$ can be written as

$$\varepsilon_{\alpha\beta}(x) = \sum_{f=p,n} \left\langle \frac{N_f(x)}{N_e(x)} \right\rangle \varepsilon_{\alpha\beta}^f \tag{4.1}$$

where $N_f(x)$ is the fermion density along the neutrino trajectory. Assuming that the flavor structure of the NSI is independent of the fermion which carries the new interaction we can write the coupling with each fermion as a product of two terms, $\varepsilon_{\alpha\beta}^p = \varepsilon_{\alpha\beta}^\eta \xi^p$ and $\varepsilon_{\alpha\beta}^n = \varepsilon_{\alpha\beta}^\eta \xi^n$, allowing to write

$$\varepsilon_{\alpha\beta}(x) = \varepsilon_{\alpha\beta}^\eta \left[\xi^p + Y_n(x)\xi^n\right] \tag{4.2}$$

where $Y_n(x)$ is the neutron fraction. In the above equation, we have assumed the matter electrically neutral. The parameters $(\xi^p, \xi^n)$ describe the relative coupling of NSI to protons and neutrons, and can be parameterized in terms of an angle $\eta$, $\xi^p = \sqrt{5}\cos\eta$ and $\xi^n = \sqrt{5}\sin\eta$. Particularly interesting are the cases when $\eta = 0$, the new interaction is proportional to the electric charge or $\eta = 90°$, NSI only couple to neutrons. In terms of the quark coupling, for $\eta = \arctan(1/2)$ ($\eta = \arctan(2)$) correspond to NSI with up (down) quarks.

In vacuum, the Hamiltonian is invariant under a CPT transformation ($H_{vac} \to -H_{vac}^*$), which can be translated into a symmetry over the mass hierarchy and the octant of $\theta_{12}$

$$\Delta m_{31}^2 \to -\Delta m_{32}^2$$
$$\theta_{12} \to \pi/2 - \theta_{12}$$
$$\delta_{CP} \to \pi - \delta_{CP} \tag{4.3}$$

This degeneracy is broken when the neutrinos propagate through matter. The symmetry can be recovered by transforming $\varepsilon_{ee}(x) - \varepsilon_{\mu\mu}(x) \to -(\varepsilon_{ee}(x) - \varepsilon_{\mu\mu}(x)) - 2$, $\varepsilon_{ee}(x) - \varepsilon_{\mu\mu}(x) \to$

$-\varepsilon_{\tau\tau}(x) - \varepsilon_{\mu\mu}(x)$ and $\varepsilon_{\alpha\beta}(x) \rightarrow -\varepsilon_{\alpha\beta}^*$. Since $\varepsilon_{\alpha\beta}(x)$ depends on the matter potential, the degeneracy becomes exact only when the matter dependence vanishes ($\varepsilon_{\alpha\beta}^n = 0$) or for $Y_n(x)$ constant. The solution where $\theta_{12}$ lives in the second octant is called LMA-Dark (LMA-D) solution.

The strongest constraints over the NSI-NC parameters come from oscillation experiments. The $\varepsilon_{\alpha\beta}$ introduced in the matter potential will modify the neutrino evolution via the MSW effect. In this work, we have studied the present sensitivity to NSI by a CP-conserving global fit of oscillation data, available until January 2018. In addition, we have studied the compatibility of the $3\nu$ scenario in the presence of NSI.

The effect introduced by $\varepsilon_{\alpha\beta}$ will be relevant for the oscillation in the regions where the matter effects dominate. This is the case for solar neutrinos, where a big fraction of the electron neutrinos are created via pp-chains and CNO-cycles in the inner core of the Sun, the region where the matter density is very high. Given the dependence of $\varepsilon_{\alpha\beta}(x)$ with the neutron fraction, the analysis will show a non-trivial dependence with $\eta$.

For experiments measuring neutrinos created in the Earth, the matter effects are also relevant. Atmospheric neutrinos, created by the collision of cosmic rays on the Earth in an energy range that cover more than six orders of magnitude, from $\sim 100$ MeV to $\sim 100$ TeV, cross a big portion, or even the entire Earth, before arriving at the detector. Particularly relevant are the matter effect for the trajectories crossing the mantle for neutrinos with energies around $\sim 200$ MeV and $\sim 6$ GeV, since they get an enhancement in the flavor oscillation due to the MSW resonance. Also, for the most energetic part of the neutrino flux ($\geq 1$ TeV), the evolution is dominated by the matter effects. No flavor oscillation is possible for those energies in the standard picture. In the presence of NSI, the matter potential is no longer diagonal, which introduce a flavor oscillation that only depends on the distance traveled by the neutrino but not on its energy. Other experiments considered are Long-baseline accelerator, where the energy beam ranges from $\sim 0.6$ GeV to $\sim 7$ GeV, and their baselines are of the order of several 100 km. For those experiments, where the oscillation parameters are measured with high precision, the matter effect can be as large as $\sim 30$ %. Since our analysis is CP-conserving, we have not included the electron appearance channel. In addition to those experiments, we have also included the data from reactor experiments, which are not sensitive to the matter effect, but can establish an independent measurement of some standard parameters, like $\Delta m_{31}^2$ and $\theta_{13}$.

In addition to the oscillation experiments, NSI-NC can also be tested by measuring the NC neutrino scattering with low momentum transfer. This is the case of coherent neutrino-nucleus scattering measured in COHERENT [7]. The flux consists of muon and electron neutrinos created by stopping pions. The expected event rate is proportional to the coupling squared with the mediator [17]

$$Q_{w\alpha}^2 \propto \sum_i \left\{ \left[ Z_i(g_p^V + \varepsilon_{\alpha\alpha}^p) + N_i(g_n^V + \varepsilon_{\alpha\alpha}^n) \right]^2 + \sum_{\beta \neq \alpha} \left[ Z_i \varepsilon_{\alpha\beta}^p + N_i \varepsilon_{\alpha\beta}^n \right]^2 \right\}, \qquad (4.4)$$

where the sum over $i$ runs over the targets in the experiment, Cesium and Iodine. $Q_{w\alpha}^2$ introduce a linear dependence on each diagonal $\varepsilon_{\alpha\alpha}$, which allows to constrain each of

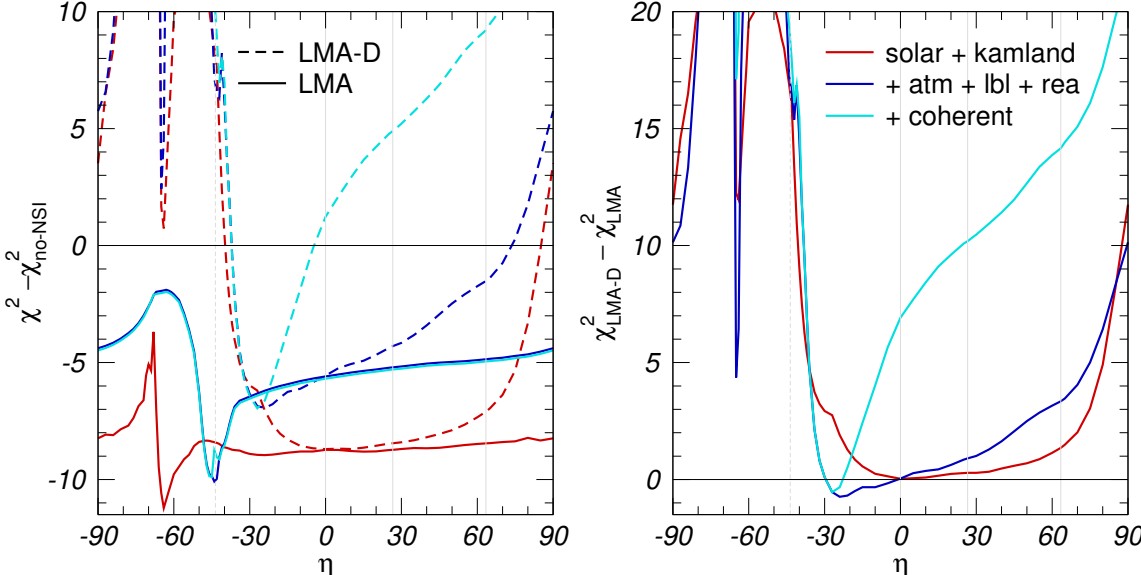

**Figure 41:** Left: $\Delta\chi^2$ between the standard ($\chi^2_{\text{no-NSI}}$) and the NSI scenario ($\chi^2(\eta)$) assuming the LMA solution (full lines) or the LAM-D solution (dash lines) for different data combinations as a function of the NSI quark coupling ($\eta$). Right: $\Delta\chi^2$ between the LMA and LMA-D solutions as a function of $\eta$.

them separately. The coupling also introduces a quadratic dependence on the non-diagonal elements.

Combining the different data sets, we have compared the preference of the data for the $3\nu$+NSI scenario as is shown in Fig. 41 (left) for the LMA solution (full lines) and the LMA-D solution (dash lines), as a function of the NSI quark coupling ($\eta$). The red lines indicate the results combining solar experiments with KamLAND. This analysis shows a preference for the $3\nu$+NSI scenario of $\sim 2\sigma$ for any value of $\eta$. The presence of new interactions is favored due to the tension between KamLAND and the solar experiments in the determination of $\Delta m^2_{21}$ [115]. KamLAND prefers higher values for the solar mass parameter than the solar experiments. If we compare the two possible solutions, the Dark solution is valid for almost the whole range of $\eta$, Fig. 41 (right). Adding the data from atmospheric, Long-baseline accelerator and reactor, we get a global analysis of oscillation data, whose results are given by the blue lines. The new data sets reduce preference for NSI, which is still favored for any value of $\eta$ over the standard scenario for the LMA solution. Regarding the LMA-D, the global analysis allows that solution at $3\sigma$ in the range $-38° \leq \eta \leq 87°$.

The results of the analysis combining oscillation experiments and COHERENT are shown in the cyan lines in Fig. 41. The main impact of scattering data over the results is disfavoring the LMA-D solution. The new dependence in $\varepsilon_{\alpha\beta}$ introduced by COHERENT [17] allows to break the degeneracy induced by Eq. 4.3. At $3\sigma$, LMA-D is allowed for $-38° \leq \eta \leq 14°$.

# 5 Neutrino Non-Standard Interactions in Dark Matter Direct Detection Experiments (Perez-Gonzalez)

Dark Matter (DM) is an unknown component present in the Universe which interacts very weakly or does not interact at all with light. Its presence, however, has been observed by its gravitational impact in distinct astrophysical and cosmological scales. Among the many candidates to be the DM, the Weakly Interacting Massive Particle (WIMP) paradigm has been extensively studied and tested by several experiments [116]. Specifically, direct detection experiments are searching for nuclear recoils produced by the possible interaction of WIMPs with nucleons. The recoil rate of the interaction between a WIMP $\chi$ with mass $m_\chi$ and $M$ nuclei with $Z$ protons and $N$ neutrons in the detector is usually parametrized as [117]

$$\frac{dR}{dE_R}\bigg|_\chi = M\frac{\rho_0}{2\,\mu_n^2\,m_\chi}\sigma_{\chi n}(Z+N)^2\mathcal{F}(E_R)^2\int_{v_{\min}}\frac{f(v)}{v}d^3v\,, \qquad (5.1)$$

where $\sigma_{\chi n}$ is the spin-independent WIMP-nucleon cross section, $\rho_0 = 0.3$ GeV/cm$^3$ the local DM density, $\mathcal{F}(E_R)$ the nuclear form factor, $\mu_n = m_n m_\chi/(m_n + m_\chi)$ the WIMP-nucleon system reduced mass, and $f(v)$ the WIMP velocity distribution [117]. Proposed large-exposure experiments such as DARWIN [118], ARGO [119] or CRESST [120] intend to explore even further the WIMP parameter space. However, they will face an irreducible background coming from neutrino interactions in such detectors. Neutrinos can produce a signal similar to a WIMP recoil event through the coherent elastic neutrino-nucleus scattering (CE$\nu$NS) process [121, 122]. The neutrino recoil rate from CE$\nu$NS in a direct detection experiment is

$$\frac{dR}{dE_R}\bigg|_\nu = M\int_{E_{\min}^\nu}\frac{d\Phi}{dE_\nu}\frac{d\sigma^\nu}{dE_R}dE_\nu, \qquad (5.2)$$

with $d\Phi/dE_\nu$ the incoming neutrino flux, $d\sigma^\nu/dE_R$ the CE$\nu$NS cross section [123] and $E_{\min}^\nu$ the minimun neutrino energy capable of produce a nuclear recoil with energy $E_R$. For the neutrino flux, we consider only solar and low-energy atmospheric contributions. The different solar neutrino flux components (pp, pep, hep,$^7$Be, $^8$B, $^{13}$N, $^{15}$O, and $^{17}$F) depend on the solar model; here we will consider the B16-GS98 Solar Model [124]. On the other hand, only the low-energy part of the atmospheric flux will be important here. We will consider the flux obtained considering FLUKA [125]. In order to define concretely at which point neutrinos become relevant for a given DM experiment, Billard et. al. [122] introduced the concept of *neutrino discovery limit*, also known as *neutrino floor*. Such discovery limit is defined as the minimum value of the WIMP cross-section for a given $m_\chi$ which has a 90 % probability to have a signal with $3\sigma$ significance over the neutrino background. In other words, the neutrino floor indicates when a direct detection experiment becomes systematics-limited by the neutrino background. Therefore, such discovery limit is highly dependent on the uncertainty of the neutrino fluxes, so any improvement on the measurement on these fluxes will certainly benefit the direct detection program. To describe the full behavior of the discovery limit, we consider an "academic" experiment with a huge exposure of

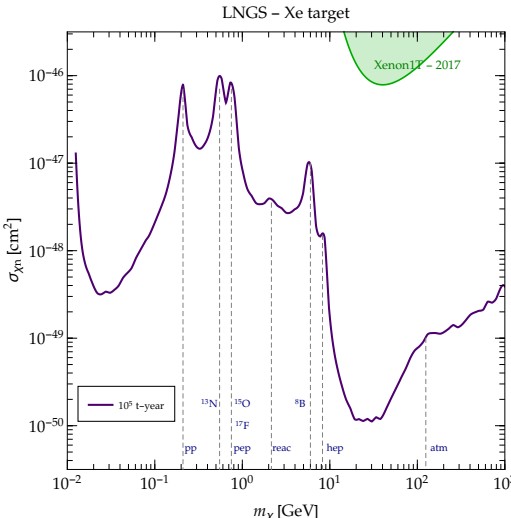

**Figure 51:** Standard Model Neutrino discovery limit of an "academic" experiment with $10^5$ ton-years exposure and $E_{\text{th}} = 0.01$ eV energy threshold.

$10^5$ ton-years and an artificial energy threshold of $E_{\text{th}} = 0.01$ eV. In Fig. 51 we present the total neutrino discovery limit for such toy experiment considering Xenon as a target. The discovery limit presents peaks in different positions in the parameter space; each peak indicates where the different component of the neutrino background mimics significantly a specific set of WIMP parameters [122, 126–128]. For instance, if the WIMP had a mass of 6 GeV and a cross-section of $\sigma_{\chi n} \sim 10^{-47}\text{cm}^2$, the spectrum in the toy experiment would be highly mimicked by the $^8$B solar component. Furthermore, we see that low-energetic components mock the spectra for lower WIMP masses, while the WIMP cross-section for each peak is related to the total neutrino flux. Finally, let us notice that there are several works on possible ways to distinguish between WIMP and neutrino spectra [129–133].

The existence of beyond Standard Model physics affecting the CE$\nu$NS process will modify the neutrino background in direct detection experiments. We can classify the new physics in two different categories, models which are flavor independent or flavor dependent. Let us consider flavor independent models in the form of simplified models [134]. In this case, we can assume the existence of additional mediators, a vector $V_\mu$ and a scalar $S$, which couple with neutrinos and, possibly, the DM, taken here to be a Dirac fermion $\chi$,

$$\mathcal{L}_{\text{vec}} = V_\mu \sum_{f=\nu,\chi} \overline{f}\gamma^\mu(g_V^f + g_A^f\gamma_5)f + \frac{1}{2}m_V^2 V_\mu V^\mu \tag{5.3a}$$

$$\mathcal{L}_{\text{sc}} = S \sum_{f=\nu,\chi} g_S^f \overline{f} f - \frac{1}{2}m_S^2 S^2. \tag{5.3b}$$

In Fig. 52 we present the neutrino discovery limit including NSI for some benchmark points of these two flavor-independent models. We observe that the different Lorentz structures play significant roles in the modification of the neutrino background. First, in the vector scenario, the neutrino floor is shifted above or below the Standard Model case.

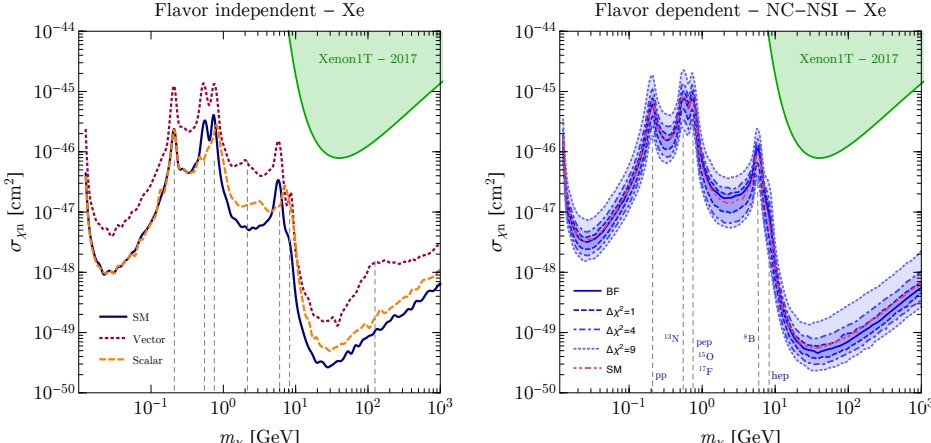

**Figure 52:** Neutrino discovery limit for a Xe experiment considering the presence of new physics, flavor dependent scenarios (left) and NC-NSI (right).

On the other hand, the scalar mediator changes the position of the peaks for neutrino components with larger energies, as the CE$\nu$NS cross section in this scenario is modified by an additive factor [134]. As a scenario in which the new physics is flavor dependent we can consider the effective approach of the neutral-current Non-Standard Interactions (NC-NSI) [121, 130, 133, 135], parametrized as in Eq. (1.1). The neutrino recoil rate should now include also neutrino oscillations, as the CE$\nu$NS cross section is now flavor dependent [136],

$$\left.\frac{dR}{dE_R}\right|_\nu = N_T \sum_{\nu_\alpha, \nu_\beta} \int_{E^\nu_{\min}} \left.\frac{d\Phi}{dE_\nu}\right|_{\nu_\alpha} P(\nu_\alpha \to \nu_\beta, E_\nu) \left.\frac{d\sigma^\nu(\nu_\beta)}{dE_R}\right|_{\rm NSI} dE_\nu. \qquad (5.4)$$

Considering the results obtained by Coloma et. al. [18], we present in Fig. 52 the best-fit and the $\{1\sigma, 2\sigma, 3\sigma\}$ regions of the neutrino discovery limit in a Xenon target experiment. We can see here that the modification on the neutrino discovery limit still allowed is about $\sim 5$ the value of the SM. In Fig. 53 we show the same regions for the proposed ARGO experiment, together with the sensitivity proposed. If a direct detection experiment has a positive signal inside the neutrino floor $3\sigma$ region, such recoil could also be interpreted as the result of a neutrino scattering produced by new physics. Therefore, in principle, a direct detection experiment could also be used to put constraints on NC-NSI. Nevertheless, we should notice that dedicated experiments trying to measure the CE$\nu$NS would improve the bounds more significantly than a future direct detection experiment. Still, it is worth noting that a precise measurement of the coherent scattering will aid DM experiments, as they will add information on the neutrino background even under the assumption of the existence of new physics in the form of NC-NSI.

## 6  Probing NSI at Colliders (Gonçalves)

Physics beyond the standard model may induce significant deviations in the couplings involving neutrinos generally referred to as Non-Standard neutrino Interactions [1]. While

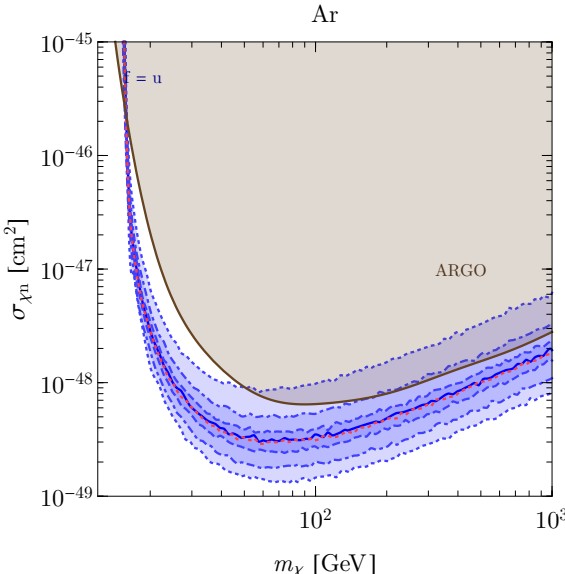

**Figure 53:** Neutrino discovery limit for the proposed ARGO experiment. The shaded regions correspond to predictions with NC-NSI $\Delta\chi^2 \leq 1, 4, 9$.

these effects are more intensively looked for in neutrino experiments, collider experiments can offer an interesting complementary probe to these new physics terms [137–139].

NSI can be generally parametrized by contact interactions as in Eq. (1.1). These new physics contributions can be induced by higher dimensional operators that are gauge invariant under the SM symmetries. They can be obtained at dimension-8 via operators of the form

$$\frac{1}{\Lambda^4} \left( \overline{HL}_\alpha \gamma_\mu HL_\beta \right) \left( \bar{q}\gamma^\mu P_X q \right) , \tag{6.1}$$

where $L$ and H are respectively the SM lepton and Higgs doublets [140]. Thus, Eq. (1.1) can be generated at low energies without requiring charged current interactions of similar strengths[2].

Although Eq. (1.1) can be safely applied to oscillation experiments, where the momentum transfer is negligible, $Q_{tr} \to 0$, it is not necessarily so for the Large Hadron Collider (LHC). At the energy scales and couplings probed at the LHC, the validity of the Effective Field Theory (EFT) approach can no longer be guaranteed. This discussion displays relevant similarities to the Dark Matter (DM) literature [141]. Namely, DM direct detection experiments can typicallly bound new physics effects in the EFT framework, accounting for EFT interactions such as $1/\Lambda^2 \left( \bar{\chi}\gamma_\mu\chi \right) \left( \bar{q}\gamma^\mu q \right)$, however this does not usually hold true for the LHC searches. Simplified models for DM have been shown to be a more adequate approach for collider studies. This includes the DM mass regime $m_\chi \to 0$ where the presently illustrated DM EFT operator maps into Eq. (1.1). Thus, the same framework can be

---

[2]NSI contributions can also be generated with dimension-6 operators, however they induce strongly constrained charged current interactions if one does not assume unnatural Wilson coefficient cancelations.

analogously applied to NSI searches, for instance, via the s-channel simplified model [142]

$$\mathcal{L}_{\text{NSI}}^{\text{Simp}} = \left( g_\nu^{\alpha\beta} \left( \bar{\nu}_\alpha \gamma^\mu \nu_\beta \right) + g_q^Y \bar{q} \gamma^\mu P_Y q \right) X_\mu \,. \qquad (6.2)$$

When the momentum transfer is significantly smaller than the mediator mass, $Q_{tr} \ll m_X$, the mediator $X$ can be integrated out and we can map the effective interaction to the simplified model description as $\varepsilon_{\alpha\beta}^{qY} = (g_\nu)_{\alpha\beta} g_q^Y / (2\sqrt{2} m_X^2 G_F)$.

At the LHC, the NSI display the characteristic mono-jet signature produced via the QCD initial state radiation of quarks and gluons, $pp \to \bar{\nu}_\alpha \nu_\beta j$ with $j = q, \bar{q}, g$ [137, 138]. The searches result into relevant sensitivity for heavy mediators, $m_X \gtrsim \mathcal{O}(100 \text{ GeV})$, with a sensitivity reach of the order of $\varepsilon \gtrsim \mathcal{O}(10^{-1} - 10^{-2})$ with 19.5 fb$^{-1}$ of data at $\sqrt{s} = 8$ TeV [142]. Differently than neutrino experiments, distinct choices of $(\alpha, \beta)$ are indistinguishable at the LHC, leading to the same observables. Another significant difference between neutrino and collider experiments is that oscillation experiments only relevantly probe vector couplings. Contrarily, the LHC can be sensitive to both vector and axial new physics contributions. Thus, the LHC probe to NSI renders further information that can potentially contribute to the global new physics analyses.

The major limitation for the NSI bounds at colliders is associated to the overwhelming SM backgrounds, $pp \to Z(\nu\nu)j$ and $pp \to W(\ell\nu)j$. However, the combination of experimental and theoretical efforts are resulting into significant improvements for the general mono-jet searches [143]. Recent studies further explore background control regions and state of the art of Monte Carlo simulation, resulting into suppressed background uncertainties and augmented sensitive to new physics. These improvements pave the way for more constraining and robust NSI (DM) searches with the forthcoming LHC data.

## 7    Non-Standard Interactions in Radiative Neutrino Mass Models (Thapa)

Models of radiative neutrino mass generation require new scalars and/or fermions. These new BSM particles explicitly break lepton number [144] and could give rise to rich LFV processes, as well as significant neutrino NSI with matter fields. We define *type-I* radiative models as those with at least one SM particle inside the loop, and *type-II* radiative models as those with no SM particle inside the loop. We have analyzed both classes of models, but find that tree-level NSI arises only in case of type-I radiative models. In particular, we focus on the Zee model and its colored variant with leptoquarks (LQs), that give rise to observable NSI, while being consistent with direct and indirect constraints from colliders, precision data and LFV searches. We then compare these model predictions for NSI with the recent global fit constraints from neutrino oscillation and scattering experiments, and discuss the future sensitivity of long-baseline experiments, such as DUNE.

**Zee Model:** Zee Model [145] is one of the simplest extensions of the SM that can generate neutrino mass radiatively at the one loop level (see Fig. 71 left). In this model, the new physics scale could be around the electroweak scale. In addition to the SM-like Higgs field $H_1(1, 2, 1/2)$, two additional scalars, $\eta^+(1, 1, 1)$ and $H_2(1, 2, 1/2)$, are introduced, where the charges in the parentheses are under the SM gauge group $SU(3)_c \times SU(2)_L \times U(1)_Y$.

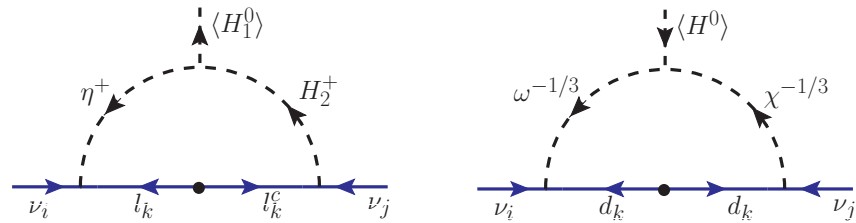

**Figure 71:** Neutrino mass generation at one-loop level: Zee model (left) and a colored variant with LQs (right).

This particle content leads to the Yukawa Lagrangian

$$-\mathcal{L}_Y \;\supset\; f_{\alpha\beta}L_\alpha^i L_\beta^j \varepsilon_{ij}\eta^+ + \widetilde{Y}_{\alpha\beta}\widetilde{H}_1^i L_\alpha^j \ell_\beta^c \varepsilon_{ij} + Y_{\alpha\beta}\widetilde{H}_2^i L_\alpha^j \ell_\beta^c \varepsilon_{ij} + \mu\, H_1^i\, H_2^j \varepsilon_{ij}\,\eta^- + \text{H.c.} \quad (7.1)$$

Here $L$ denotes the $SU(2)_L$ lepton doublet, $\{\alpha,\beta\}$ are family indices, $\{i,j\}$ are $SU(2)_L$ indices, and $\varepsilon_{ij}$ is the $SU(2)_L$ antisymmetric tensor. The neutrino mass matrix reads as:

$$M_\nu = \frac{1}{16\pi^2}\sin 2\varphi \log\left(\frac{m_{h^+}^2}{m_{H^+}^2}\right)\left(f M_\ell Y + Y^T M_\ell f^T\right), \quad (7.2)$$

where $M_\ell$ is the diagonal charged lepton mass matrix, $\varphi$ is the mixing between the singlet and doublet charged scalars (which is proportional to the cubic term $\mu$) and $h^+$, $H^+$ are the physical charged scalar fields. Small neutrino mass can be realized with either $Y \sim \mathcal{O}(1)$ and $f \ll 1$, or vice versa in Eq. (7.2). The latter case (with large $f$) would give rise to NSI from pure $\eta^+$ exchange [146], which is very small due to muon and tau decay constraints, whereas the former case (with large $Y$) would give rise to NSI from the mixed $h^\pm$ exchange, which can be large, as we show here [147]. Since $h^\pm$ is leptophilic, for neutrinos propagating in the ordinary matter, we have following canonical NSI parameter, according to the definition given in Eq. (1.1):

$$\varepsilon_{\alpha\beta} \equiv \varepsilon_{\alpha\beta}^{(h^+)} + \varepsilon_{\alpha\beta}^{(H^+)} = \frac{1}{4\sqrt{2}G_F}Y_{\alpha e}Y_{\beta e}^\star\left(\frac{\sin^2\varphi}{m_{h^+}^2} + \frac{\cos^2\varphi}{m_{H^+}^2}\right). \quad (7.3)$$

Thus for observable NSI, we require either large Yukawa coupling $Y_{\alpha e}$ or large mixing angle $\sin\varphi$, and small singly-charged scalar masses $m_{h^+}$ and/or $m_{H^+}$.

We show in Fig. 72(a) the LEP and LHC constraints on the light charged scalar $h^\pm$ in the Zee model. Once produced on-shell, $h^+$ decays into the $\nu_\alpha\ell_\beta$ final leptonic states with branching ratio $\text{BR}_{\beta\nu}$, via the Yukawa coupling $Y_{\alpha\beta}$. We assume there is no coupling between the charged scalar and quarks, in order to eschew the stringent constraints from meson decays. Since our goal is to obtain large NSI, electrons must interact with charged scalar, i.e. $Y_{\alpha e} \neq 0$ for at least one flavor $\alpha$. It is to be noted that both $Y_{\alpha e}$ and $Y_{\alpha\mu}$ cannot be large simultaneously due to stringent LFV limits. So we consider the case where $\text{BR}_{e\nu} + \text{BR}_{\tau\nu} = 1$ and $\text{BR}_{\mu\nu}$ is negligible. We have explicit fits in Ref. [147] to show that this choice of Yukawa couplings is consistent with the observed neutrino oscillation data. For this choice of Yukawa couplings, we can reinterpret the LEP [148] and LHC [149, 150]

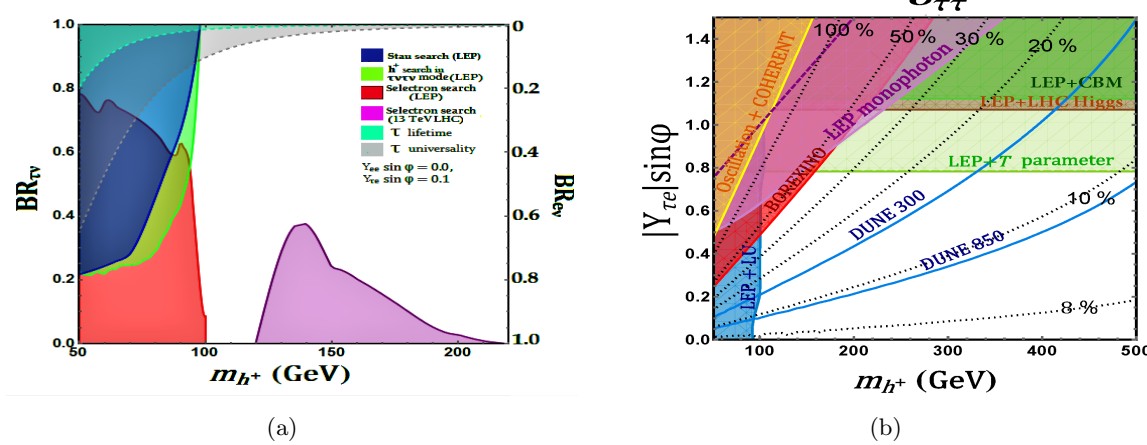

**Figure 72:** (a) Collider constraints on light charged higgs in the Zee model. (b) $\varepsilon_{\tau\tau}$ prediction. The shaded regions are excluded. See text and Ref. [147] for more details.

selectron and stau searches in the massless neutralino limit to derive limits on the charged Higgs mass. For $\mathrm{BR}_{\tau\nu} \neq 0$, a slightly stronger limit can be obtained from the LEP searches for the charged Higgs boson pairs in the two-Higgs-doublet model [151] in both $\tau\nu\tau\nu$ and $c\bar{s}\tau\nu$ channels. Note that the LEP limits derived here are somewhat more stringent than those reported in Ref. [152]. The tau lifetime and lepton universality (LU) in tau decay [35] are also modified due to the $h^+$-induced tau decay: $\tau \to \nu h^{+*} \to e\nu\nu$, which impose further constraints on large $\mathrm{BR}_{\tau\nu}$. All these limits are summarized in Fig. 72(a) for the benchmark choice of $Y_{\tau e} \sin\varphi = 0.1$, which shows that $m_h^+$ as low as 100 GeV is allowed. Here we have chosen $Y_{ee} = 0$ to avoid the stringent constraints from lepton universality in $W$ decays [35], which would otherwise restrict $m_h^+$ to be larger than 130 GeV.

Taking into account these constraints, as well as the theoretical constraints from electroweak $T$ parameter and charge breaking minima (CBM), as well as the LHC Higgs constraints, we show the model predictions for $\varepsilon_{\tau\tau}$ in Fig. 72(b). Here we have chosen $m_{H^+} = 700$ GeV, so its contribution to NSI is sub-dominant. Also shown here are the additional LEP constraints from monophoton searches [153] and from neutrino scattering experiment BOREXINO [154], as well as the global fit constraints on $\varepsilon_{\tau\tau}$ [24] from neutrino oscillation plus COHERENT data. We find that $\varepsilon_{\tau\tau}$ as large as 43% is still allowed, with the upper limit coming from BOREXINO. It can be probed down to 14.5% (9.5%) at DUNE in the 300 (850) kt.MW.yr configuration. For the maximum allowed NSI in other flavors, see Table 71.

**One-Loop LQ Model:** This is a variant of the Zee model where, in addition to the SM Higgs doublet $H(1, 2, 1/2) = (H^+, H^0)$, two $SU(3)_c$-colored scalar LQs, $\Omega(3, 2, 1/6) = \left(\omega^{2/3}, \omega^{-1/3}\right)$ and $\chi(3, 1, -1/3)$ are introduced [155, 156]. Relevant Lagrangian terms are

$$\begin{aligned} -\mathcal{L}_Y \supset\ & \lambda_{\alpha\beta} L^i_\alpha d^c_\beta \Omega^j \varepsilon_{ij} + \lambda'_{\alpha\beta} L^i_\alpha Q^j_\beta \chi^\star \varepsilon_{ij} + \mu H^\dagger \Omega \chi^\star + \mathrm{H.c.} \\ =\ & \lambda_{\alpha\beta} \left(\nu_\alpha d^c_\beta \omega^{-1/3} - \ell_\alpha d^c_\beta \omega^{2/3}\right) + \lambda'_{\alpha\beta} \left(\nu_\alpha d_\beta - \ell_\alpha u_\beta\right) \chi^\star \end{aligned}$$

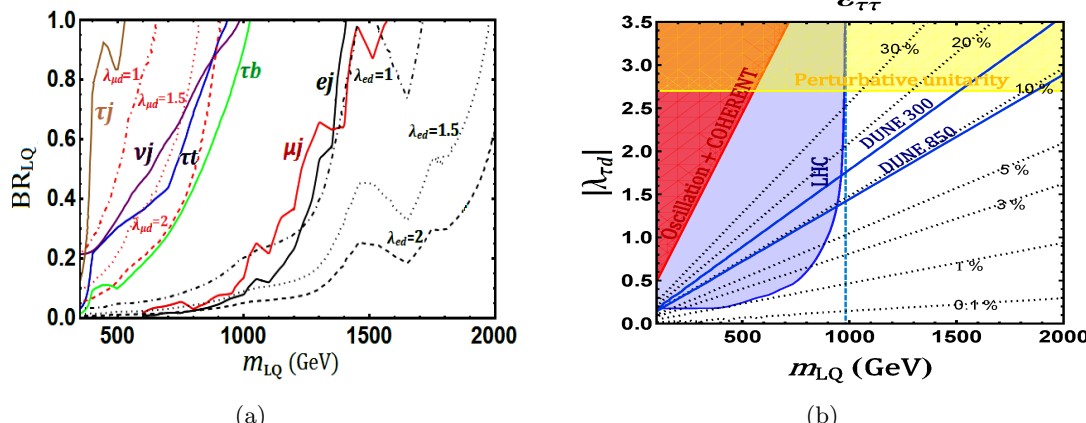

(a)                                                    (b)

**Figure 73:** (a) LHC constraints on scalar LQ from pair and single production. (b) $\varepsilon_{\tau\tau}$ prediction compared with various constraints. The shaded regions are excluded. See text and Ref. [147] for more details.

$$+\mu\left(\omega^{2/3}H^- + \omega^{-1/3}\overline{H}^0\right)\chi^\star + \text{H.c.} \qquad (7.4)$$

The breaking of lepton number by two units occurs due to the cubic term. Neutrino mass matrix is induced in one-loop as shown in Fig. 71 (right). The mass matrix reads as:

$$M_\nu = \frac{3\sin 2\alpha}{32\pi^2}\log\left(\frac{m_1^2}{m_2^2}\right)\left(\lambda M_d \lambda'^T + \lambda' M_d \lambda^T\right), \qquad (7.5)$$

where $\sin 2\alpha$ (which is proportional to $\mu$) represents the mixing between $\omega^{-1/3}$ and $\chi^{-1/3}$, $m_{1,2}$ being their physical mass eigenstates, and $M_d$ is the down-type quark mass matrix. The LQs $\omega^{-1/3}$ and $\chi^{-1/3}$ in this model induce NSI at tree level. Significant NSI can be obtained by taking either of the Yukawas, $\lambda$ or $\lambda'$, of order 1. We obtain

$$\varepsilon_{\alpha\beta} \equiv 3\varepsilon_{\alpha\beta}^d = \frac{3}{4\sqrt{2}\,G_F}\left(\frac{\lambda^\star_{\alpha d}\lambda_{\beta d}}{m_\omega^2} + \frac{\lambda'^\star_{\alpha d}\lambda'_{\beta d}}{m_\chi^2}\right). \qquad (7.6)$$

The LHC constraints on $\omega^{-1/3}$ and $\chi^{-1/3}$ are shown in Fig. 73(a) from various final states as a function of the branching ratio to that channel (while varying the other Yukawa couplings to make the total branching ratio one). The limits from pair-production are independent of the Yukawa coupling $\lambda$, while those from single production depend on $\lambda$. Fig. 73(b) shows the $\varepsilon_{\tau\tau}$ predictions (valid for both $\omega$ and $\chi$ LQs) compared with the LHC, global fit and perturbativity constraints. We can get maximum $\varepsilon_{\tau\tau}$ of 34.3% in this model. It can be probed down to 14.5% (9.5%) at DUNE in the 300 (850) kt.MW.yr configuration.

Table 71 summarizes the maximum allowed tree-level NSI in each flavor for all type-I radiative neutrino mass models. See Ref. [147] for more details.

| Term | $\mathcal{O}$ | Model | Loop level | $\mathcal{S}/\mathcal{F}$ | New particles | Max NSI @ tree-level | | | | | |
|---|---|---|---|---|---|---|---|---|---|---|---|
| | | | | | | $\|\varepsilon_{ee}\|$ | $\|\varepsilon_{\mu\mu}\|$ | $\|\varepsilon_{\tau\tau}\|$ | $\|\varepsilon_{e\mu}\|$ | $\|\varepsilon_{e\tau}\|$ | $\|\varepsilon_{\mu\tau}\|$ |
| $L\ell^c\Phi^\star$ | $\mathcal{O}_2^2$ | Zee [145] | 1 | $\mathcal{S}$ | $\eta^+(\mathbf{1},\mathbf{1},1),\ \Phi_2(\mathbf{1},\mathbf{2},1/2)$ | 0.08 | 0.038 | 0.43 | $\mathcal{O}(10^{-5})$ | 0.0056 | 0.0034 |
| $LL\eta$ | $\mathcal{O}_9$ | Zee-Babu [48, 157] | 2 | $\mathcal{S}$ | $h^+(\mathbf{1},\mathbf{1},1),\ k^{++}(\mathbf{1},\mathbf{1},2)$ | 0 | 0.0009 | 0.003 | 0 | 0 | 0.003 |
| | $\mathcal{O}_9$ | KNT [158] | 3 | $\mathcal{S}$ $\mathcal{F}$ | $\eta_1^+(\mathbf{1},\mathbf{1},1),\ {\color{red}\eta_2^+}(\mathbf{1},\mathbf{1},1)$, ${\color{red}N}(\mathbf{1},\mathbf{1},0)$ | 0 | 0.0009 | 0.003 | 0 | 0 | 0.003 |
| | $\mathcal{O}_9$ | 1S-1S-1F [159] | 3 | $\mathcal{S}$ $\mathcal{F}$ | $\eta_1(\mathbf{1},\mathbf{1},1),\ \eta_2(\mathbf{1},\mathbf{1},3)$, $F(\mathbf{1},\mathbf{1},2)$ | 0 | 0.0009 | 0.003 | 0 | 0 | 0.003 |
| | $\mathcal{O}_2^1$ | 1S-2VLL [160] | 1 | $\mathcal{S}$ $\mathcal{F}$ | $\eta(\mathbf{1},\mathbf{1},1)$, $\Psi(\mathbf{1},\mathbf{2},-3/2)$ | 0 | 0.0009 | 0.003 | 0 | 0 | 0.003 |
| SM | $\mathcal{O}_2'$ | AKS [161] | 3 | $\mathcal{S}$ $\mathcal{F}$ | $\Phi_2(\mathbf{1},\mathbf{2},1/2),\ \eta^+(\mathbf{1},\mathbf{1},1),\ {\color{red}\eta^0}(\mathbf{1},\mathbf{1},0)$, ${\color{red}N}(\mathbf{1},\mathbf{1},0)$ | $\mathcal{O}(10^{-10})$ | $\mathcal{O}(10^{-10})$ | $\mathcal{O}(10^{-10})$ | $\mathcal{O}(10^{-10})$ | $\mathcal{O}(10^{-10})$ | $\mathcal{O}(10^{-10})$ |
| | $\mathcal{O}_9$ | Cocktail [162] | 3 | $\mathcal{S}$ | ${\color{red}\eta^+}(\mathbf{1},\mathbf{1},1),\ k^{++}(\mathbf{1},\mathbf{1},2),\ {\color{red}\Phi_2}(\mathbf{1},\mathbf{2},1/2)$ | 0 | 0 | 0 | 0 | 0 | 0 |
| $W/Z$ | $\mathcal{O}_1'$ | MRIS [163] | 1 | $\mathcal{F}$ | $N(\mathbf{1},\mathbf{1},0),\ S(\mathbf{1},\mathbf{1},0)$ | 0.024 | 0.022 | 0.10 | 0.0013 | 0.0035 | 0.012 |
| $L\Omega d^c$ $(LQ\chi^\star)$ | $\mathcal{O}_3^8$ | LQ variant of Zee [155] | 1 | $\mathcal{S}$ | $\Omega(\mathbf{3},\mathbf{2},1/6),\ \chi(\mathbf{3},\mathbf{1},-1/3)$ | 0.004 | 0.216 | 0.343 | $\mathcal{O}(10^{-7})$ | 0.0036 | 0.0043 |
| | $\mathcal{O}_8^4$ | 2LQ-1LQ [164] | 2 | $\mathcal{S}$ | $\Omega(\mathbf{3},\mathbf{2},1/6),\ \chi(\mathbf{3},\mathbf{1},-1/3)$ | (0.0069) | (0.0086) | 0.343 | $\mathcal{O}(10^{-7})$ | 0.0036 | 0.0043 |
| | $\mathcal{O}_3^3$ | 2LQ-1VLQ [165] | 2 | $\mathcal{S}$ $\mathcal{F}$ | $\Omega(\mathbf{3},\mathbf{2},1/6)$, $U(\mathbf{3},\mathbf{1},2/3)$ | 0.004 | 0.216 | 0.343 | $\mathcal{O}(10^{-7})$ | 0.0036 | 0.0043 |
| $L\Omega d^c$ | $\mathcal{O}_3^6$ | 2LQ-3VLQ [160] | 1 | $\mathcal{S}$ $\mathcal{F}$ | $\Omega(\mathbf{3},\mathbf{2},1/6)$, $\Sigma(\mathbf{3},\mathbf{3},2/3)$ | 0.004 | 0.216 | 0.343 | $\mathcal{O}(10^{-7})$ | 0.0036 | 0.0043 |
| | $\mathcal{O}_8^2$ | 2LQ-2VLL [160] | 2 | $\mathcal{S}$ $\mathcal{F}$ | $\Omega(\mathbf{3},\mathbf{2},1/6)$, $\psi(\mathbf{1},\mathbf{2},-1/2)$ | 0.004 | 0.216 | 0.343 | $\mathcal{O}(10^{-7})$ | 0.0036 | 0.0043 |
| | $\mathcal{O}_8^3$ | 2LQ-2VLQ [160] | 2 | $\mathcal{S}$ $\mathcal{F}$ | $\Omega(\mathbf{3},\mathbf{2},1/6)$, $\xi(\mathbf{3},\mathbf{2},7/6)$ | 0.004 | 0.216 | 0.343 | $\mathcal{O}(10^{-7})$ | 0.0036 | 0.0043 |
| $L\Omega d^c$ $(LQ\bar{p})$ | $\mathcal{O}_3^9$ | Triplet-Doublet LQ [160] | 1 | $\mathcal{S}$ | $\rho(\mathbf{3},\mathbf{3},-1/3),\ \Omega(\mathbf{3},\mathbf{2},1/6)$ | 0.0059 | 0.0249 | 0.517 | $\mathcal{O}(10^{-8})$ | 0.0050 | 0.0038 |
| $LQ\chi^\star$ | $\mathcal{O}_{11}$ | LQ/DQ variant Zee-Babu [166] | 2 | $\mathcal{S}$ | $\chi(\mathbf{3},\mathbf{1},-1/3)$ , $\Delta(\mathbf{6},\mathbf{1},-2/3)$ | 0.0069 | 0.0086 | 0.343 | $\mathcal{O}(10^{-7})$ | 0.0036 | 0.0043 |
| | $\mathcal{O}_{11}$ | Angelic [167] | 2 | $\mathcal{S}$ $\mathcal{F}$ | $\chi(\mathbf{3},\mathbf{1},1/3)$, $F(\mathbf{8},\mathbf{1},0)$ | 0.0069 | 0.0086 | 0.343 | $\mathcal{O}(10^{-7})$ | 0.0036 | 0.0043 |
| | $\mathcal{O}_{11}$ | LQ variant of KNT [168] | 3 | $\mathcal{S}$ $\mathcal{F}$ | $\chi(\mathbf{3},\mathbf{1},-1/3),\ {\color{red}\chi_2}(\mathbf{3},\mathbf{1},-1/3)$, ${\color{red}N}(\mathbf{1},\mathbf{1},0)$ | 0.0069 | 0.0086 | 0.343 | $\mathcal{O}(10^{-7})$ | 0.0036 | 0.0043 |
| $Lu^c\delta$ $(LQ\bar{p})$ | $\mathcal{O}_3^4$ | 1LQ-2VLQ [160] | 1 | $\mathcal{S}$ $\mathcal{F}$ | $\chi(\mathbf{3},\mathbf{1},-1/3)$, $Q(\mathbf{3},\mathbf{2},-5/6)$ | | | | | | |
| | $\mathcal{O}_2'$ | 3LQ-2LQ-1LQ (New) | 1 | $\mathcal{S}$ | $\bar{\rho}(\bar{\mathbf{3}},\mathbf{3},1/3),\ \delta(\mathbf{3},\mathbf{2},7/6),\ \xi(\mathbf{3},\mathbf{1},2/3)$ | 0.004 (0.0059) | 0.216 (0.007) | 0.343 (0.517) | $\mathcal{O}(10^{-7})$ | 0.0036 (0.005) | 0.0043 (0.0038) |
| $Lu^c\delta$ | $\mathcal{O}_{d=13}$ | 3LQ-2LQ-2LQ(New) | 2 | $\mathcal{S}$ | $\delta(\mathbf{3},\mathbf{2},7/6),\ \Omega(\mathbf{3},\mathbf{2},1/6),\hat{\Delta}(\mathbf{6}^\star,\mathbf{3},-1/3)$ | 0.004 | 0.216 | 0.343 | $\mathcal{O}(10^{-7})$ | 0.0036 | 0.0043 |
| $LQ\bar{p}$ | $\mathcal{O}_3^5$ | 3LQ-2VLQ [160] | 1 | $\mathcal{S}$ $\mathcal{F}$ | $\bar{\rho}(\bar{\mathbf{3}},\mathbf{3},-1/3)$, $Q(\mathbf{3},\mathbf{2},-5/6)$ | 0.0059 | 0.0007 | 0.517 | $\mathcal{O}(10^{-7})$ | 0.005 | 0.0038 |
| | | All Type-II Radiative models | | | | 0 | 0 | 0 | 0 | 0 | 0 |

**Table 71:** A comprehensive summary of type-I radiative neutrino mass models, with the new particle content and their ($SU(3)_c$, $SU(2)_L$, $U(1)_Y$) charges, and the maximum tree-level NSI allowed in each model. Red-colored exotic particles are odd under a $Z_2$ symmetry. $\mathcal{S}$ and $\mathcal{F}$ represent scalar and fermion fields respectively.

# 8 Confronting Neutrino Mass Generation Mechanism with MiniBooNE Anomaly (Jana)

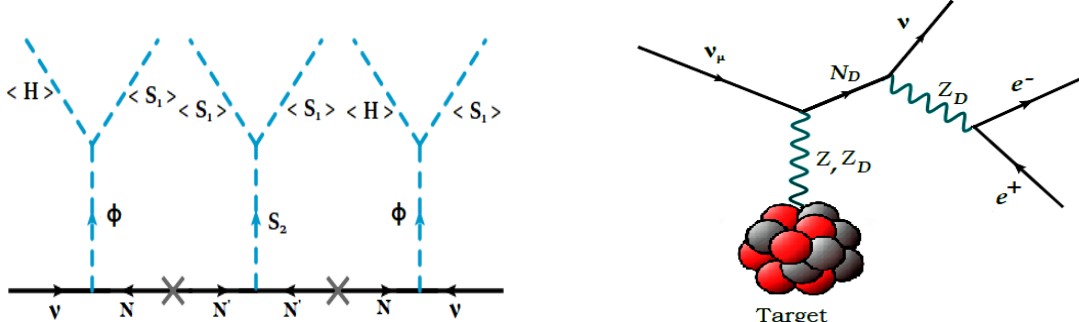

**Figure 81:** Left: diagram for the dynamically induced light neutrino masses in our model; Right: contributions to the cross section that in our model gives rise to MiniBooNE's excess of electron-like events.

One of the most robust proofs that points to an important inadequacy of the SM is the existence of non-zero tiny neutrino masses. In the SM, due to the absence of suitable right-handed partner, it is forbidden to add a renormalizable mass term to the SM for the neutrinos. Non-zero neutrino masses and large neutrino mixing demands new physics beyond the SM. A more 'natural' way to generate tiny neutrino masses involve the inclusion of new states that, once integrated out, generate the dimension five Weinberg operator

$$\mathcal{O}_5 = \frac{c}{\Lambda}LLHH. \tag{8.1}$$

This is embodied by the so-called seesaw mechanisms [45, 46, 171–178]. However, the scale of new physics behind the neutrino mass generation mechanism can be anywhere. Despite numerous searches for neutrino mass models (at TeV scale) at high-energy colliders [108, 109, 113, 114, 179–193], no compelling evidence has been found so far. Then, the following question may arise. Is it really sufficient to search for new physics scale behind neutrino mass generation mechanism at the LHC only? The new physics scale behind neutrino mass generation mechanism might be at low scale and which is less sensitive to high energy collider experiments. It may show up at low energy neutrino experiments at near future and which is not explored in the literature in great detail.

Most of the models for neutrino mass generation have one of the two following features: (i) The model is realized at very high scales, or (ii) the model is based on explicit breaking of lepton number or other symmetries that protect neutrino masses (e.g. in TeV scale type II or inverse seesaw models). We propose a new model [194] to connect the generation of neutrino masses to a light dark sector, charged under a new $U(1)_\mathcal{D}$ dark gauge symmetry. We introduce the minimal number of dark fields to obtain an anomaly free theory with spontaneous breaking of the dark symmetry and obtain automatically the inverse seesaw Lagrangian. Neutrino masses are generated via dimension 9 operator, and hence, in this way, we are able to lower the scale of neutrino mass generation below the electroweak (EW)

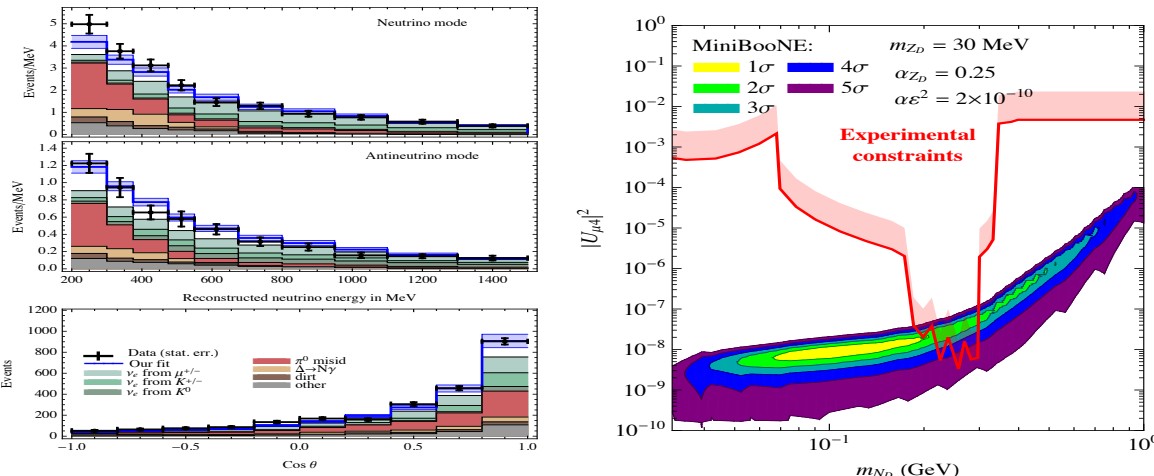

**Figure 82:** Left: The MiniBooNE electron-like event data [169] in the neutrino (top panel) and antineutrino (middle panel) modes as a function of $E_\nu^{rec}$, as well as the $\cos\theta$ distribution (bottom panel) for the neutrino data. Note that the data points have only statistical uncertainties, while the systematic uncertainties from the background are encoded in the light blue band.. The predictions of our benchmark point $m_{N_\mathcal{D}} = 420$ MeV, $m_{Z_\mathcal{D}} = 30$ MeV, $|U_{\mu 4}|^2 = 9 \times 10^{-7}$, $\alpha_\mathcal{D} = 0.25$ and $\alpha\varepsilon^2 = 2 \times 10^{-10}$ are also shown as the blue lines. Right: Region of our model in the $|U_{\mu 4}|^2$ versus $m_{N_\mathcal{D}}$ plane satisfying MiniBooNE data at $1\sigma$ to $5\sigma$ CL, for the hypothesis $m_{Z_\mathcal{D}} = 30$ MeV, $\alpha_{Z_\mathcal{D}} = 0.25$ and $\alpha\varepsilon^2 = 2 \times 10^{-10}$. The region above the red curve is excluded at 99% CL by meson decays, the muon decay Michel spectrum and lepton universality [108, 170].

one by resorting to a dynamical gauge symmetry breaking of this new sector. In addition, the so-called $\mu$-term of the inverse seesaw is dynamically generated and technically natural in this framework. The dark sector is mostly secluded from experimental scrutiny, as it only communicates with the SM by mixing among scalars, among neutrinos and dark fermions, and through kinetic and mass mixing between the gauge bosons. This scheme has several phenomenological consequences at lower energies, and in particular it offers a natural explanation [195] for the long-standing excess of electron-like events reported by the MiniBooNE collaboration[169].

Our proposal to explain MiniBooNE's low energy excess from the production and decay of a dark neutrino relies on the fact that MiniBooNE cannot distinguish a collimated $e^+e^-$ pair from a single electron. Muon neutrinos produced in the beam would up-scatter on the mineral oil to dark neutrinos, which will subsequently lead to $Z_\mathcal{D} \to e^+e^-$ as shown schematically in Fig. 81. If $N_\mathcal{D}$ is light enough, this up-scattering in $CH_2$ can be coherent, enhancing the cross section. To take that into account, we estimate the up-scattering cross section to be

$$\frac{d\sigma_{total}/dE_r}{\text{proton}} = \frac{1}{8}F^2(E_r)\frac{d\sigma_C^{coh}}{dE_r} + \left(1 - \frac{6}{8}F^2(E_r)\right)\frac{d\sigma_p}{dE_r}, \qquad (8.2)$$

where $F(E_r)$ is the nuclear form factor [196] for Carbon, while $\sigma_C^{coh}$ and $\sigma_p$ are the elastic scattering cross sections on Carbon and protons, which can be easily calculated. For

Carbon, $F(E_r)$ is sizable up to proton recoil energies of few MeV.

Since MiniBooNE would interpret $Z_\mathcal{D} \to e^+e^-$ decays as electron-like events, the reconstructed neutrino energy would be incorrectly inferred by the approximate CCQE formula (see e.g. Ref. [197])

$$E_\nu^{\rm rec} \simeq \frac{m_p E_{Z_\mathcal{D}}}{m_p - E_{Z_\mathcal{D}}(1 - \cos\theta_{Z_\mathcal{D}})}, \tag{8.3}$$

where $m_p$ is the proton mass, and $E_{Z_\mathcal{D}}$ and $\theta_{Z_\mathcal{D}}$ are the dark $Z_\mathcal{D}$ boson energy and its direction relative to the beam line. The fit to MiniBooNE data was then performed using the $\chi^2$ function from the collaboration official data release [169], which includes the $\nu_\mu$ and $\bar\nu_\mu$ disappearance data, re-weighting the Montecarlo events by the ratio of our cross section to the standard CCQE one, and taking into account the wrong sign contamination from Ref. [198]. Note that the official covariance matrix includes spectral data in electron-like and muon-like events for both neutrino and antineutrino modes.

In Fig. 82 we can see the electron-like event distributions, including all of the backgrounds, as reported by MiniBooNE. We clearly see the event excess reflected in all of them. The neutrino (antineutrino) mode data as a function of $E_\nu^{\rm rec}$ is displayed on the top (middle) panel. The light blue band reflects an approximated systematic uncertainty from the background estimated from Table I of Ref. [169]. On the bottom panel we show the $\cos\theta$ distribution of the electron-like candidates for the neutrino data, as well as the distribution for $\cos\theta_{Z_\mathcal{D}}$ for the benchmark point (blue line). The $\cos\theta$ distribution of the electron-like candidates in the antineutrino data is similar and not shown here and our model is able to describe it comparably well. We remark that our model prediction is in extremely good agreement with the experimental data. In particular, our fit to the data is better than the fit under the electronVolt sterile neutrino oscillation hypothesis [169] if one considers the constraints from other oscillation experiments. We find a best fit with $\chi^2_{bf}/\mathrm{dof} = 33.2/36$, while the background only hypothesis yields $\chi^2_{bg}/\mathrm{dof} = 63.8/38$, corresponding to a $5.2\sigma$ preference for our model.

In our framework, as the dark boson decays dominantly to charged fermions, the constraints on its mass and kinetic mixing are essentially those from a dark photon [199]. In the mass range $20 \sim 60$ MeV, the experiments that dominate the phenomenology are beam dump experiments and NA48/2. Regarding the dark neutrino, the constraints are similar but weaker than in the heavy sterile neutrino scenario with non-zero $|U_{\mu 4}|^2$ [108, 170]. Since $N_\mathcal{D} \to \nu e^+e^-$ is prompt, limits from fixed target experiments like PS191 [200], NuTeV [201], BEBC [202], FMMF [203] and CHARM II [204] do not apply. Besides, $W \to \ell N \to \ell\nu e^+e^-$ in high energy colliders can constrain $|U_{\mu 4}|^2 > \mathrm{few}\times10^{-5}$ for $m_{N_\mathcal{D}} > \mathcal{O}(\mathrm{GeV})$ [205]. Finally, we do not expect any significant constraints from the MiniBooNE beam dump run [206] due to low statistics.

In general, this model may in principle also give contributions to the muon $g - 2$, to atomic parity violation, polarized electron scattering, neutrinoless double $\beta$ decay, rare meson decays as well as to other low energy observables such as the running of the weak mixing angle $\sin^2\theta_W$. There might be consequences to neutrino experiments too. It can, for instance, modify neutrino scattering, such as coherent neutrino-nucleus scattering, or im-

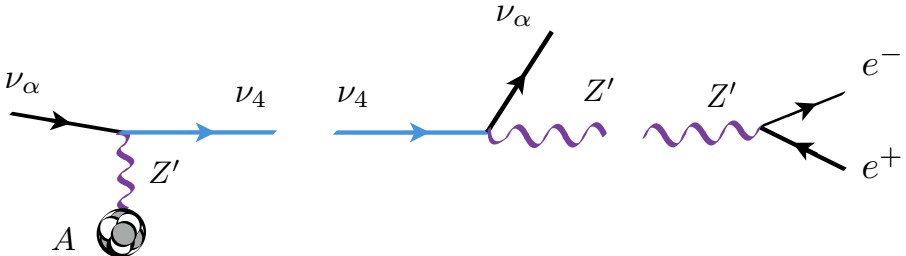

**Figure 91: Diagram of production of MiniBooNE signature.** A heavy neutrino, $\nu_4$, is produced via the exchange of a dark photon with the nucleous. For the realization in [195], $m_{Z'} < m_4$ and the decay chain follows as shown. In the scenario by [221], $m_{Z'} > m_4$ and the decay chain shown is forbidden; instead once $\nu_4$ is produced a three body decay into $\nu_\alpha e^+ e^-$.

pact neutrino oscillations experimental results as this model may give rise to non-standard neutrino interactions in matter. Furthermore, data from accelerator neutrino experiments, such as MINOS, NO$\nu$A, T2K, and MINER$\nu$A, may be used to probe $Z_\mathcal{D}$ decays to charged leptons, in particular, if the channel $\mu^+\mu^-$ is kinematically allowed. We anticipate new rare Higgs decays, such as $h_{\mathrm{SM}} \to ZZ_\mathcal{D}$, or $H_\mathcal{D}^\pm \to W^\pm Z_\mathcal{D}$, that depending on $m_{Z_\mathcal{D}}$ may affect LHC physics. Finally, it may be interesting to examine the apparent anomaly seen in $^8$Be decays [207] in the light of this new dark sector. The investigation of these effects is currently under way and shall be presented in a future work.

## 9 Testing the Low Energy MiniBooNE Anomaly at Neutrino Scattering Experiments (Argüelles)

One of the largest problems in contemporary neutrino physics is finding a satisfactory explanation to the LSND and MiniBooNE observation of electron-neutrino appearance at $L/E \sim 1$ km/GeV [169, 208, 209], where $L$ is the experiment baseline and $E$ the neutrino energy. Explanations of these *anomalies* in terms of a single vanilla eV-sterile neutrinos, often called "3+1" models, are significantly disfavored as they cannot explain both appearance and disappearance data sets [210–213]. Attempts have been made to reduce the tension between the sets by, *e.g.* adding new interactions to the sterile neutrino state [214–217] or allowing the new mostly-sterile mass state to decay [213, 218, 219]. Exploration of this non-vanilla eV-sterile neutrino model space is still on going, *e.g.* it has recently been pointed out in [213] that adding $\nu_4$ decay reduces the tension from a p-value of $3.7 \times 10^{-6}$ to $7.07 \times 10^{-4}$ [220].

The above solutions to the LSND-MiniBooNE anomaly work under the premise that these two pieces of evidence – and perhaps the recent hints in reactor neutrino experiments – are related and point to the same physics. This does not need to be the case; it is possible that we accidentally stumbled upon an unrelated anomaly in MiniBooNE, while chasing for confirmation of the original one in LSND. A set of novel explanations of MiniBooNE take

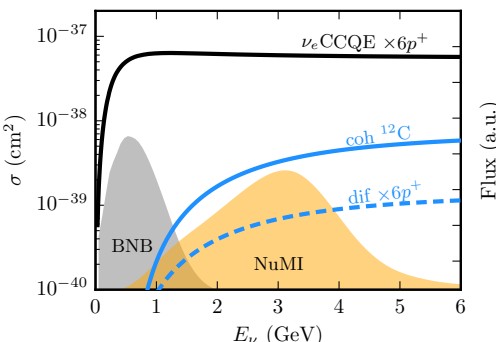

**Figure 92: Cross section of dark neutrinos at benchmark point.** The benchmark point is the same as in [195]. The blue lines correspond to the dark neutrino production cross section: solid blue for the coherent contribution and dashed blue for the diffractive contribution. For reference the $\nu_e$ charged-current quasielastic cross section is shown as a solid black line.

this stance and focus on only explaining the second anomaly. These explanations introduce simplified models that contain a dark neutrino, with a corresponding mass state $m_4$, and a dark photon of mass $m_{Z'}$ [195, 221]. Interestingly, these models can be embedded into complete theories that would not only be an explanation of the MiniBooNE anomaly, but could also explain neutrino masses [194, 222, 223]. To explain the MiniBooNE anomaly they notice that MiniBooNE, a Cherenkov detector, cannot distinguish between photons and electrons. More over, it can misclassify two electrons as a single one if they are close to collinear [195] or produced with very asymmetric momenta causing one of them to beunobserved [221]. These two electrons are produced in the decay chain of the dark neutrino, as shown in Fig. 91. In the case of $m_{Z'} < m_4$, a boosted $Z'$ is produced which decays into a pair of collinear electrons [195], in the case of $m_{Z'} > m_4$ the heavy neutrino undergoes a three-body decay producing an electron pair with asymmetric momenta [221].

Unlike the vanilla "3+1" models, which we know how to prove or disprove – namely by making dedicated experiments to measure the oscillation probability at $L/E \sim 1\,\text{km/GeV}$ [213, 224]–, for the dark neutrino models no such road map exists at the moment. The task ahead of us is then to construct robust strategies for confirming or definitely ruling out these scenarios. One way of confirming of these models is to look at neutrino scattering data [225]. These studies are more effective in the case of light $Z'$ as in that case the coherent contribution of the cross section dominates the production of the heavy neutrino. The cross section for parameter-point values that are able to explain the MiniBooNE excess for $m_{Z'} < m_4$ is shown in Fig. 92. As noted earlier, the cross section is dominated by the coherent contribution and flattens to be $\sim 10\%$ of the charged-current quasielastic cross section. Note that, for this parameter point, the cross section is just starting to turn on at the end of the BNB energy spectrum and is significantly larger for the NuMI energy range.

The main signature of the dark neutrino for this model is no hadronic activity and a pair of collinear electrons it makes sense to look for it in experiments that measure neutrino-electron scattering. The signature of the latter is similar, but with only one out-

going electron instead of two. The relatively heavy masses involved imply that we cannot make use of neutrino-electron cross section measurements performed with solar or reactor neutrinos. In our analysis we decide to use two complementary – due to the different beams, backgrounds, and detectors – experiments: Minerva low-energy mode and CHARM-II. It must be noted that though similar, the signatures of neutrino-electron scattering and dark neutrino production are mutually exclusive: one has a single electron coming out the other one has two. This implies that an analysis whose aim is to measure the former also very effectively removes the latter if the experiment can resolve one electron versus two [226, 227]. This is the case of both the Minerva and CHARM-II event selections [228, 229]. It is however necessary to modify – undo – some of their cuts to see the dark neutrino signal. Fig. 93 shows the distributions we use in our analysis and how these are related to the final analyses cuts. For Minerva we use the their event distribution without their final $dE/dx$ cut shown in Fig. 93 (left-top), while for CHARM-II we use their sample prior to the final $E\theta^2$ cut shown in Fig. 93 (left-bottom). In Fig. 93 (right) we show how these these cuts are related to the final event selection (panel A): our Minerva study uses panels A and B, while our CHARM-II analysis uses panels A and D.

Our analyses and results are conservative. Not only do we not use panels A, B, C, and D simultaneously, but we also simplify the data analysis by considering only rate information. This simplification makes our results more robust to distribution uncertainties within each panel. For both experiments we use a $\chi^2$ test-statistic with nuisance parameters to account for uncertainties in the flux and background normalization. Due to the difficulty in properly implementing the hadronic cuts, we consider only the coherent part of the cross section; this is again a conservative choice. We report our results in Fig. 94, where we show the new constraints as a function of the heavy neutrino mass and its mixing with the muon-flavor. The other parameters of the model are fixed to the same values as in [195] to facilitate comparison; we note that changing these parameters does not change this figure qualitatively since the constraints and the preferred region scale similarly. The Minerva constraint is weaker for high masses than the CHARM-II limit because of the lower energy of the NuMI beam. In the case of Minerva, the number of predicted events at the benchmark point overshoots the Minerva data. To demonstrate the robustness of the bound, we show – as the dashed blue line – the limit when computed with an unrealistic assumption of 100% uncertainty on the background component . The backgrounds are more relevant for the CHARM-II bounds. In this case we use the sideband region of $E\theta^2 > 30\text{MeV}$ rad to constrain the background size in the $E\theta^2 < 30\text{MeV}$ rad region by fitting a change in normalization and slope in that energy range. This procedure yields an uncertainty of 3% on the size of background in the analysis region. We note that the background in this region is dominated by $\nu A \to \nu \pi^0 A$ and $\nu N \to \nu \pi^0 N$, with a sub-dominante component, at the $\sim 10\%$ level, of $\nu_e N \to eN$. The larger two background components angular dependence is better understood, but the latter can have significant uncertainties. For this reason we also consider the scenario of 30% uncertainty in the background for the CHARM-II bounds which is comparable to null knowledge of the latter component. This limit is plotted as the dashed cherry-red line.

We have performed two analyses, one with Minerva and the other with CHARM-II,

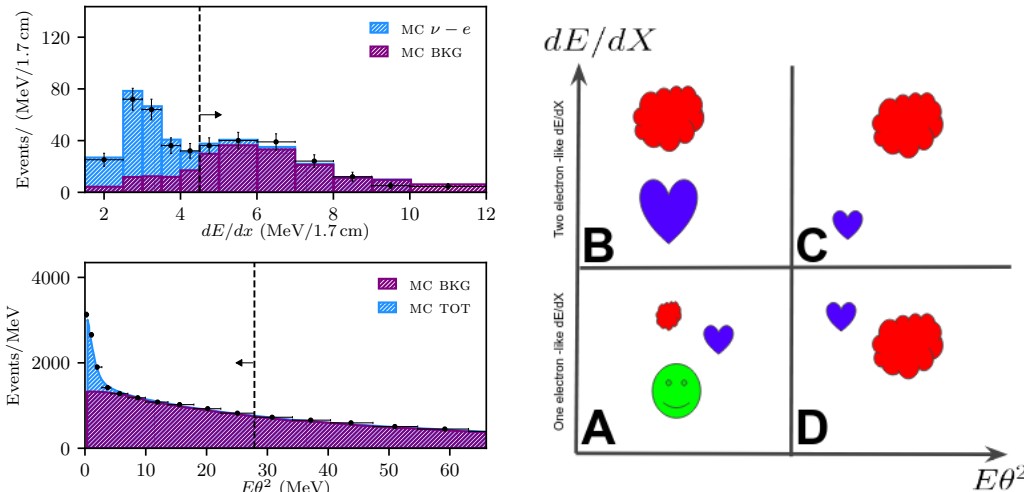

**Figure 93: Data plots used in the analysis and a cartoon that explains how they are related.** Left top: Distribution of Minerva low-energy mode neutrino-electron cross section event selection without the $dE/dx$ cut. We use the rate of events to the right of the cut, as signalled by the arrow, to place perform our analysis. Left bottom: Distribution of CHARM-II neutrino-electron cross section analysis without the $E\theta^2$ cut. The rate to the left of the cut, as pointed by the arrow, is used in our analysis. Right: The content of the different parts of phase space in the two variables used in our analysis, $dE/dx$ and $E\theta^2$. The Minerva distribution used in our analysis corresponds to panels A and B, while the CHARM-ii information used corresponds to panels A and D. The markers signal electron-neutrino (green happy face), di-electrons produced in the dark neutrino model (purple heart), and backgrounds such as neutral-current $\pi_0$ (red clouds). The sizes of the markers are meant to illustrate their relative contributions, though exact sizes are experiment and dark neutrino parameter-point dependent; the location of the makers within each of the panels (A,B,C, and D) does not have meaning.

seeking confirmation of the dark neutrino as an explanation of the MiniBooNE excess. We do not find evidence for dark neutrino induced events, but find the solutions where the dark photon is lighter than the heavy neutrino disfavored. Unfortunately, due to limitations in the data accessible it is not possible at the moment to make a quantitative assessment of the tension between neutrino scattering experiments and these solutions. It is encouraging that very soon more neutrino-electron scattering data sets will be made available for Minerva medium-energy and NovA. These new data sets open the possibility of searching for dark neutrino models, as explanations of the MiniBooNE observation, explanations of neutrino masses, or both.

## 10  Leptogenesis from Low Energy CP Violation (Turner)

Leptogenesis via the decays of heavy Majorana neutrinos is a plausible explanation for the predominance of matter over anti-matter of the Universe. This simple mechanism augments

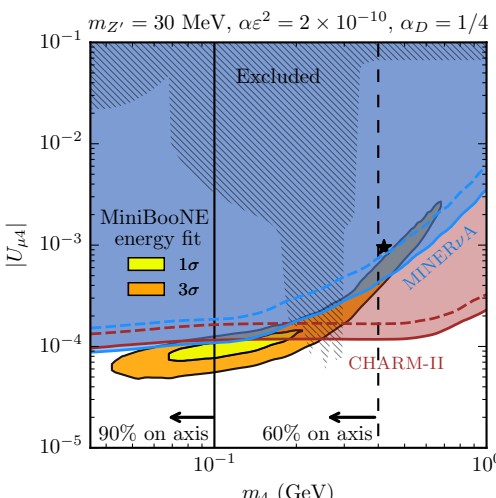

**Figure 94: Constraints in the dark neutrino parameter space.** New constraints are shown as light shaded regions: blue for Minerva and cherry-red for CHARM-II. The preferred region to explain the MiniBooNE excess energy distribution is shown in yellow for the one and three sigma regions. The dashed vertical line signals the value of $m_4$ for which the angular distribution of the MiniBooNE excess is also a good fit. The benchmark point of [195] is shown as a black star. Other relevant assumed parameters are written above the plot; changing these parameters does not change the relationship between preferred signal regions and constraints as they both scale approximately the same.

the Standard Model (SM) with a number of heavy Majorana neutrinos which consequently generate light neutrino masses via the type I seesaw mechanism [44, 171, 172, 230]. In addition, the decay of these SM singlets can occur out of thermal equilibrium and be CP asymmetric thereby producing a lepton asymmetry which is the partially converted to a baryon asymmetry [231] via non-perturbative SM processes. $CP$ violation is fundamental to the creation of the matter-antimatter asymmetry and in thermal leptogenesis this results from both low-scale measurable phases and high-scale immeasurable ones. We summarize the results of [232] which revisits the question: can leptogenesis via decays produce the observed baryon asymmetry of the Universe if the only source of $CP$ violation is the low energy observable phases of the PMNS matrix? We apply the notation and conventions of the aforementioned work and refer the interested reader to that paper for further details and a more in-depth analysis.

In order to establish the connection between the low and high scale phases with the dynamically generated lepton asymmetry, we write the Yukawa coupling of the heavy Majorana neutrinos to leptonic and Higgs doublets in terms of the Casas-Ibarra (CI) parametrization [233]:

$$Y = \frac{1}{v} U \sqrt{\hat{m}_\nu} R^T \sqrt{f(M)^{-1}}, \tag{10.1}$$

where $Y$ is the Yukawa matrix, $v$ is the vacuum expectation value of the Higgs, $\hat{m}_\nu$ is the positive diagonal matrix of light neutrino masses, $R$ is a complex orthogonal matrix given

by

$$R = \begin{pmatrix} 1 & 0 & 0 \\ 0 & c_1 & s_1 \\ 0 & -s_1 & c_1 \end{pmatrix} \begin{pmatrix} c_2 & 0 & s_2 \\ 0 & 1 & 0 \\ -s_2 & 0 & c_2 \end{pmatrix} \begin{pmatrix} c_3 & s_3 & 0 \\ -s_3 & c_3 & 0 \\ 0 & 0 & 1 \end{pmatrix}, \tag{10.2}$$

where $c_i = \cos w_i$, $s_i = \sin w_i$ and the complex angles are given by $w_i = x_i + iy_i$ ($i \in \{1, 2, 3\}$). $f(M)$ has the following form

$$\begin{aligned} f(M) &= M^{-1} - \frac{M}{32\pi^2 v^2} \left( \frac{\log\left(\frac{M^2}{m_H^2}\right)}{\frac{M^2}{m_H^2} - 1} + 3\frac{\log\left(\frac{M^2}{m_Z^2}\right)}{\frac{M^2}{m_Z^2} - 1} \right) \\ &= \mathrm{diag}\left( \frac{1}{M_1}, \frac{1}{M_2}, \frac{1}{M_3} \right) \\ &\quad - \frac{1}{32\pi^2 v^2} \mathrm{diag}\left( g\left(M_1\right), g\left(M_2\right), g\left(M_3\right) \right), \end{aligned}$$

which includes the one-loop corrections to the light neutrino masses [234]. In the limit such corrections are negligible, $f(M)$ is simply the diagonal heavy Majorana mass matrix. In the case of interest, we focus on the addition of three heavy Majorana neutrinos and therefore the parameter space is 18-dimensional: nine parameters of which are, in principle, measurable and the remaining nine of which remain immeasurable even if the leptogenesis occurs at the PeV scale.

In the scenario we presently discuss, all the $CP$ violation stems from the measurable phases. We therefore *assume* that the high scale phases are $CP$ conserving and therefore the $R$ matrix entries must be purely imaginary or real. There have been a number of works which motivate this assumption such as minimal flavor violation [235, 236], flavor symmetries [237–239] or a generalized $CP$ symmetry [240–242].

The basis of this work is then to solve the density matrix equations, which track the time evolution of the lepton in time (or inverse temperature), for a given point in the CI parameter space. We solve the density matrix equations of [243] whose details are further elucidated upon in [232]. In the simplest formulation, these kinetic equations are in the one-flavored regime, in which only a single flavor of charged lepton is accounted for. This regime is only realized at sufficiently high temperatures ($T \gg 10^{12}$ GeV) when the rates of processes mediated by the charged lepton Yukawa couplings are out of thermal equilibrium and therefore there is a single charged lepton flavor state which is a coherent superposition of the three flavor eigenstates. However, if leptogenesis occurs at lower temperatures ($10^9 \ll T \ll 10^{12}$ GeV), scattering induced by the tau Yukawa couplings can cause the single charged lepton flavor to decohere and the dynamics of leptogenesis must be described in terms of two flavor eigenstates. There exists the possibility that thermal leptogenesis occurs at even lower temperatures, $T < 10^9$ GeV, during which the interactions mediated by the muon have equilibrated. In such a regime, the Boltzmann equations should be given in terms of all three lepton flavors. In this summary we focus on the three-flavored scenario in which the scale of the leptogenesis era occurs at $T < 10^9$ GeV.

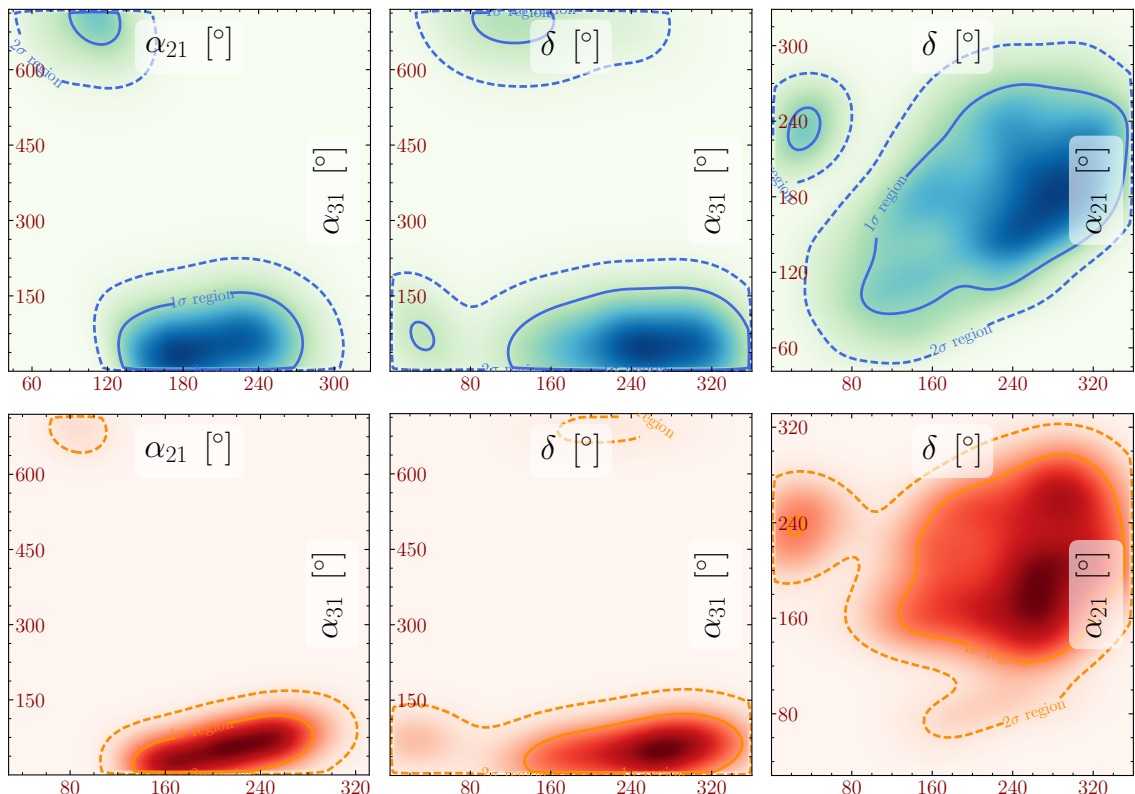

**Figure 101:** The two-dimensional projections for intermediate scale leptogenesis with $M_1 = 3.16 \times 10^6$ GeV for $x_1 = 0$, $y_2 = 0$, $x_3 = 180°$, $y_1 = y_2 = 180°$, with $CP$ violation provided only by the phases of the PMNS matrix. The normal ordered case is colored blue/green and inverted ordering orange/red and contours correspond to 68% and 95% confidence levels. We fix $m_1 = 0.21$ eV, the mixing angles of the PMNS matrix and mass squared splittings of the light neutrinos are fixed at their best fit values as given by global fit data [245]. This plot was created using SUPERPLOT [246].

## 10.1 Leptogenesis in the regime $T < 10^9$ GeV

Successful thermal leptogenesis at intermediate scales (temperatures $\sim 10^6$ GeV) may be accomplished through the combination of flavor effects and exploration of regions of the parameter space in which there is a certain degree of (less than three decimal place) cancellation between the tree and one-loop light neutrino masses [244]. We present and analyze the results of a comprehensive search of the model parameter space for regions with successful leptogenesis where the sole source of $CP$ violation derives from the Majorana and Dirac phases. To do so we search for regions of the model parameter space, $\mathbf{p}$, that yield values of $\eta_B(\mathbf{p})$ that are consistent with the measurement $\eta_{B_{\text{CMB}}} = (6.10 \pm 0.04) \times 10^{-10}$. We apply an efficient sampling method for three reasons. First, the parameter space has a relatively high dimension. Second, the function $\eta_B(\mathbf{p})$ itself does not vary smoothly with changes of $\mathbf{p}$. In fact, tiny variations of the input parameters yield function values differing in many orders of magnitude and sign. Third, the computation of $\eta_B(\mathbf{p})$ for a single point

is relatively expensive and can take up to the order of seconds. Thus any attempt of a brute-force parameter scan is doomed to fail. Finally, we are not only interested in a single best-fit point but also a region of confidence that resembles the measurement uncertainty.

We found the use of MULTINEST [247, 248] to be particularly well suited to address all the aforementioned complications associated to this task. The MULTINEST algorithm has seen wide and very successful application in astronomy and cosmology. It provides a nested sampling algorithm that calculates Bayesian posterior distributions which we will utilize in order to define regions of confidence.

In all our scenarios, MULTINEST uses a flat prior and the following log-likelihood as objective function

$$\log L = -\frac{1}{2} \left( \frac{\eta_B(\vec{p}) - \eta_{BCMB}}{\Delta \eta_{BCMB}} \right)^2. \tag{10.3}$$

Once a MULTINEST run is finished, we use SUPERPLOT [246] to visualize the posterior projected onto a two-dimensional plane.

The lowest scale (i.e. lowest value of $M_1$) for which thermal leptogenesis was found to be possible, using only $CP$ violating phases from the PMNS matrix, was found to be $M_1 = 3.16 \times 10^6$ GeV. For normal ordering the regions of parameter space consistent with the observed baryon asymmetry at the one (two) $\sigma$ are shown in dark (light) blue. To produce the observed baryon asymmetry within the one $\sigma$ range, the viable values of Majorana phase are $130 \leq \alpha_{21}(°) \leq 260$ while $\alpha_{31}$ can take smaller values with $0 \leq \alpha_{31}(°) \leq 150$. Likewise the one $\sigma$ favored values of the Dirac delta phase is $120 \leq \delta(°) \leq 360$. Unlike the Majorana phases, the Dirac phase comes with a suppression of $\sin \theta_{13}$ and therefore larger values are required in order to provide sufficient $CP$ violation. Qualitatively, the viable parameter space for successful leptogenesis in the inverted ordering spectrum is similar to that of normal ordering.

## 10.2 Summary

We revisit the possibility of producing the observed baryon asymmetry of the Universe via thermal leptogenesis, where $CP$ violation comes exclusively from the low-energy phases of the neutrino mixing matrix. We demonstrate the viability of producing the baryon asymmetry of the Universe from nonresonant thermal leptogenesis at lower scales than previously thought possible, ($M_1 \sim 10^6$ GeV).

## 11 More New Physics with Long-Baseline Experiments (de Gouvêa)

Our understanding of neutrino properties changed dramatically over the last twenty years. Almost all neutrino data are consistent with the three-massive-neutrinos paradigm: there are three neutrino species which interact via the Standard Model (SM) weak interactions and at least three of the neutrino masses are not zero. The charged-current weak-interactions are such that leptons mix, similar to what happens to quarks. These conditions lead to neutrino oscillations and precision measurements of neutrino oscillations allow one to measure many of the new fundamental parameters in the "enhanced' SM Lagrangian: neutrino mass-squared differences and the elements of the leptonic mixing matrix.

If the three-massive-neutrinos paradigm is correct, we have come a long way when it comes to measuring all the neutrino oscillation parameters. Most mixing parameters are measured quite precisely (better than 10% precision), some not as well (worse than 10% precision), and one parameter is virtually unknown: the CP-odd phase in the leptonic mixing matrix that can be probed via neutrino oscillations, often referred to as the Dirac phase [115]. The search for CP-violating phenomena in the lepton sector is among the defining reasons for pursuing bigger and better long-baseline neutrino experiments, including Hyper-Kamiokande in Japan [249] and the DUNE experiment in the United States [250]. Both projects are moving ahead and expect to begin data taking in the second half of the next decade.

As important as the pursuit of leptonic CP-violation is the job of testing the three-massive-neutrinos paradigm. While most neutrino data are consistent with it, it is fair to say that it has undergone very few non-trivial "stress-tests." In particular, large effects beyond the three-massive-neutrinos paradigm could be lurking just beyond the reach of the current neutrino oscillation data. The next-generation of long-baseline experiment will provide qualitatively better opportunities to test the three-massive-neutrinos paradigm.

In order to test the limitations of the three-massive-neutrinos paradigm, it is important to identify candidate-models for the new physics beyond the three-massive-neutrinos paradigm. These candidate-models serve many purposes, including gauging the reach of different experimental set-ups, comparing neutrino oscillation experiments with other particle physics probes of new phenomena, and identifying targets for next-generation endeavors. There are many such candidate models including:

- New neutrino states. In this case, the $3 \times 3$ mixing matrix would not be unitary.

- New short-range neutrino interactions. These lead to, for example, new matter effects. If we don't take these into account, there is no reason for the three flavor paradigm to "close."

- New, unexpected neutrino properties. Do they have nonzero magnetic moments? Do they decay? The answer is 'yes' to both, but nature might deviate dramatically from the expectations of the three-massive-neutrinos paradigm.

- More exotic phenomena, including CPT-violation, violations of unitary evolution of quantum mechanical systems (decoherence), etc.

Here I very briefly summarize the results of a few case studies: [251–253]. These were mostly interested in new physics impacting the $\nu_\mu$-disapparance channel and the $\nu_e$-appearance channel in the DUNE far detector. For results including $\nu_\tau$-appearance, see [254]. Simulations involving Hyper-Kamiokande were also considered, mostly in the context of elucidating the nature of the new phenomenon that manifested itself in long-baseline oscillations. The cases studies revolved around different concrete scenarios: the fourth neutrino hypothesis [251], non-standard neutrino interactions [252], and the possibility that neutrino and antineutrino oscillation parameters are different [253, 255]. The studies addressed three broad questions:

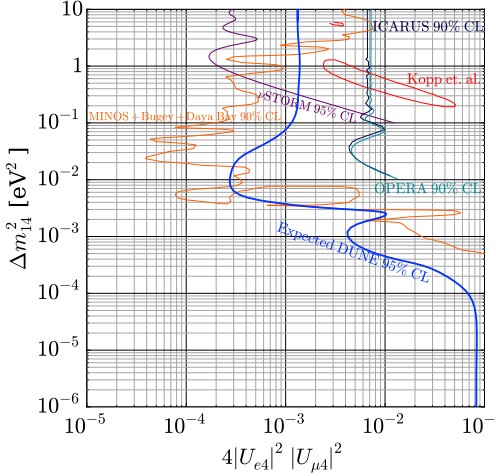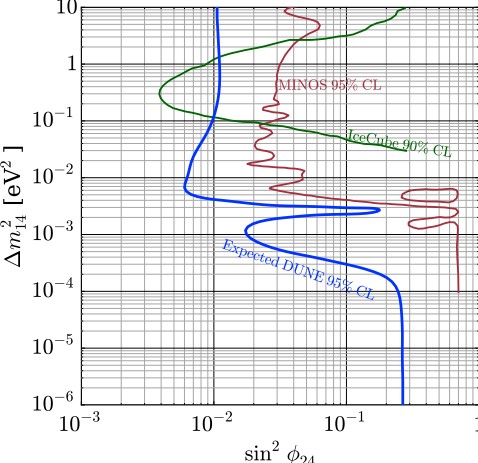

**Figure 111:** Sensitivity of the DUNE experiment to the fourth neutrino hypothesis, updated from [251]. See [251] for details. $\Delta m^2_{14} \equiv m^2_4 - m^2_1$ while, in the right-side panel, $\sin^2 \theta_{24} \equiv |U_{\mu 4}|^2$.

- How sensitive are next-generation long-baseline efforts?

- How well they can measure the new-physics parameters, including new sources of CP-invariance violation?

- Can they tell different new-physics models apart?

Very detailed results are presented in [251–253] and document the sensitivity of next-generation experiments to new phenomena. One example is depicted in Figure 111, in the context of the fourth-neutrino hypothesis. In a nutshell, significant increase in sensitivity is expected from next-generation experiments. It is also exciting to learn that DUNE and Hyper-Kamiokande have the potential to measure the new-physics parameters and even identify new sources of CP-invariance violation.

A more subtle question is related to whether, once the presence of a new phenomena is identified, next-generation long-baseline experiments can positively identify the nature of the new phenomenon. This, it turns out, is a more subtle question. For example, it is possible to fit data consistent with non-standard neutrino interactions with the fourth neutrino hypothesis. Distinguishing different new phenomena will, most likely, rely on the combination of information from different experimental probes, including comparing results from Hyper-Kamiokande and DUNE.

In summary, we are still deciphering the physics uncovered by oscillation experiments over twenty years ago. While the hypothesis that neutrinos have mass and leptons mix fits almost all neutrino oscillation data, there remains the possibility of running into more unexpected phenomena. It is important to identify clear ways of looking for new phenomena,

quantifying how well all data are consistent the three-massive-neutrinos paradigm, and identifying new-physics ideas one can constrain uniquely, or best, with neutrino oscillation experiments. Next-generation experiments will see "first light" in about a decade and there is still a lot of preparatory work to do in order to fully exploit the new unprecedented high-quality, high-statistics data sets.

## 12 Neutrinophilic Dark Matter at the DUNE Near Detector (Kelly)

Much of the work throughout this report assumes new interactions among neutrinos exist without any specific requirement on the form of the interaction or the mass of the particle that mediates it. Here, we explore scenarios in which the new particle is light enough to be emitted in certain processes – decays of heavier particles (mesons, higgs boson) or neutrino scattering. We also consider the possible that this new mediator is a portal to a thermal dark sector.

We assume that a new scalar $\phi$ with mass $m_\phi$ interacts with the SM neutrinos via the Lagrangian

$$\mathcal{L} \supset \frac{(L_\alpha H)(L_\beta H)}{\Lambda_{\alpha\beta}^2}\phi + \text{h.c.} \longrightarrow \frac{1}{2}\sum_{\alpha,\beta=e,\mu,\tau}\lambda_{\alpha\beta}\nu_\alpha\nu_\beta\phi + \text{h.c.}, \tag{12.1}$$

where $L_\alpha$ is the $SU(2)$ lepton doublet with flavor $\alpha$, $H$ is the higgs doublet, and $\Lambda_{\alpha\beta}$ is the effective scale of this dimension-six interaction. After EWSB, represented by the arrow in Eq. (12.1), we expect couplings of $\phi$ to the light neutrinos proportional to the dimensionless coupling $\lambda_{\alpha\beta} \equiv v^2/\Lambda_{\alpha\beta}^2$, where $v = 246$ GeV is the higgs vacuum expectation value. See Refs. [256, 257] for more detail.

Since we are interested in the DUNE Near Detector, we will focus on the $\lambda_{\mu\mu}$ coupling. Constraints on $\lambda_{\mu\mu}$ and $m_\phi$ come predominantly from invisible decays of the higgs boson ($h \to \nu\nu\phi$) [35, 256] and charged kaon decays ($K^+ \to \mu^+\nu_\mu\phi$) [258–261]. The measured invisible branching ratio of the higgs boson constrains $\lambda_{\mu\mu} \lesssim 0.7$ for $m_\phi \ll m_h$. Charged kaon decays constrain $\lambda_{\mu\mu} \lesssim 2 \times 10^{-2}$ for $m_\phi \ll m_{K^\pm} - m_\mu$, however, off-shell decays $K^+ \to \mu\nu_\mu\nu\overline{\nu}$ can constrain this parameter for $m_\phi > m_{K^\pm} - m_\mu$. These constraints are shown as grey regions in Fig. 122 in the top and top-left regions, respectively.

Of interest here is the process in which an incoming neutrino may radiate an on-shell $\phi$, proceeding to scatter with a nucleus and produce a charged lepton. Because the $\phi$ is radiated as initial-state radiation, it will carry away energy and transverse momentum – the resulting neutrino scattering event will appear to have a large missing transverse momentum and lower neutrino energy than it actually has. Fig. 121 displays this type of "mono-neutrino" scattering event.

Ref. [257] explored in detail the capability of the DUNE Near Detector to search for this signal among a background of standard charged-current $\nu_\mu$ events. This search may be aided by the fact that such signal events have a wrong-sign charged-lepton in the final state. While the liquid argon component of the DUNE Near Detector is not magnetized, the proposed gaseous argon component is, and some fraction of the final-state muons in the event sample will reach the gas and have their charged identified [23]. Additionally,

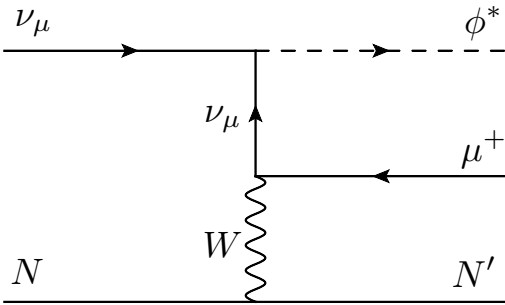

**Figure 121:** Example of the neutrino emission of $\phi$ before scattering off a nucleus, a "mono-neutrino" event that can be searched for in the DUNE near detector.

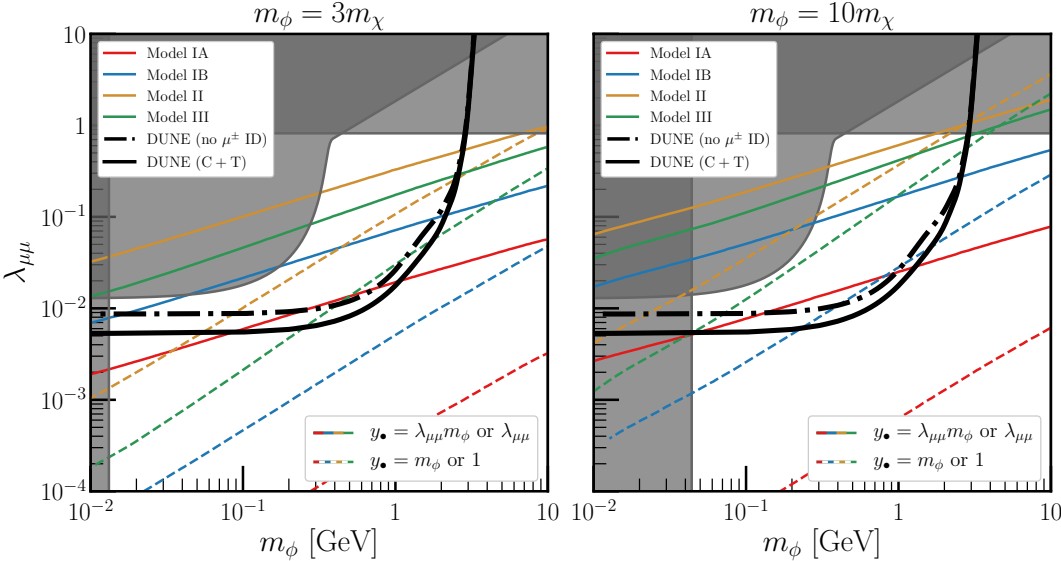

**Figure 122:** Expected DUNE sensitivity to a scalar $\phi$ with mass $m_\phi$ and coupling to $\nu_\mu$ $\lambda_{\mu\mu}$ assuming 10 years of data collection at the near detector. Solid black line: sensitivity using charge-identification techniques, Dot-dashed black line: no charge identification. Colored lines indicate thermal relic targets for the dark matter scenarios discussed in the text.

muons stopping in the liquid and producing Michel electrons may be used to identify charge on a statistical basis. In Fig. 122 we display the sensitivity of DUNE in this parameter space assuming 10 years of data collection at the near detector, both using these charge-identifying techniques (solid black lines) and without (dot-dashed black lines). The two panels are identical for the sake of DUNE – the differences have to do with the assumptions regarding dark matter in the following. We find that, for all $m_\phi$ below about 2 GeV, DUNE will be able to explore parameter space previously unconstrained by higgs/Kaon decays.

If $\phi$ is a mediator between the standard model and a dark sector, there are several possible types of dark matter (DM) that couple to $\phi$ in interesting ways. We ex-

plore four here. Model IA: scalar DM $\chi$, $\mathcal{L} \supset 1/2y_{IA}\chi^2\phi$. Model IB: fermionic DM $\chi$, $\mathcal{L} \supset 1/2y_{IB}\bar{\chi}^c\chi\phi$. Model II: scalar triple-coupled DM $\mathcal{L} \supset 1/6y_{II}\chi^3\phi$. Model III inelastic fermion DM $\mathcal{L} \supset y_{III}\bar{\chi}_1\chi_2\phi$. In all four cases, we may calculate the expected relic abundance assuming an initial thermal population, with DM freeze-out into neutrinos. We do this calculation making two assumptions about how $\phi$ couples to DM and neutrinos; (a) $\phi$ couples democratically, $y_X = \lambda_{\mu\mu}$ (times $m_\phi$ for the dimensionful coupling $y_{IA}$) and (b) $\phi$ couples preferentially to DM, $y_X = 1$ ($\times m_\phi$). Regions of parameter space for which the observed relic abundance of DM is satisfied are shown in Fig. 122 for $m_\phi = 3m_\chi$ (left) and $m_\phi = 10m_\chi$ (right). CMB observations disfavor light DM, and a conservative lower bound on the DM mass is shown in grey on the left side of each panel [262–264].

We see that the DUNE Near Detector has the capability of reaching thermal relic targets across a large range of mediator masses, $10^{-2}$ GeV $\lesssim 2$ GeV. With precise measurements of the final-state particles in Near Detector interactions, DUNE may perform interesting measurements of new neutrino interactions in a previously unseen way.

## 13  Light Mediators (Dutta)

The scale of new physics can be anywhere. The question is whether it can be motivated by any physics consideration. For example, an understanding of the origin of electroweak scale requires new physics around a TeV which is being searched thoroughly at the Large Hadron collider(LHC). However, the scales associated with the origin of tiny neutrino mass, dark matter(DM) masses can be anywhere. For example, a tiny neutrino mass can arise as a combination of the Dirac and Majorana neutrino scales and the location of these scales can be anywhere between 1 eV and $10^{16}$ GeV. The scale of dark matter can also be anywhere between 1 KeV to a multiple TeV unless it is a weakly interacting massive particle.

In this talk, I will discuss models with mediator masses between MeV and 10 GeV. This range has been found to be very interesting for new neutrino interactions and DM models. Further, the low mass mediator models with mediator masses $\lesssim$ GeV do not have much constraints from the LHC and these models may be associated with sub-GeV DM [265]. In this talk, I will utilize the the neutrinos and DM to search for these light mediator models in the ongoing searches and a few novel possibilities. For the neutrino based light mediator model, I would use the recent COHERENT timing + energy data to show constraints on models. For DM searches, first, I will talk about a model for light DM scenario motivated by a solution to the fermion mass hierarchy problem and then mention how the cosmic ray upscatter can investigate the light mediator models even if the DM mass is in the sub GeV regime. I would then use the COHERENT experiment in a novel way to search for DM.

### 13.1  Utilizing neutrinos to probe light mediators

The COHERENT collaboration has reported the first detection of coherent neutrino-nucleus elastic scattering (CE$\nu$NS) [7]. The COHERENT data provides an important new channel to search for beyond the Standard Model (BSM) physics. For example, the data constrains non-standard neutrino interactions (NSI) [12, 266] due to heavy or light mediators [17–19, 267–271], generalized scalar and vector neutrino interactions [272], and

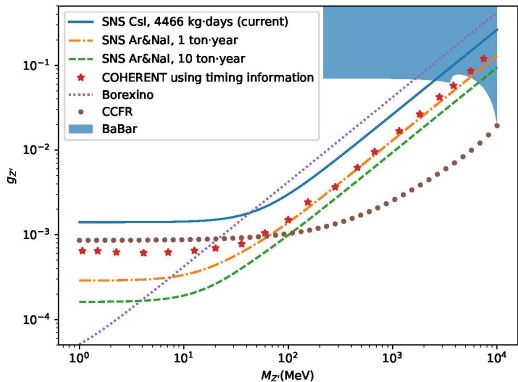

**Figure 131:** $g_{Z'}$ as a function of $M_{Z'}$ for $L_\mu$-$L_\tau$ models. The red stars show the best fit of the COHERENT data(energy+ timing).

| field | $q_R^u$ | $q_R^d$ | $\ell_R$ | $\nu_R$ | $\eta_L$ | $\eta_R$ | $\phi$ |
|-------|---------|---------|----------|---------|----------|----------|--------|
| $q_{T3R}$ | -2 | 2 | 2 | -2 | 1 | -1 | -2 |

**Table 131:** The charges of fields which transform under $U(1)_{T3R}$. The charges are given for the left-handed component of each Weyl spinor. The anomalies cancel by construction.

hidden sector models [273]. It also sets independent constraints on the effective neutron size distribution of CsI [274–276], and on sterile neutrinos [277, 278].

We have performed a fit to the energy and timing distribution of nuclear recoil events from the COHERENT data which provides information on the flavor content of the neutrino flux beyond what is obtained with the energy data alone. We have shown that including the information in both the energy and timing distributions of the COHERENT data, there is a $\sim 2\sigma$ deviation between the best-fitting model and the SM prediction. Light mediators in the mass range $\sim 10 - 1000$ MeV are able to provide a good fit to the data [279]. In Fig.131, we show the best fit points for $L_\mu - L_\tau$ model [280, 281] which are below the CCFR constraint line.

## 13.2 Utilizing the direct detection of DM to probe light mediators

Since it is important to develop a model of sub-GeV DM with light mediators, We first describe one complete model which contains a sub-GeV DM and two light mediators: scalar and gauge boson.

The low energy gauge symmetry of this new is $SU(3)_C \times SU(2)_L \times U(1)_Y \times U(1)_{T3R}$ [282]. We will assume that the new gauge group $U(1)_{T3R}$ is not connected to electric charge, defined as $Q=T_{3L}+Y$. This model can be constructed with the first two generation of right handed fermions of the SM to ameliorate the Yukawa coupling hierarchies associated with the. This mode can satisfy the thermal relic abundance as shown in table 132.

One of the most promising experimental avenues is to search for the small energy depositions from DM elastically scattering in very sensitive detectors on Earth. This "direct detection" of DM can proceed from scattering on either nuclear electrons. In either case

| | $m'_A$ | $m'_\phi$ | $m_\eta$ | $m_{\nu_s}$ | $m_{\nu D}$ | $\langle \sigma v \rangle$ (cm$^3$/sec) | $\sigma_{\rm SI}^{scalar}$(pb) | $\sigma_{\rm SI}^{vector}$(pb) |
|---|---|---|---|---|---|---|---|---|
| muon case | 55 | 200 | 100 | 10 | $10^{-3}$ | $3\times10^{-26}$ | 33.00 | 6.50 |
| | 70 | $10^4$ | 50 | $10^{16}$ | $10^4$ | $3\times10^{-26}$ | $2.30\times10^{-8}$ | 1.80 |
| electron case | 0.4 | 200 | 100 | 0.1 | $10^{-4}$ | $3\times10^{-26}$ | 33.00 | 6.50 |

**Table 132:** Masses of $A'$, $\phi'$ and $\eta$ (DM) in MeV and the corresponding thermal relic abundances are shown for two different model scenarios, i.e., muon and electron cases. The dark matter-nucleon scattering cross sections for each benchmark point are also shown. For the case of $A'$-mediated inelastic scattering, $\delta$ is taken small.

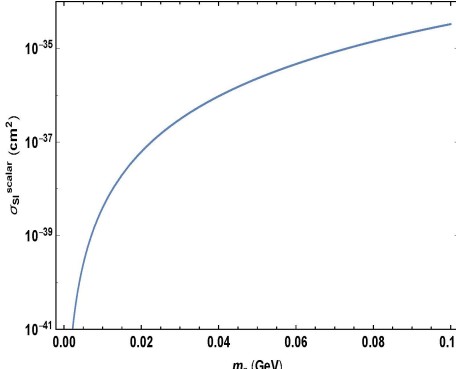 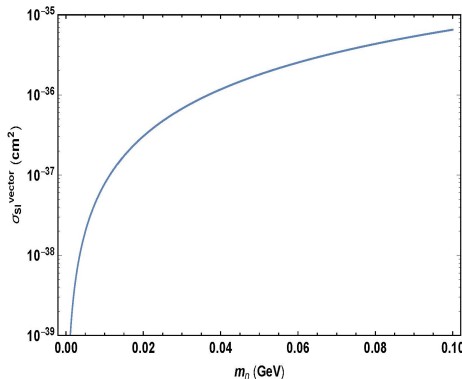

**Figure 132:** Dark matter-nucleon scattering cross section as a function of the dark matter mass. The cross sections are calculated for $m_{\phi'} = 100$ MeV, $\delta = 0$ and $V = 10$ GeV. These dark matter-nucleon cross sections are allowed by CRESST III, XENON1T and CDEX-1B constraints.

however, the same interactions with ordinary matter allows high energy cosmic rays (CRs) to scatter on background DM. This can improve detection prospects for light DM by giving such particles much larger energies so that that are more easily detected in terrestrial nuclear [283, 284], electron scattering [285]. In Fig. 132, we show the the allowed parameter space of this model after including bounds from CRESST III, XENON1T and CDEX-1B constraints after including the effects of cosmic ray up-scattering.

### 13.3 Search of DM at the COHERENT experiment

We can search for Low-mass DM at the COHERENT experiment in a novel analysis. The signal can be initiated by production of dark photon, say $A'$, via a process, $\pi^- + p \to n + A'$, followed by the decay of $A'$ to a pair of dark matter, say $\chi$ (there can be an additional contribution from $\pi^- + p \to n + \pi^0$ [286], where $\pi^0$ decays to $\gamma + A'$). Apart from the $\pi^-$ absorption, there can be a direct production of $\pi^0$s which are, however, relativistic and since the detectors at COHERENT are at $\sim 90°$ from the beam direction, the DM arising from the relativistic $\pi^0$ decay are mostly in the forward direction and they would miss the detectors. The DM, from the decays of $A'$ from $\pi^-$ absorption flies toward the detector and scatters elastically off a target nucleus. Since both timing and energy distribution of

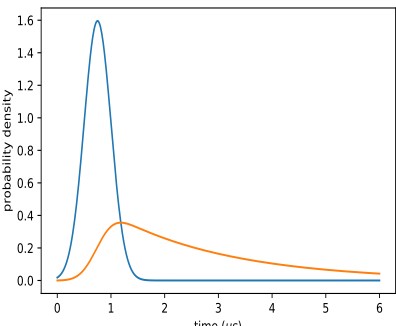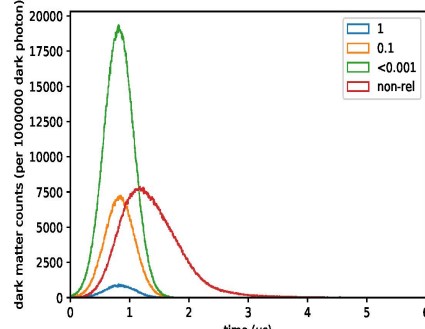

**Figure 133:** Timing distribution for the prompt and delayed components arising from the neutrinos(left) and DM arising from the dark photon (relativistic with $m_{\mathrm{darkphoton}} \leq 138$ MeV and non relativistic cases) for various dark photon lifetime scenarios ($1, 2, \leq 10^{-3} \, \mu$s).

the DM events will be different compared to the neutrino scattering arising from the $\pi^+$ decay, the timing and energy spectra due to DM scattering would be able to distinguish the light DM models. In fig.133, we show the timing spectra of neutrinos and compare that with neutrinos. Applying energy cut and timing cut, we will show, in an upcoming publication, that the DM signal can be distinguished from the neutrino scattering signal at the COHERENT (or similar type) experiment [287].

## 14 Does NSI Hide at Low Mediator Masses? (Shoemaker)

By possessing nonzero masses, neutrinos are messengers of physics Beyond the Standard Model. New physics associated with neutrinos can be parameterized in an Effective Field Theory valid at energies below some cutoff, $\lesssim \Lambda$, by an expansion of higher dimensional operators of SM particles

$$\mathcal{L} = \mathcal{L}_{\mathrm{SM}} + a \, \frac{(LH)^2}{\Lambda} + b \, \frac{\mathcal{O}_\nu \, \mathcal{O}_f}{\Lambda^2} + \cdots \tag{14.1}$$

where $\mathcal{O}_i$ is a fermion bilinear of particles $i$. The sole dimension-5 operator has the well-known feature of being able to account for neutrino masses once the electroweak symmetry is broken and the Higgs boson acquires a vacuum expectation value. Here we focus on the class of dimension-6 interactions known as neutral-current "Non-Standard Neutrino Interactions" or NSI, typically written as

$$\mathcal{L}_{\mathrm{NSI}} = -2\sqrt{2}G_F \, \varepsilon_{\alpha\beta}^f (\nu_\alpha \gamma_\mu \nu_\beta)(f \gamma^\mu P \, f), \tag{14.2}$$

where $f$ is a SM fermion, the indices $\alpha, \beta = e, \mu, \tau$ span neutrino flavor, and $P$ is the left/right projector. In the above formulation the new physics scale $\Lambda$ is traded for the the dimensionless coefficient $\varepsilon_{\alpha\beta}^f$ such that the strength of NSI is measured in units of the Fermi constant, $G_F$.

## 14.1 Models of NSI and their phenomenolgy

We focus on the neutral current type of NSI which via the matter effect can impact neutrino propagation. These can be realized in simple $Z'$ type completions coupling neutrinos to quarks and/or electrons [41, 43, 288]. Other classes of completions leading to Eq. 14.2 (upon Fierzing) include scalar mediators such as "leptoquarks" [289] and electrophilic doublets [39].

These models can be probed in a number of ways including scattering, oscillation, and collider data [41, 138, 140, 142]. The size of neutral current NSI is important since it mitigates our ability to precisely determine the standard oscillation parameters. The most striking example of this occurs in the so-called "LMA-dark" solution, which in oscillation data is completely degenerate with the SM LMA solution:

$$\text{LMA}: \ \theta_{12} \simeq 34^\circ \oplus \varepsilon \ \simeq 0 \ \iff \ \text{LMA} - \text{D}: 45^\circ < \theta_{12} < 90^\circ \ \oplus \varepsilon \ \simeq 1. \tag{14.3}$$

This makes it impossible to determine the octant of $\theta_{12}$ using oscillation data alone [16]. However, scattering and collider data can break the degeneracy and reveal the correct oscillation parameters.

## 14.2 COHERENT Partially Breaks LMA-D Degeneracy

COHERENT data can be used to probe NSI as well. In fact, in the contact interaction limit it is strong enough to rule out LMA-D [18]. However, this conclusion does not persist in the light mediator regime. This can be easily understood by examining the neutrino-nucleus scattering amplitude in the cases for which the $Z'$ mass is above or below the characteristic momentum transfer

$$\delta \mathcal{M} \propto \begin{cases} \frac{g_\nu g_q}{M_{Z'}^2} & \text{if } M_{Z'} \gg q, \\ \frac{g_\nu g_q}{q^2} & \text{if } M_{Z'} \ll q. \end{cases} \tag{14.4}$$

Thus it is clear that for mediators lighter than about $\lesssim 10$ MeV, the scattering data from COHERENT only provides a bound on the coupling $g$ and not the mediator mass. However since the NSI matter effect scales as $(\varepsilon \, G_F) \sim g^2/m_{Z'}^2$, light enough $Z'$ masses will be sufficient to generate large NSI. We see that relaxing the contact operator assumption opens up allowed parameter space for large NSI.

The bounds on a class of $Z'$ models is summarized in the left panel Fig. 141. Together with CMB and COHERENT bounds only a narrow range of mediator masses can still account for NSI large enough for LMA-Dark. In the future, CONUS may be able to probe the remaining region.

## 14.3 Can NSI be sourced by Dark Matter?

Though COHERENT data nearly excludes standard NSI at the size needed for the LMA-Dark solution, there are other possible contributions to the matter potential of neutrinos. In particular, Refs. [290, 291] explored dark matter models wherein the local DM background can source a large matter potential. These are models in which sterile neutrinos and fermionic dark matter interact through new scalar, vector, or tensor interactions. The

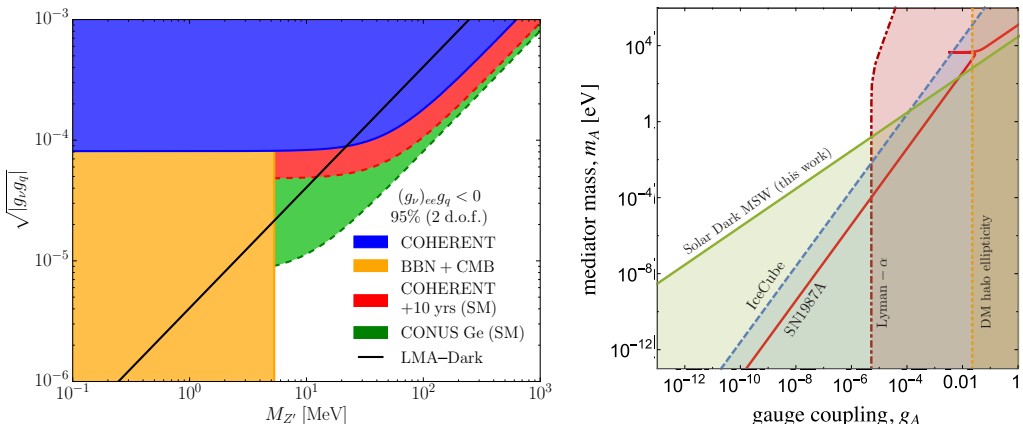

**Figure 141:** Light mediator NSI (*left panel*) is still large enough to account for LMA-Dark, but only in a narrow range of masses [20]. Large matter effects (*right panel*) can come about in models of dark matter-neutrino interactions [290].

resulting bounds on a class of vector mediator models are summarized in the right panel of Fig. 141. These include IceCube and SN1987A constraints on neutrino-neutrino interactions [292], Lyman-alpha bounds on the first dark matter proto-halos, and constraints on dark matter self-interactions from halo ellipticity. As can be seen a large parameter space at low masses and couplings can evade all other existing constraints, and is only probed via neutrino oscillations. Future multi messenger transient sources can also provide strong probes of such models [293].

As we have seen, the experimental constraints on neutral current NSI continue to strengthen from a number of directions. Given the strength of current experimental sensiitivites however, it is important to carefully exmaine the underlying assumptions underpinning these bounds and to chart new theoretical directions. As an example of this we explored implications of the LMA-Dark solution of neutrino oscillations, demonstrating a narrow range of available parameter space in $Z'$ completions of NSI. A distinct class of models in which the $Z'$ also interacts with background dark matter particles opens up new possibilities for distinctive matter potential effects, with novel complementary phenomenological probes [290, 291].

## 15 Complete Model of NSI: Flavor Physics and Light Mediators (Mocioiu)

Flavor physics is one of the least understood issues in the Standard Model. For quarks, the third generation stands apart as much heavier and less mixed with the others. From the gauge theory point of view, each generation is self-consistent within the Standard Model, with all potential anomalies canceling within each generation. With these observations in mind, it is natural to consider the possibility that the third generation could be charged under a new gauge group, $U(1)_{B-L}^{(3)}$. This preserves the anomaly cancelation within each generation, provided one adds a right-handed sterile neutrino to each generation as well.

Ref. [41] proposed this model and explored its very wide range of consequences, which also include the generation of neutrino NSI.

The new gauge interaction implies that the third family cannot mix with the other two using the standard model Higgs. This leads to an extended Higgs sector that includes a Higgs field with the standard model quantum numbers, $\phi_2$, and a new field, $\phi_1$, charged under the new gauge symmetry, that allows the third family to mix with the first two. To make the theory phenomenologically viable it is also necessary to introduce one more scalar field, $s$, that is a singlet under the standard model group, but is charged under the new $U(1)_{B-L}^{(3)}$ gauge group. The vacuum expectation values of $\phi_1$ and $s$ spontaneously break the new gauge symmetry around the weak scale, generating a relatively light mass $M_X$ for the new gauge boson $X$. A mixing of $X$ and $Z$ is also generated, leading to couplings of the new gauge bosons to all families, with appropriate suppression factors for the first two generations. The full consistency of flavor observables has to be reanalyzed in the available parameter space, as the standard effective field theory description of flavor physics constraints no longer applies when the new gauge boson is light. The new neutral currents also lead to non-universal neutrino interactions in matter, thus manifesting as an effective $\varepsilon_{\tau\tau}$ NSI. While two Higgs doublet models, with or without additional singlets, have been extensively studied in the literature, our realization has a number of unique features due to the unusual flavor structure. The extended Higgs sector of this model can be studied at the LHC, as well as with precision electroweak data and flavor observables.

The phenomenology of the model is extremely rich, with different constraints relevant in different regions of parameter space. In addition to the neutrino NSI, the $Z - X$ mixing leads to potential observable effects in Atomic Parity Violation experiments and invisible Higgs decays. Other processes that provide strong constraints on the parameter space include $\Upsilon$, $K$, $B$ and $D$ decays, $D - \bar{D}$ oscillations and electroweak precision observables. A total of 20 processes affecting different types of physics experiments have been considered. The most important constraints and the surviving parameter space are presented in Fig. 151 for the choice of $\tan\beta = v_2/v_1 = 10$.

It is important to note that different observables are primarily sensitive to different model parameters, allowing a very detailed exploration of all sectors of the model (gauge, scalar vevs, Yukawas).

For a range of parameters with $g_X \sim 10^{-3} - 10^{-2}$ and the $X$ gauge boson mass of order 300 MeV-1 GeV, neutrino oscillations become the main probe even at present. The induced neutrino non-standard interaction is flavor conserving, namely $\varepsilon_{\tau\tau}$ in the present model, and its values are in the interesting range for DUNE, Hyper-Kamiokande, PINGU, and other present and future experiments with increased sensitivity. While the gauge symmetry is necessary to induce the new matter effect, the observables only depend on the size of the scalar VEVs and not on the gauge coupling. The neutrino NSI thus probe the scale of the symmetry breaking, which can be even above a few TeV and still lead to potentially observable effects.

Another class of relevant observables that is sensitive to the structure of the extended Higgs sector is based on precision meson decay data. This includes $K^+ \to \pi^+ X$, $B^+ \to K^+ X$, $t \to cX$, $\Upsilon \to X\gamma$, $D^+ \to \pi^+ X$ decays, among others. For decay processes, at high

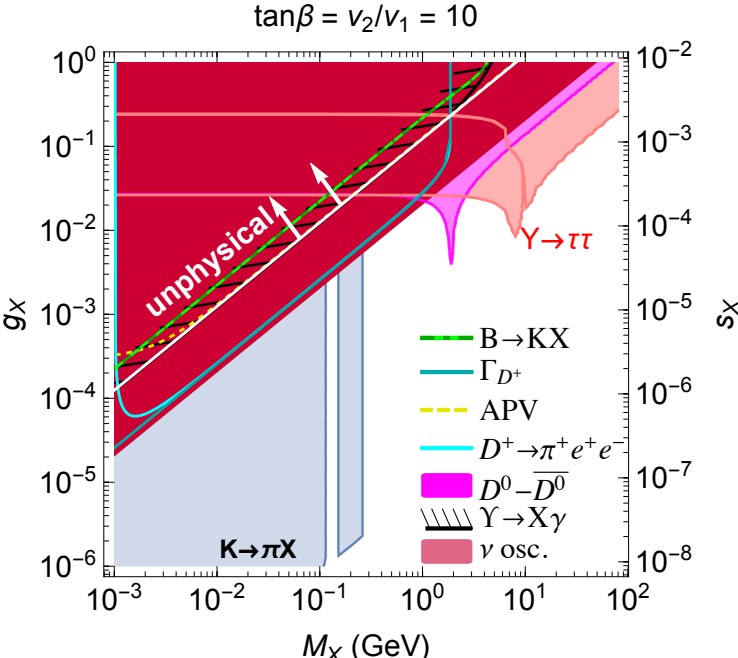

**Figure 151:** Constraints on the $U(1)_{B-L}^{(3)}$ gauge boson mass $M_X$ and coupling $g_X$ for $\tan\beta = 10$. For convenience the $X - Z$ mixing, $s_X$, is also shown. Notice that for a given $g_X$, the mass of the gauge boson $M_X$ is bounded from below, so there is an unphysical region in the upper left corner of the $M_X \times g_X$ plane (delineated by the white line). The "$\nu$ osc." bound comes from non standard interaction effects (matter potential) on atmospheric neutrinos. "APV" refers to atomic parity violation. Here the charged Higgs mass, relevant to the $B \to KX$ constraint, is taken to be 1200 GeV.

energies, the equivalence theorem implies that the longitudinal mode can be replaced by the Goldstone boson associated with the breaking of the symmetry. Thus, many of the constraints would survive even in the limit of a global symmetry. When the decay channel $K^+ \to \pi^+ X$ is kinematically viable, kaon physics poses by far the most important bound on the model. $B^+ \to K^+ X$ is also quite stringent, but depends strongly on $M_{H^+}$ and $\tan\beta$. Another class of observables involves processes such as the $D - \bar{D}$ mixing, atomic parity violation, and Møller scattering: for a heavy mediator they probe a combination of VEVs in the Higgs sector, related to the $X - Z$ mixing, while for a low mediator mass they are also sensitive to the gauge coupling $g_X$. A unique role is played by the process $\Upsilon \to \tau^+ \tau^-$, which operates entirely in the third generation and gives the most direct access to the coupling $g_X$.

This model has an important ambiguity: which leptons should carry the new gauge quantum numbers. While for quarks the choice is clear – the top and bottom have very small mixing with the quarks from the other generations, which we are trying to explain – in the case of leptons there is no natural choice. We have so far considered the tau lepton and the corresponding neutrino, but only for definitiveness. From the point of view of anomaly cancelation, which was our guiding principle in selecting the flavor symmetry, any lepton flavor could have been selected to be charged under this new symmetry. This means that it is possible to generate any flavor diagonal neutrino non-standard interaction ($\varepsilon_{ee}$ or $\varepsilon_{\mu\mu}$) in a similar way. Some of the other constraints and future search strategies

are different in this case and would require a reanalysis.

# 16 Effective Field Theory for Non-Standard neutrino Interactions in Elastic Neutrino - Nucleus Scattering (Tammaro)

Neutrino NSI have been studied extensively in neutrino oscillations. However, the common NSI effective Lagrangian relevant for neutrino oscillations contains only dimension 6 operators. Namely, only NSI parametrized by $\varepsilon_{\alpha\beta}^{fV}$ and $\varepsilon_{\alpha\beta}^{fA}$.

A qualitatively new set of NSI probes is opening up through the coherent elastic neutrino-nucleus scattering measurements (CE$\nu$NS [3]), achieved for the first time by the COHERENT collaboration [7]. This result now makes it possible to probe a wide variety of NSI at low momenta exchanges.

## 16.1 Operator basis

In experiments where momenta exchanges are $q \lesssim \mathcal{O}(100\text{MeV})$, well below the electroweak scale, the interactions of neutrinos with matter are described by an effective Lagrangian, obtained by integrating out the heavier degrees of freedom.

The interaction Lagrangian for $\nu_\alpha \to \nu_\beta$ transition is thus given by a sum of non-renormalizable operators,

$$\mathcal{L}_{\nu_\alpha \to \nu_\beta} = \sum_{a,d=5,6,7} \hat{\mathcal{C}}_a^{(d)} \mathcal{Q}_a^{(d)} + \text{h.c.} + \cdots, \qquad \text{where} \quad \hat{\mathcal{C}}_a^{(d)} = \frac{\mathcal{C}_a^{(d)}}{\Lambda^{d-4}}. \qquad (16.1)$$

Here the $\mathcal{C}_a^{(d)}$ are dimensionless Wilson coefficients, while $\Lambda$ can be identified, for $\mathcal{O}(1)$ couplings, with the mass of the new physics mediators. We consider a complete basis of EFT operators up to and including dimension seven.

Here we write down the full basis of EFT operators assuming neutrinos are Dirac fermions, while in [294] we also comment on what changes are needed if neutrinos are Majorana. We use four-component notation, following the conventions of Ref. [295]. There is one dimension-five operator

$$\mathcal{Q}_1^{(5)} = \frac{e}{8\pi^2}(\bar{\nu}_\beta \sigma^{\mu\nu} P_L \nu_\alpha) F_{\mu\nu}, \qquad (16.2)$$

where $F_{\mu\nu}$ is the electromagnetic field strength tensor. The dimension-six operators are

$$\mathcal{Q}_{1,f}^{(6)} = (\bar{\nu}_\beta \gamma_\mu P_L \nu_\alpha)(\bar{f}\gamma^\mu f), \qquad \mathcal{Q}_{2,f}^{(6)} = (\bar{\nu}_\beta \gamma_\mu P_L \nu_\alpha)(\bar{f}\gamma^\mu \gamma_5 f). \qquad (16.3)$$

We disregard the operators with $P_L \to P_R$, as these operators cannot be well tested in neutrino experiments, since the production of right-handed neutrinos through SM weak interactions is neutrino mass suppressed. The dimension seven operators are

$$\mathcal{Q}_1^{(7)} = \frac{\alpha}{12\pi}(\bar{\nu}_\beta P_L \nu_\alpha) F^{\mu\nu} F_{\mu\nu}, \qquad \mathcal{Q}_2^{(7)} = \frac{\alpha}{8\pi}(\bar{\nu}_\beta P_L \nu_\alpha) F^{\mu\nu} \widetilde{F}_{\mu\nu}, \qquad (16.4)$$

---

[3]While not all of the NSI scatterings will be coherently enhanced we keep the, by now standard, CE$\nu$NS terminology.

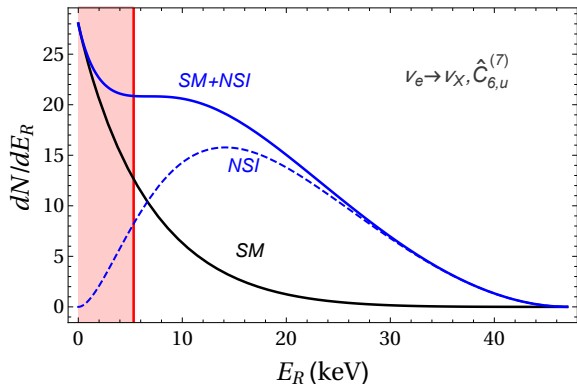

**Figure 161:** Recoil spectrum at COHERENT adding the new scalar NSI $\mathcal{Q}_{6,u}^{(7)}$, assuming $\Lambda \sim 2$ GeV. The red line is the detector energy threshold.

$$\mathcal{Q}_3^{(7)} = \frac{\alpha_s}{12\pi}(\bar{\nu}_\beta P_L \nu_\alpha)G^{a\mu\nu}G_{\mu\nu}^a, \qquad \mathcal{Q}_4^{(7)} = \frac{\alpha_s}{8\pi}(\bar{\nu}_\beta P_L \nu_\alpha)G^{a\mu\nu}\widetilde{G}_{\mu\nu}^a, \qquad (16.5)$$

$$\mathcal{Q}_{5,f}^{(7)} = m_f(\bar{\nu}_\beta P_L \nu_\alpha)(\bar{f}f), \qquad \mathcal{Q}_{6,f}^{(7)} = m_f(\bar{\nu}_\beta P_L \nu_\alpha)(\bar{f}i\gamma_5 f), \qquad (16.6)$$

$$\mathcal{Q}_{7,f}^{(7)} = m_f(\bar{\nu}_\beta \sigma^{\mu\nu} P_L \nu_\alpha)(\bar{f}\sigma_{\mu\nu}f), \qquad \mathcal{Q}_{8,f}^{(7)} = (\bar{\nu}_\beta i\overset{\leftrightarrow}{\partial}_\mu P_L \nu_\alpha)(\bar{f}\gamma^\mu f), \qquad (16.7)$$

$$\mathcal{Q}_{9,f}^{(7)} = (\bar{\nu}_\beta i\overset{\leftrightarrow}{\partial}_\mu P_L \nu_\alpha)(\bar{f}\gamma^\mu \gamma_5 f), \qquad \mathcal{Q}_{10,f}^{(7)} = \partial_\mu(\bar{\nu}_\beta \sigma^{\mu\nu} P_L \nu_\alpha)(\bar{f}\gamma_\nu f), \qquad (16.8)$$

$$\mathcal{Q}_{11,f}^{(7)} = \partial_\mu(\bar{\nu}_\beta \sigma^{\mu\nu} P_L \nu_\alpha)(\bar{f}\gamma_\nu \gamma_5 f). \qquad (16.9)$$

Here $G_{\mu\nu}^a$ is the QCD field strength tensor, $\widetilde{G}_{\mu\nu} = \frac{1}{2}\varepsilon_{\mu\nu\rho\sigma}G^{\rho\sigma}$ its dual (and similarly for QED, $\widetilde{F}_{\mu\nu} = \frac{1}{2}\varepsilon_{\mu\nu\rho\sigma}F^{\rho\sigma}$), and $a = 1,\ldots,8$ the adjoint color indices. The fermion label, $f = u, d, s, e, \mu$, denotes the light quarks, electrons or muons, while $(\bar{\nu}i\overset{\leftrightarrow}{\partial}_\mu\nu) = (\bar{\nu}i\partial_\mu\nu) - (\bar{\nu}i\overset{\leftarrow}{\partial}_\mu\nu)$. We assume flavor conservation for charged fermions, while we do allow changes of neutrino flavor. Note that in general, except dimension 6 operators with $\alpha = \beta$, the above operators are not Hermitian, and thus can have complex Wilson coefficients.

## 16.2 Neutrino - nucleus scattering

The neutrons and protons inside nuclei are non-relativistic and their interactions are well described by a chiral EFT with nonrelativistic nucleons. The momentum exchange, $q$, in CE$\nu$NS scattering is of $\mathcal{O}(10)$ MeV, small enough that nuclei remain intact, while neutrons and protons are non-relativistic throughout the scattering event. At leading chiral order the neutrino interacts only with a single nucleon, while interactions of a neutrino with two nucleons are suppressed by powers of $q/\Lambda_{\mathrm{ChEFT}}$.

The matching of the operators in Sec. 16.1 into a nuclear operator basis proceeds in three steps:

1. hadronization of quarks into nucleons, parametrized by nucleon form factors $F_i^{q/N}(q^2)$ [296];

2. Non-Relativistic reduction of nucleon currents, performed using Heavy Baryon Chiral Perturbation Theory [297, 298];

3. embedding of nucleons into nuclei, described by nuclear response functions $W_i^{\tau\tau'}$ [299, 300].

After these three steps we can write the CE$\nu$NS matrix element squared

$$\overline{\mathcal{M}^2} = \frac{1}{2J_A+1}\sum_{\text{spins}}|\mathcal{M}|^2 = \frac{4\pi}{2J_A+1}\sum_{\tau,\tau'=0,1}\left(R_M^{\tau\tau'}W_M^{\tau\tau'} + R_{\Sigma''}^{\tau\tau'}W_{\Sigma''}^{\tau\tau'} + R_{\Sigma'}^{\tau\tau'}W_{\Sigma'}^{\tau\tau'}\right).$$

(16.10)

The $W_M^{\tau\tau'}$ encodes the spin-independent nuclear response, induced by scalar and vector operators, thus scales as $W_M^{\tau\tau'} \sim \mathcal{O}(A^2)$, while the other two encode the spin-dependent response, induced by pseudoscalar, axial and tensor operators; as such their size is $W_{\Sigma'}^{\tau\tau'} \sim W_{\Sigma''}^{\tau\tau'} \sim \mathcal{O}(1)$.

The $R_i^{\tau\tau'}$ are the respective kinematic factors, which in general depends on the specific NSI and $q^2$. Some operators can induce particular kinematic dependence, which can sizably affect the recoil energy spectrum. In Fig. 161 is shown the effect of a new scalar NSI on the $dN/dE_R$ behavior at COHERENT. A change in the shape of the spectrum will then immediately signal the presence of NSI in CE$\nu$NS .

## 16.3   Limits on New Physics scale

Assuming a single NSI operators[4] is present with $\mathcal{O}(1)$ coupling, we can use the results from previous section to get a lower limit on the NP scale associated with the NSI. As an example, the result for $\nu_e$ dimension 5 and 7 operators is shown in Fig. 162. We include also results from Borexino, CHARM, and projections from one of the upgrades proposed by COHERENT [301]. While $\mathcal{Q}_1^{(5)}$ is strongly bounded at COHERENT and even more at Borexino, dimension 7 operators allow for relatively light NP mediators, of $\mathcal{O}(1-10)$ GeV for scalar, pseudoscalar and tensor operators, and $\mathcal{O}(100)$ GeV for derivative operators.

As final remark, the theoretical predictions of CE$\nu$NS rates in presence of NSI is affected by additional uncertainties from the nuclear response functions and the form factors, which needs to be taken into account. A dedicated study on these uncertainties and more precise Lattice evaluations of form factors would then be highly desired.

## 17   New Physics in Coherent Neutrino Scattering (Xu)

Coherent elastic neutrino-nucleus scattering (CE$\nu$NS), which has been recently observed by the COHERENT collaboration [7], provides a new approach to searching for new physics beyond the standard model. There have been extensive studies on various new physics related to neutrinos, including NSI, SPAVT interactions[5], light mediators, sterile neutrinos, neutrino magnetic moments, and dark matter, etc.

---

[4]This assumption means that all possible interferences at nuclear level are disregarded, which could be wrong if a new mediator generates more than one operator in the basis.

[5]Generalized four-fermion interactions with Scalar, Pseudo-Scalar, Vector, Axial, and Tensor couplings.

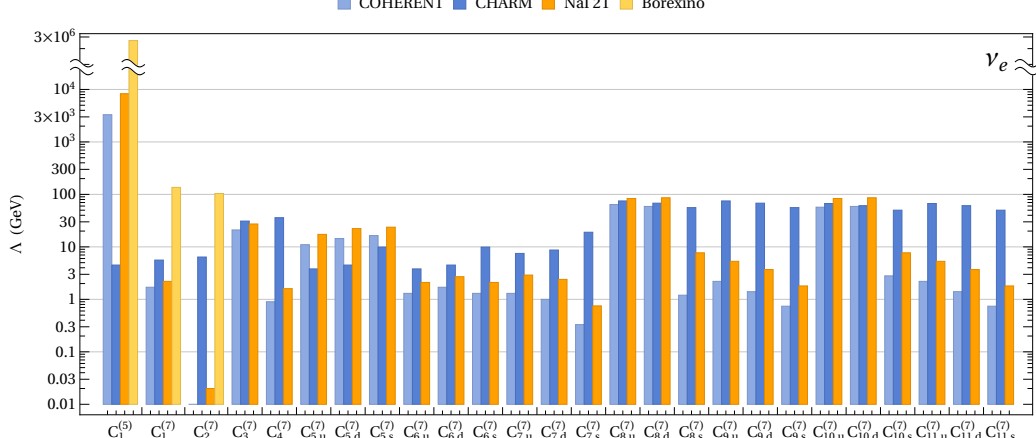

**Figure 162:** Limits from COHERENT, CHARM, Borexino, and projected limits from a NaI 2T experiment on the scale $\Lambda$ of dimension 5 and dimension 7 NSI operators for electron neutrinos.

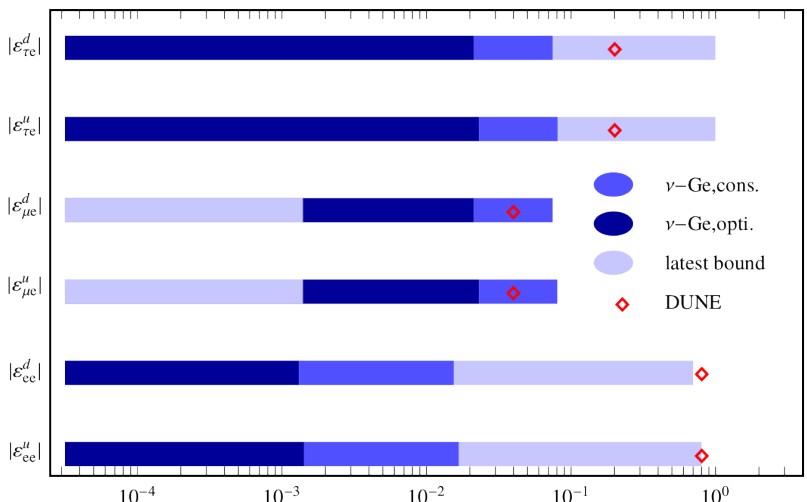

**Figure 171:** Future coherent bounds on NSIs, from Ref. [269].

The SM cross section of coherent elastic neutrino-nucleus scattering, assuming full coherency, is given by

$$\frac{d\sigma}{dT} = \frac{G_F^2 Q_{SM}^2 M}{4\pi}\left(1 - \frac{T}{T_{\max}}\right),\tag{17.1}$$

$$Q_{SM}^2 = \left[N - (1 - 4s_W^2)Z\right]^2,\; T_{\max} = \frac{2E_\nu^2}{M + 2E_\nu},\tag{17.2}$$

where $M$ is the mass of the nucleus, $T$ is the recoil energy of the nucleus, $T_{\max}$ is the maximal recoil energy that can be generated for a certain value of neutrino energy, $N$ and $Z$ are the neutron and proton numbers of the nucleus.

Since in coherent scattering, the cross section is proportional to the square of the total amplitude of neutrino-nucleon scattering amplitudes, it is significantly enhanced by the number of nucleons (mostly $N$). However, if $N$ is large, then the nucleus is heavy, and the recoil energy, which is the only observable effect in any elastic neutrino scattering processes, has to be very small. So even though the cross section can be very large, detection of such a process is technically challenging. Due to recent significant development of ultra-low threshold detection technology, which is also demanded by dark matter direct detection experiments, CE$\nu$NS has been observed successfully and will soon become an effective way to detect neutrinos with robust statistics.

- NSI

Generally speaking, if neutrino interactions are mediated by a new gauge boson, then integrating it out gives rise to NSIs. In addition, loop corrections could also give rise to NSIs, which can be potentially large if some dark sectors contribute to the loop corrections [302]. Note that NSIs remain the well-known V$-$A form, which implies that there are no light right-handed neutrinos involved in such interactions. Because of the V$-$A form, the effect of NSI in CE$\nu$NS is simple. We simply need to replace the SM weak charge of the nucleus in Eq. (17.1) with the NSI charge (i.e., $Q_{\rm SM}^2 \to Q_{\rm NSI}^2$):

$$
Q_{\rm NSI}^2 \equiv 4 \left[ N(-\frac{1}{2} + \varepsilon_{ee}^{uV} + 2\varepsilon_{ee}^{dV}) + Z(\frac{1}{2} - 2s_W^2 + 2\varepsilon_{ee}^{uV} + \varepsilon_{ee}^{dV}) \right]^2
$$
$$
+ 4 \sum_{\alpha=\mu,\tau} \left[ N(\varepsilon_{\alpha e}^{uV} + 2\varepsilon_{\alpha e}^{dV}) + Z(2\varepsilon_{\alpha e}^{uV} + \varepsilon_{\alpha e}^{dV}) \right]^2 . \tag{17.3}
$$

For NSI, what CE$\nu$NS can measure is actually $Q_{\rm NSI}^2$, which leads to a lot of degeneracy among those NSI parameters. From the studies in Refs. [18, 19, 268, 303], one can summarize the current COHERENT constraints on NSIs, which are that generally around $\mathcal{O}(0.5)$. In the future, CE$\nu$NS experiments based on reactor sources can provide very strong constraints on NSIs, due to the potentially high statistics. As shown in Fig. 171, with a low-threshold 100 kg Germanium detector being set near a 1 GW reactor, the NSI bounds on $\varepsilon_{ee}$ and $\varepsilon_{e\tau}$ can be improved by one or two orders of magnitudes.

- SPVAT

SPVAT interactions refer to four-fermion contact interactions with the the most general Lorentz invariant forms. More explicitly, the interactions can be formulated as follows [269]

$$
\mathcal{L} \supset \frac{G_F}{\sqrt{2}} \sum_{a=S,P,V,A,T} \overline{\nu}\Gamma^a \nu \left[ \overline{\psi}\Gamma^a(C_a + D_a i\gamma^5)\psi \right], \tag{17.4}
$$

where $\psi$ denotes electrons, quarks or nucleons, and $\Gamma^a$ can be one of the five possible Dirac matrices:

$$
\Gamma^a = \{1,\, i\gamma^5,\, \gamma^\mu,\, \gamma^\mu\gamma^5,\, \sigma^{\mu\nu} \equiv \frac{i}{2}[\gamma^\mu, \gamma^\nu]\}. \tag{17.5}
$$

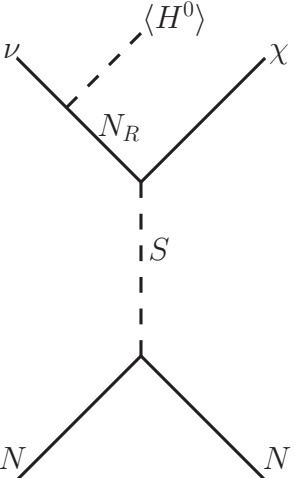

**Figure 172:** A process to produce dark matter in CEVNS proposed in Ref. [271].

Such a formalism serves as a comprehensive EFT description of new physics involving heavy particles which can be integrated out.

In coherent neutrino scattering, SPVAT interactions have more interesting phenomenology (e.g., distortions of the recoil spectrum) than NSIs. Some interactions here involves light right handed neutrinos, which in principle, allows to distinguish between Dirac and Majorana neutrinos [304]. The current COHERENT constraints on SPVAT interactions are roughly at the same order of magnitude as the COHERENT constraints on NSIs [272, 294, 303].

- Light mediator

If new neutrino interactions are mediated by light particles (e.g., a light scalar $\phi$, or a light gauge boson $Z'$), then they should not be integrated out in neutrino scattering. Their masses have crucial effect when the momentum transfer is close to the mediator masses. This can be seen from the cross section [270]:

$$\frac{d\sigma}{dT} = \frac{y^4 M}{4\pi(2MT + m^2)^2}[\cdots],\tag{17.6}$$

where $[\cdots]$ denotes an $\mathcal{O}(1)$ quantity that depends on the type of the mediator (scalar or vector) and the kinematics; while $(2MT + m^2)^2$ shows the dependence on the mediator mass $m$. This causes additional distortion of the recoil spectrum, which if observed, could be used to reconstrcut the mass and coupling of the light mediator. The current COHERENT bounds have not yet exceeded $\nu + e$ scattering bounds [305, 306], but future reactor-based CE$\nu$NS experiments will produce strongest constraints on light mediators.

- Neutrino magnetic moments

Neutrinos magnetic moments (NMM) can be generated with pure SM interactions at the 1-loop level, provided that they have nonzero masses. The SM values [307] are of the order of

$\mathcal{O}(10^{-20})\mu_B$, suppressed by neutrino masses. However, some BSM interactions could make potentially large contributions to NMMs. Especially if the new interactions has a chirality-flipping feature, then NMMs could be enhanced by about 10 orders of magnitudes [308], reaching the current/future observable ranges. Therefore, observations of large NMMs could real the underlying new physics of neutrinos.

Coherent neutrino scattering, as a low energy elastic neutrino scattering process, is sensitive to large NMMs. The recent COHERENT data has put a bound on NMMs ($\mu_\nu \lesssim 10^{-9}\mu_B$) [268, 303], which is still weaker than the $\nu + e$ scattering bound obtained by GEMMA ($\mu_\nu \lesssim 10^{-11}\mu_B$) [309]. However, the GEMMA bound only applies to $\nu_e$ while the bound from COHERENT applies to both $\nu_e$ and $\nu_\mu$.

- Other new physics

There are also studies on other new physics related to coherent neutrino scattering. For example, if neutrinos interact with dark matter, then dark matter could be produced in CE$\nu$NS. Fig. 172 demonstrates one of the possibilities, where an MeV dark matter particle appears in the final states, which would lead to distinct distortion of the recoil spectrum [271]. In addition, coherent neutrino scattering can also be used to search for sterile neutrinos, which has been explored in Refs. [277, 303, 310].

In summary, CE$\nu$NS opens a new channel to probe SM and BSM neutrino interactions. The recent data from the COHERENT experiment has provided important constraints on NSI, SPVAT, light mediators, neutrino magnetic moments, etc. Future reactor-based experiments have great potential to achieve high statistics and probe more neutrino-related new physics.

## 18    New Physics in Rare Neutrino Scattering (Hostert)

Testing the three neutrino oscillation paradigm requires precision measurements of oscillation parameters. The current and future accelerator experiments operate with neutrino energies of a few GeV, and so rely on near detectors (ND) with excellent particle identification (PID) capabilities to overcome the lack of theory and data on neutrino-nucleus cross sections at these energies. The DUNE ND, for instance, is expected to gather over $10^8$ CCQE events over its lifetime and its Liquid Argon (LAr) technology will provide improved PID on an event-by-event basis. Beyond allowing for precision in oscillation physics, these factors also bring about a rich program of neutrino interactions. In particular, searching or measuring rare neutrino processes can provide powerful insights into beyond the SM physics in the neutrino sector. In this section, we discuss some of the work developed in Refs. [311, 312] to show that with reasonable assumptions about detector size and POTs, the DUNE ND can probe with world-leading statistics a series of theoretically well-understood rare neutrino scattering processes, namely *neutrino-electron* scattering and *neutrino trident production*. Our projection of the measurement of these (semi-)leptonic scattering channels is then used to assess the sensitivity of DUNE to popular $Z'$ models, where the SM gauge group is extended by a new $U(1)$ with $L_e - L_\mu$ or $L_\mu - L_\tau$ charges.

| Design | Mode | $\mu^+\mu^-$ trident | $e^+e^-$ trident | $\nu - e$ scattering | POTs/year |
|---|---|---|---|---|---|
| 120 GeV $p^+$ | $\nu$ | 47.6 | 110 | 8930 | $1.1 \times 10^{21}$ |
| | $\overline{\nu}$ | 40.7 | 97.6 | 6450 | $1.1 \times 10^{21}$ |
| $\nu_\tau$ app optm | $\nu$ | 210 | 321 | 24900 | $1.1 \times 10^{21}$ |
| | $\overline{\nu}$ | 156 | 243 | 14700 | $1.1 \times 10^{21}$ |

**Table 181:** The SM rates for neutrino trident production and neutrino-electron scattering per year at the 75-t DUNE ND after kinematical cuts. The two different rows stand for different assumptions regarding the neutrino flux, the second row corresponding to a higher neutrino energy configuration.

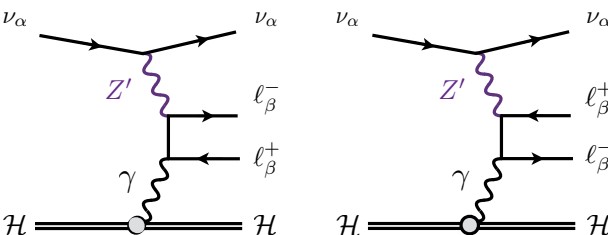

**Figure 181:** New physics contribution to neutrino trident production.

The neutrino-electron scattering cross section is well known, and has been measured with low-energy neutrinos [313–317], as well as with accelerator neutrinos with energies in the several GeV range at CHARM-II [318] and MINERvA [228, 319], the latter measurement serving as a neutrino flux measurement for the NuMI beam. The previous measurements provide some of the most stringent constraints on new vector mediators that might couple to neutrinos and matter in the MeV scale [305]. This cross section is of the order of $m_e/m_p \approx 1/2000$ of the CCQE, and so it is a rare process. The signal is a single electromagnetic shower satisfying $E_e\theta^2 < 2m_e$, its forward nature serving as the main feature to reduce backgrounds from $NC\pi^0$ and $\nu_e$CCQE processes. The measurement at DUNE will count with a large number of events, as shown in 181.

Neutrino trident scattering is a semi-leptonic scattering in the Coulomb field of the nucleus where a charged lepton pair is produced. For any incoming neutrino flavour with a few GeV in energy, $\mu^+\mu^-$, $\mu^\pm e^\mp$ and $e^+e^-$ pairs may be produced. Mixed charged lepton flavour channels only receive CC contributions, and, despite containing the largest event numbers, will not be discussed here. The only trident channel measured to date is the $\nu_\mu A \to \nu_\mu \mu^+\mu^- A$ first at CHARM-II [320] and later with increased precision at CCFR [321]. DUNE is planned to have $\gtrsim 90\%$ of its neutrino flux composed of $\nu_\mu$ ($\overline{\nu}_\mu$) in neutrino (antineutrino) mode, so the dominant channels for trident production are of the form $\nu_\mu A \to \nu_\mu \ell_\alpha^+ \ell_\alpha^- A$. Thus, if $\alpha = e$ only NC contributions exist, and if $\alpha = \mu$ then both NC and CC contribute, where a cancellation of $\approx 40\%$ due to interference. This process has been calculated in Refs. [322–325], and more recently in Ref. [311], where it was also

shown that the use of the Equivalent Photon Approximation can lead to unacceptably large errors in the cross section. The relevant cross sections are of the order $10^{-7} - 10^{-6}$ of the CCQE one, and therefore one expects backgrounds to be the limiting factor for such measurements. Backgrounds to dimuon tridents arise mainly from CC pion production, where a $\pi^\pm$ is mistaken for a $\mu^\pm$ in the detector (see also Ref. [326] for a more in depth discussion of this background at DUNE), while for dielectrons they come mainly from NC pion production, where a $\pi^0$ decaying into two photons can mimic the dielectron signature through $\gamma/e$ mis-ID.

The neutrino processes discussed above are ideal candidates to probe novel neutral currents beyond the SM. In fact, we can already expect precise measurements of these cross sections probe greater-than-weak interactions between neutrinos and charged leptons. For concreteness, we focus on the sensitivity of DUNE to two popular benchmark models for novel leptonic neutral currents. When extending the SM by a new Abelian $U(1)_X$ symmetry, one can show that gauging the differences of individual lepton number, $X = L_\alpha - L_\beta$, leads to a theory free of anomalies, requiring no additional fields ($B-L$ is anomaly free only if right-handed neutrino fields are added to the theory). Due to the flavour composition of the beam, we find that $L_e - L_\mu$ and $L_\mu - L_\tau$ are the only models for which DUNE has competitive sensitivity with other experiments. These models have connections to the flavour structure in neutrino mixing if right-handed neutrinos are introduced for neutrino mass generation via the Type-I seesaw mechanism [327], and so also may also shed light on the origin of neutrino masses. In minimal extensions, the $L_e - L_\mu$ model was shown to not be consistent with mixing data [328], however it has been extensively discussed in various phenomenological contexts. One interesting possibility is that it may lead to novel contributions to the neutrino matter potential in the form of long range forces [329, 330]. The $L_\mu - L_\tau$ model, on the other hand, has received great attention recently due to experimental anomalies and due to its poorly explored parameter space. The discrepancy between the measurement [331] and theory prediction [332, 333] of the muon $(g-2)$ can be explained through the exchange of the $Z'$. It is also easy to connect this model to $R_K$ experimental anomalies in the flavour sector [334], and to discrepancies in the Hubble expansion rate measurement [335]. Dimuon tridents have been shown to provide the leading bounds for light $Z'$ masses [336] since it is sensitive to muon-specific forces. Finally, kinetic mixing between the $Z'$ and hypercharge provides an additional way to constrain this model using neutrino-electron scattering measurements, albeit in a model dependent way.

In Fig. 182 we show the sensitivity of DUNE to the two models considered. We assume a 75 t LAr ND running for 5 years in neutrino and 5 years in antineutrino mode with $1.1 \times 10^{21}$ POTs/year. In the case of $L_e - L_\mu$, the most sensitive measurement is that of $\nu - e$ scattering. The degree with which DUNE can probe untested parameter space will depend strongly on the uncertainty achieved on the neutrino flux, currently at the level of $\approx 9\%$ at NuMI [337], but expected to improve significantly with hadro-production measurements [338] and with the use of neutrino-hadron scattering measurements [339]. Other $\nu - e$ scattering measurements have been used to constrain this model using TEXONO, BOREXINO and CHARM-II data [340]. For the $L_\mu - L_\tau$ model, the parameter space is

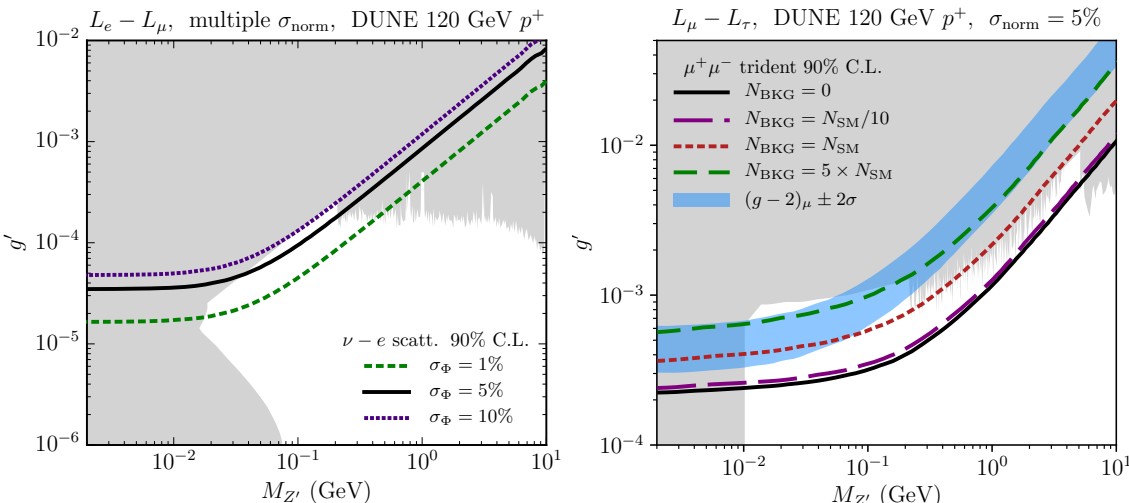

**Figure 182:** The DUNE ND sensitivity to $L_e - L_\mu$ (left) and $L_\mu - L_\tau$ (right) model with no kinetic mixing at 90% C.L. On the left panel we vary the normalization uncertainty from a pessimistic 10% to an aggressive 1% value, and on the right we vary the background to dimuon tridents from 0 to a total of 5 times the number of SM dimuon events after cuts.

much less constrained, and dimuon trident measurements at DUNE can bring about significant improvements. On the right panel of Fig. 182 we show the DUNE sensitivities as a function of the number of background events, the limiting factor of this measurement. A clear goal in this parameter space is the $g - 2$ explanation region, and we find that if backgrounds are below the total number of SM dimuon events after our kinematical cuts, DUNE will be able to probe over 90% of the allowed region.

The new physics scenarios we have considered so far cannot lead to observable enhancements to $e^+e^-$ trident signatures at DUNE in allowed parameter space [312]. Nevertheless, these signatures appear in a variety of new physics models where neutrino experiments provide very competitive bounds [194, 341]. For instance, recent explanations of the excess of electron-like events at MiniBooNE have been put forward [195, 221, 223], where the excess is due to dielectron pairs which are spatially overlapping or highly asymmetric in energy. These models provide an interesting alternative to endow active neutrinos with new interactions and have strong connections to neutrino mass generation at low scales [194, 222]. Testing the upscattering signatures of such models using an $e^+e^-$ trident search or measurement is a feasible goal of current and future neutrino experiments, and should be considered further.

A dedicated effort to look for rare processes, including multi-lepton final states, in neutrino experiments is an important goal for DUNE and other neutrino near detectors. We have shown that a neutrino-electron scattering and dimuon trident measurements can probe untested parameter space of minimal extensions of the SM. In addition, measuring dielectron tridents for the first time is also an important milestone, and can serve as a novel probe of new physics scenarios which are more complex than a simple mediator extension.

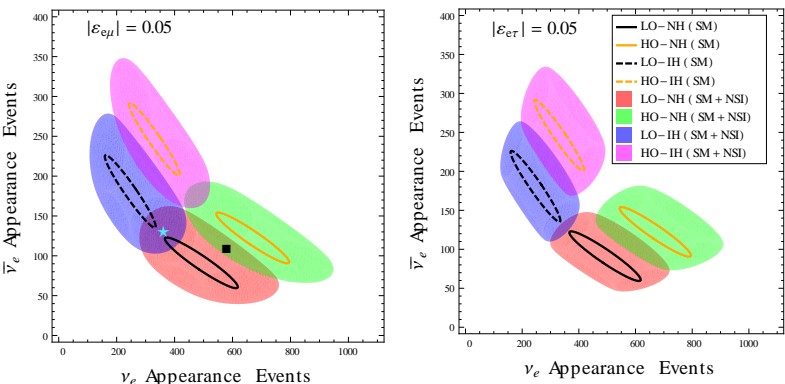

**Figure 191:** Figure represents the bi-event plot for DUNE setup. In case of SM ellipses, the running parameter is $\delta$, whereas in case of SM+NSI blobs, the running parameters are $\delta$ and $\phi_{e\mu}$ ($\delta$ and $\phi_{e\tau}$) in the left (right) panel respectively. For details please see the text and [31]. This figure has been taken from [31].

Overall, (semi-)leptonic scatterings are rare but theoretically well-understood processes. We now have a chance to return to the study of such processes with improved statistics and PID capabilities.

## 19 The Octant of $\theta_{23}$ and NSI Degeneracy (Chatterjee)

Recent data from T2K [342] and NOνA [343] and the current $3\nu$ global data analysis [115, 344, 345] hint towards the non-maximal value with two degenerate solutions of the atmospheric mixing angle $\theta_{23}$: one is $\theta_{23} < \pi/4$, known as lower octant (LO), and the other possibility is $\theta_{23} > \pi/4$, called higher octant (HO). The resolution of the octant of $\theta_{23}$ [346] ambiguity is one of the top most priorities of the neutrino oscillation experiments in the standard $3\nu$ framework. Long-baseline neutrino oscillation experiments (LBL) offer tremendous opportunity to resolve this issue with the help of $\nu_\mu \to \nu_e$ appearance channel which is well complemented by the $\nu_\mu \to \nu_\mu$ survival channel. However it is well known that there may exists several new physics possibilities which require the substantial modification of the standard framework. In addition, these new physics scenarios can have significant effects on the stadard $3\nu$ phenomena. For example, neutrinos may have different kind of new interactions, popularly known as neutrino non-standard interactions (NSI)[1]. In the past few years the work on NSI in the context of LBL experiments have received a lot of attention [13, 28, 30, 252, 347–356]. Here we discuss in detail the impact of non-diagonal flavor changing NSIs (specifically $\varepsilon_{e\mu}$ and $\varepsilon_{e\tau}$ taken one at a time) on the resolution of the octant of $\theta_{23}$ taking DUNE [23] as a case study. We show that even for small values of the NSI coupling strength and for unfavorable combinations of the Dirac and NSI CP-phases, the discovery potential of the octant of $\theta_{23}$ gets completely lost.

***Theoretical framework:*** NSIs can be of two types: one is charged current (CC) and other is neutral current (NC). CC NSI takes place in production and detection mechanism and

NC NSI occurs during the propagation of neutrinos through the matter. Here we focus on the neutral current NSI. A neutral-current NSI can be described by a four-fermion dimension-six operator [1] as given in Eq. (1.1). For other details on the neutral-current NSI parametrization and its appearance in the modified effective Hamiltonian, please see Sec. (1). Let us now discuss the behavior of the transition probability in presence of NSI. Following [357], it is shown in [31] that in the presence of NSI, the transition probability can be approximately written as the sum of three terms,

$$P_{\mu e} \simeq P_0 + P_1 + P_2 \,, \tag{19.1}$$

where the first two are the standard 3-flavor probability terms and the third one arises because of the presence of NSI. Treating $\sin \theta_{13}$, the matter parameter $v \, (\equiv 2V_{CC}E/\Delta m_{31}^2)$ and the modulus $|\varepsilon|$ of the NSI as the same order of magnitude $\mathcal{O}(\varepsilon)$, while $\alpha \equiv \Delta m_{21}^2/\Delta m_{31}^2 = \pm 0.03$ is $\mathcal{O}(\varepsilon^2)$, we can expand the probability terms upto third order as

$$P_0 \simeq 4s_{13}^2 s_{23}^2 f^2 \,, \tag{19.2}$$

$$P_1 \simeq 8s_{13}s_{12}c_{12}s_{23}c_{23}\alpha f g \cos(\Delta + \delta) \,, \tag{19.3}$$

$$P_2 \simeq 8s_{13}s_{23}v|\varepsilon|[af^2 \cos(\delta + \phi) + bfg \cos(\Delta + \delta + \phi)] \,, \tag{19.4}$$

where we have used the same notation as [28, 358]. $\Delta \equiv \Delta m_{31}^2 L/4E$, and,

$$f \equiv \frac{\sin[(1-v)\Delta]}{1-v} \,, \qquad g \equiv \frac{\sin v\Delta}{v} \,, \tag{19.5}$$

$$a = s_{23}^2 \,, \quad b = c_{23}^2 \qquad \text{if} \qquad \varepsilon = |\varepsilon_{e\mu}|e^{i\phi_{e\mu}} \,, \tag{19.6}$$

$$a = s_{23}c_{23} \,, \quad b = -s_{23}c_{23} \qquad \text{if} \qquad \varepsilon = |\varepsilon_{e\tau}|e^{i\phi_{e\tau}} \,. \tag{19.7}$$

Now to get a sensitivity for distinguishing the two octants, we must satisfy the following condition:

$$\Delta P \equiv P_{\mu e}^{\text{HO}}(\theta_{23}^{\text{HO}}, \delta^{\text{HO}}, \phi^{\text{HO}}, ..) - P_{\mu e}^{\text{LO}}(\theta_{23}^{\text{LO}}, \delta^{\text{LO}}, \phi^{\text{LO}}, ..) \neq 0 \,. \tag{19.8}$$

where one of the two octants should be considered to generate data and the other octant should be used to simulate the theoretical model. $\Delta P$ can be written as,

$$\Delta P = \Delta P_0 + \Delta P_1 + \Delta P_2 \,. \tag{19.9}$$

$\Delta P_0$ is positive-definite. $\Delta P_1$ depends on $\delta$, whereas $\Delta P_2$ depends on both the CP-phase $\delta$ and the NSI-phase $\phi$, and they can be both positive or negative. The exact expressions for these terms are given in [31]. It is evident from Eq. (19.9) that to achieve the octant discovery reach at certain confidence level, $\Delta P$ term must not get vanished completely due to the positve and negative combinations of $\Delta P_0$, $\Delta P_1$ and $\Delta P_2$ respectively.

***Results and discussion:*** We have performed the DUNE [23] simulations assuming a total 248 kt.MW.yr of exposure[6] shared equally between neutrino and antineutrino mode

---

[6]We have checked that the results remain almost similar even with the 300 kt.MW.yr of exposure used in DUNE-CDR [23].

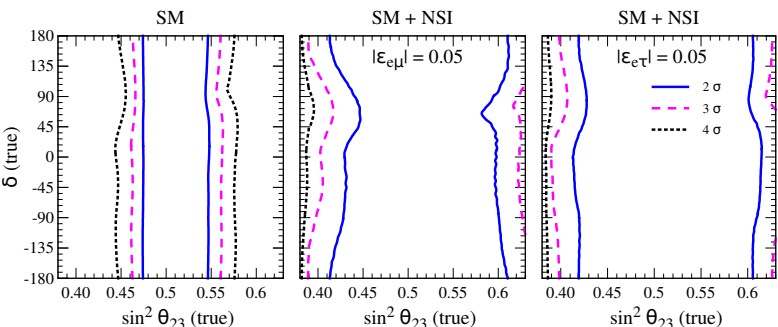

**Figure 192:** Discovery reach of the octant $\theta_{23}$ assuming NH as true choice. The left, middle and right panels correspond to SM, SM+NSI with $|\varepsilon_{e\mu}| = 0.05$ and SM+NSI with $|\varepsilon_{e\tau}| = 0.05$ cases respectively. In the SM case, we have marginalized away $(\theta_{23}, \delta)$ (test). The solid blue, dashed magenta, and dotted black curves depict the $2\sigma$, $3\sigma$, and $4\sigma$ confidence levels (1 d.o.f.) discovery reach. In SM, we have marginalized over CP-phase $\delta$ and $\theta_{23}$ (opposite octant), while in SM+NSI cases, in addition, we have marginalized over the true and test values of the NSI CP-phases ($\phi_{e\mu}$ ($\phi_{e\tau}$) in the middle (right) panel respectively. For more details please see [31]. This figure has been taken from [31].

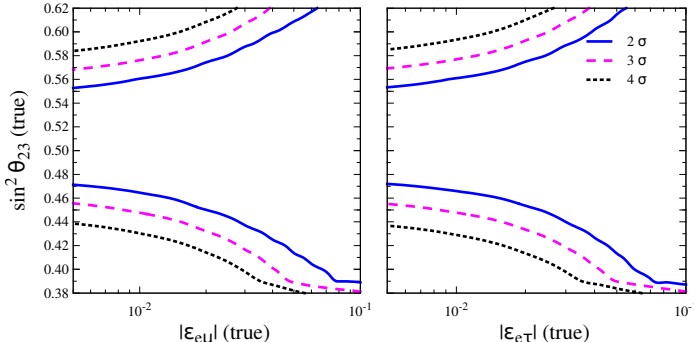

**Figure 193:** Octant of $\theta_{23}$ discovery reach as a function of $|\varepsilon|$ (true) assuming NH as true choice. The solid blue, dashed magenta, and dotted black curves depict the $2\sigma$, $3\sigma$, and $4\sigma$ confidence levels (1 d.o.f.). We have fixed the $|\varepsilon|$ both in data and theory. In both the panel we have marginalized over the hierarchies, $\theta_{23}$ (opposite octant), and $\delta$ (both in data and theory). In addition, the NSI CP-phases $\phi_{e\mu}$ and $\phi_{e\tau}$ have been marginalized away both in data and theory in the left and right panel respectively. This figure has been taken from [31].

(see [31]). Fig. (191) represents the bi-event plot. The solid and dashed ellipses (colored blobs) correspond to the standard $3\nu$ ($3\nu$+NSI) scenario. The left (right) panel is for

the fixed NSI coupling strength $|\varepsilon_{e\mu}|^7 = 0.05$ $(|\varepsilon_{e\tau}| = 0.05)$ and we have also assumed $\sin^2\theta_{23} = 0.42\,(0.58)$ as a benchmark value for the LO (HO). It is clear that the ellipses are well separated from each other in $3\nu$ framework. However in presence of NSI the ellipses become blobs due to the convolution of the different combinations of $\delta$ & $\phi_{e\mu}$ ($\delta$ & $\phi_{e\tau}$) in the left (right) panel respectively. Concentrating on a fixed hierarchy we see that the blobs overlap with each other due to the presence of new phases in the term $\Delta P_2$, clearly indicating a new degeneracy between the octant and the NSI CP-phases [31]. It is worth to note here that a degeneracy between the normal and inverted hierarchy due to a small overlap can be broken using the spectral information [359] of DUNE's wide band beam. Fig. (192) displays the sensitivity of distinguishing one octant of $\theta_{23}$ from the other in $\left[\sin^2\theta_{23}, \delta\right]$ (true) plane. Solid blue, dashed magenta, and dotted black represent the $2\sigma$, $3\sigma$, and $4\sigma$ C.L. reach of DUNE. Left, middle, and right panels correspond to the sensitivities in the SM, SM + NSI with $|\varepsilon_{e\mu}| = 0.05$, and SM + NSI with $|\varepsilon_{e\tau}| = 0.05$ respectively. It is worth to note that we have fixed $|\varepsilon_{e\mu}|$ and $|\varepsilon_{e\tau}|$ both in data and theory while we have marginalized over the NSI CP-phases $\phi_{e\mu}$ and $\phi_{e\tau}$ from $-\pi$ to $\pi$ both in data and theory. For details of the simulation method adopted for the analysis please see [31]. It is evident from the figure that in case SM, a minimum of $2\sigma$ sensitivity can be achieved for $\sin^2\theta_{23} \leq 0.475$ and $\sin^2\theta_{23} \geq 0.545$, whereas in presence of NSI, the discovery reach of the octant of $\theta_{23}$ sensitivity gets substantially lost and a minimum of $2\sigma$ sensitivity can only be assured in a very small parameter space of the entire $\left[\sin^2\theta_{23}, \delta\right]$ (true) plane. Finally it is interesting to ask how the sensitivity changes with the various values of the NSI coupling strength and in that context Fig. (193) represents the desired analysis. Assuming NH as the true choice of hierarchy, we show that for the increasing values of NSI coupling strength (fixed both in data and theory) the discovery reach of $\theta_{23}$ octant decreases and even for a small value of $|\varepsilon|$ (3% to 4% relative to $G_F$), the sensitivity gets completely lost. For the details of the simulations please see [31]. So in the near future if the prediction of NSI becomes reality then the resolution of the octant of $\theta_{23}$ at a good confidence level may require precise information on both the modulous of the NSI coupling strengths and the NSI CP-phases. As a whole it is very clear from here as well as from all other kind of works on NSI available in the literature, that NSI's not only can be a source of confusion but also they can pose a great challenge in disentangling their effects from the standard 3-flavor effects.

## 20 Beginnings of Quantum Monte Carlo Short-Time Approximation Implementations for $l^4$He and $l^{12}$C Scattering in GENIE (Barrow)

### 20.1 Linkages between $\nu$ experiments and $\nu$ interaction simulations

It cannot be understated the importance and difficulty of mapping experimentally derived, "final state" neutrino ($\nu$) properties and energies onto initial $\nu$ states given the complexity of target nuclear systems. Many technicalities and their interrelations limit the certainty of interpretations of a given experiment's results, including:

---

[7]It is worth to note here that the notations of NSI coupling strength used in the text and in all the figures serve the same purpose. For the present constraints on NSI parameters, please see [24, 25].

1. **The $\nu$ cross section model:** This is a function of the precise mathematical structure of the $\nu$ interaction (standard, non-standard, their relative strengths, etc.), along with the degrees of freedom such interactions are assumed to probe

2. **The $\nu$ flux (or beam) model:** This is a function of one's knowledge of the rates of largely hadronic scattering and decay processes upstream of a detector, leading to beam energy and angular distribution uncertainties

3. **The model of the nuclear target:** This is a function of the interaction's momentum transfer, probing different levels of nuclear structure

4. **The nuclear final state interaction model:** This is a function of the identity of the probed particle, the interaction's momentum transfer, and any intranuclear cascade, particle knock-out or emission, break-up, de-excitation, etc. which can constitute all possible observable final states

5. **The detector's ability to reconstruct a $\nu$ event:** This is a function of the $\nu$ identity, interaction type, the outgoing particle energies and multiplicities, detection medium and efficiency, noise, background rates, and other nontrivial detector effects

All of these can (currently) only be efficiently modeled using Monte Carlo (MC) $\nu$ event generators, a popular candidate being GENIE [360]. Many experiments rely on such simulation to model their beam, $\nu$ interactions, final state topologies, and even detector responses on an event-by-event basis; together, these smear the initial $\nu$ energy and properties. To understand $\nu$ properties, all these separate yet inextricably linked processes must be understood enough to quantify their collective uncertainties, all small enough to ascertain unmeasured or not well known properties, thus constraining nonstandard physics. Given these, necessities include development of precise $\nu$ total inclusive cross sections, their implementation in $\nu$ event generators such as GENIE for use in experiments, and their validation using lepton ($l$) scattering data.

## 20.2 Short-time approximation calculations of $l^4He$ and $l^{12}C$ cross sections

*Most* nuclear models in MC event generators are based on an inherently single nucleon interaction paradigm. However, while this constitutes a fine first-order approximation, the advent of high Bjorken $x$ experiments at $e$ beam laboratories have led to knowledge of multinucleon, short range interactions in many nuclei [361], emphasizing the correlated nature of nucleons' momenta. Thus, one must construct a more complete formalism of nuclear dynamics which accounts for many-body nuclear effects. One way to consider this problem is using Quantum Monte Carlo (QMC) [362] computational methods to solve the nuclear many-body problem. Nuclei are systems made of strongly correlated nucleons described by a nuclear Hamiltonian consisting of two- and three-body potentials which correlate nucleons in pairs and triplets. QMC methods allow one to solve the associated many-body Schrödinger equation to obtain both static (energies, form factors, initial states) and dynamic (nuclear responses, cross sections, and decay rates) properties of nuclei. The QMC community has so far delivered exact calculations of inclusive nuclear responses

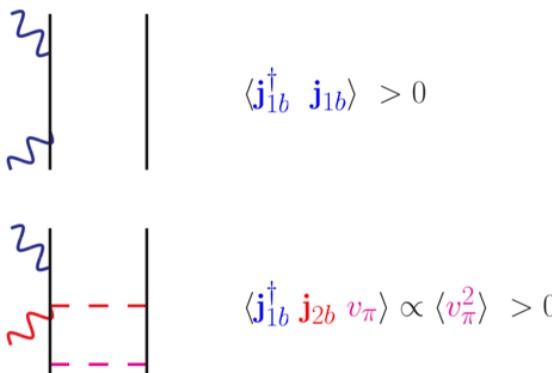

$\langle \mathbf{j}_{1b}^{\dagger} \ \mathbf{j}_{1b} \rangle \ > 0$

$\langle \mathbf{j}_{1b}^{\dagger} \ \mathbf{j}_{2b} \ v_{\pi} \rangle \propto \langle v_{\pi}^2 \rangle \ > 0$

**Figure 201:** Lepton scattering via inherently two body objects is considered via chiral effective field theory Feynman diagrams in single and interference interactions.

induced by $e$'s and $\nu$'s [363, 364]. These studies indicate that two-body physics, namely two-body correlations and associated two-body currents (describing the interactions of the external probe with pairs of correlated nucleons) are essential to explain the available scattering data. However, QMC methods are computationally limited to $A \lesssim 14$ nuclei.

The double-differential inclusive lepton ($l$) scattering cross sections for a given four momentum transfer $q^{\mu} = (\omega, \mathbf{q})$ can be factorized into longitudinal ($L$) and transverse ($T$) polarizations as

$$\frac{d^2\sigma}{dE'd\Omega'} = \sigma_M[\nu_L R_L(\mathbf{q}) + \nu_T R_T(\mathbf{q})], \ R_\alpha = \sum_f \delta(\omega + E_0 - E_f)|\langle f| \mathcal{O}_\alpha(\mathbf{q}) |0\rangle|^2$$

where $\nu_\alpha$, with $\alpha = L, T$, is a kinematic factor associated with the incoming lepton, $\sigma_M$ is the Mott cross section, and $R_\alpha$ is the nuclear response function, $\delta(\cdots)$ is the energy-conserving $\delta$-function, and all possible final states encapsulating all interaction vertex information–encoded by the transition operators $O_\alpha$–are called $|f\rangle$. Using the integral representation of the $\delta$, the equation above can be recast as

$$R_\alpha = \int dt \, \langle 0| \mathcal{O}_\alpha^{\dagger}(\mathbf{q}) e^{i(\hat{H}-\omega)t} O_\alpha(\mathbf{q}) |0\rangle .$$

In order to retain two-body physics, the short-time approximation (STA) [365] of the time evolution operator is taken as

$$e^{i(\hat{H}-\omega)t} = e^{i(\sum_{i=1}^A t_i + \sum_{i<j}^A v_{i,j} + \cdots - \omega)t} \approx \sum_i t_i + \sum_{i<j} v_{i,j} = P(t),$$

where the many-body Hamiltonian is expanded to include up to two-body terms. Here, $t_i$ is the kinetic energy of a single nucleon, and $v_{ij}$ is a two-nucleon interaction. In addition, one allows the probe to couple with $i$) individual nucleons via a one-body current operator $O_{\alpha;i}$, as well as $ii$) pairs of correlated nucleons via a two-body current operator $O_{\alpha;i,j}$. Thus,

$$\mathcal{O}_\alpha^{\dagger}(\mathbf{q}) e^{i(\hat{H}-\omega)t} \mathcal{O}_\alpha(\mathbf{q}) \sim \mathcal{O}_{\alpha;i}^{\dagger} P(t) \mathcal{O}_{\alpha;i} + \mathcal{O}_{\alpha;i}^{\dagger} P(t) \mathcal{O}_{\alpha;j} + \mathcal{O}_{\alpha;i}^{\dagger} P(t) \mathcal{O}_{\alpha;i,j} + \mathcal{O}_{\alpha;i,j}^{\dagger} P(t) \mathcal{O}_{\alpha;i,j}.$$

This approximation allows one to fully retain two-body dynamics. The response function can then be expressed in terms of pairs of correlated nucleon states $|\mathbf{p}', \mathbf{P}'\rangle$, where $\mathbf{p}'$ and $\mathbf{P}'$ are the relative and center of mass momenta of the (*semi-*)final state nucleon pair, as

$$R_\alpha(\mathbf{q}) \sim \int d\Omega_{P'} d\Omega_{p'} dP' dp' p'^2 P'^2 \langle 0| \mathcal{O}_\alpha^\dagger(\mathbf{q}) |\mathbf{p}', \mathbf{P}'\rangle \langle \mathbf{p}', \mathbf{P}'| \mathcal{O}_\alpha(\mathbf{q}) |0\rangle \cdot \delta(\omega + E_0 - E_f) = \cdots$$

$$\cdots = \int dP' dp' \mathcal{D}(\mathbf{p}', \mathbf{P}'; \mathbf{q}) \delta(\omega + E_0 - E_f),$$

where all primed variables represent post-interaction, semi-final quantities at the vertex, and $\mathcal{D}(\mathbf{p}', \mathbf{P}'; \mathbf{q})$ is the two-body response *density*.

With supercomputers, one may calculate these densities for the quasielastic energy transfer region for all permissible total and relative nucleon pair momenta, creating large tabulated data sets. These response densities have multiple facets, and can be broken down into particular nucleon pairs ($pp$, $np$, $nn$) and their dependencies on one- and two-body physics extracted for a given momentum transfer. Of particular interest is the effects of one- and two-body interference, something no previous calculation has accounted for, and which can be rather large, enhancing the total inclusive cross section by $\sim$ 5-10% (the electromagnetic component can be much more, up to $\sim$ 30-40%).

## 20.3 Developing a new implementation within GENIE for $^4He$ and $^{12}C$

A tool has been made by Steven Gardiner (FNAL) for GENIE to interface with these response tables, calculate cross sections; in general, this information can be fed into the GENIE event generator (which passes particle dynamics information from the model to intranuclear cascade module for transport). Scaling these cross sections and their underlying dynamics to larger nuclei has not yet been investigated. Preliminary design of this generator module is complete, which will now be briefly described; coding work will begin soon.

One of the quantum mechanical limitations of this new momentum-based formalism, in contraversion to most event generators, is the lack of available angular/positional information in the nucleus; one cannot know a particle's location and momentum simultaneously. This means one must use other information to pass necessary event configurations to GENIE, such as single nucleon position and two-nucleon separation [366] distributions to permit (semi-torroidally degenerate) triangulation in the nucleus. In concert with the tabulated nuclear response densities containing nucleon pair momentum information for given values of momentum transfer, and assuming a uniform or Guassian two-body angular distribution, via Laws of Cosines, one can reconstruct the semi-final state in the nucleus before transport through the intranuclear cascade; more elegant methods involving a geometric interpretation of the responses may also be possible.

## 20.4 Conclusions

Once complete, validation of this generator on quasielastic $e$ scattering data will proceed. A goal of this comparison will be to understand dependence of two-nucleon final state multiplicities on initially correlated intranuclear systems; this could directly constrain free

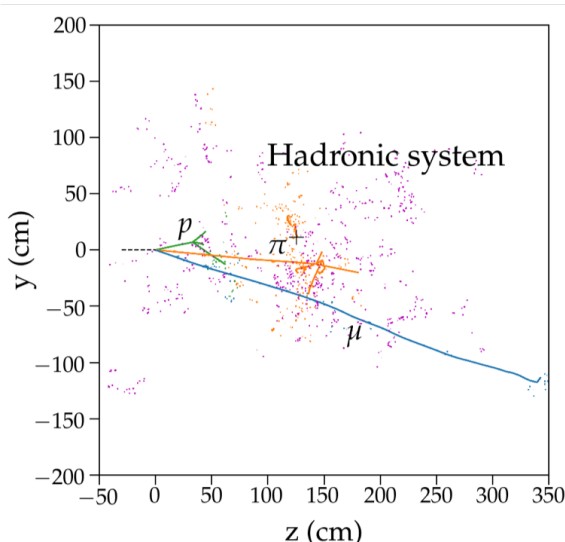

**Figure 211:** An example simulated 4 GeV $\nu_\mu$ event using GENIE and FLUKA. The magenta energy deposits are caused by neutrons undergoing multiple scatterings; the orange color denotes energy originally carried by the prompt charged pion. The figure is taken from Ref. [368].

parameters in the intranuclear cascade in GENIE. After this, $\nu$ scattering comparisons in the quasielastic region will begin. The hope is that this more complete simulation can help clarify some extraordinary measurements in some past experiments, possibly directly constraining nonstandard physics in the process. Altogether, we hope this will reduce the uncertainties in the technical points mentioned above, aiding experimentalists in the precise reconstruction of $\nu$ properties for use in beam and atmospheric oscillation searches.

## 21 Basic Considerations of Liquid Argon Technologies (Li)

There is a suite of liquid argon neutrino experiments in operation and more will come on line in the future. These experiments will pin down the last unknown neutrino mixing parameters, i.e., the octant of $\theta_{23}$, mass hierarchy, and $\delta_{\rm CP}$. They will also qualitatively improve our test of the 3-flavor neutrino oscillation paradigm, as shown in previous sections of this report.

As liquid argon technology is still in its infant stage, much of its capability is not known precisely. For many experimental inputs relevant for phenomenological work, e.g., neutrino energy resolution and particle thresholds, numbers in the literature vary by a factor of a few or even orders of magnitude [23, 367–369] and are being actively investigated. It is therefore important in phenomenological works to choose experimental inputs carefully. It may even be beneficial to vary experimental inputs in a reasonable range, and demonstrate how these inputs affect physics sensitivities.

One can assess many experimental inputs based on a few considerations. Figure 211 shows a 4 GeV $\nu_\mu$ charged-current event in liquid argon, simulated by the event gener-

ator GENIE and particle propagation code FLUKA. A neutrino was injected at (0, 0), and it produces a proton, a $\pi^+$, a muon, and two neutrons. All charged particles lose energy continuously by ionization. These ionized electrons get drifted in a detector and collected and they are the direct observables. The ionization rates are nearly particle-independent and energy-independent, $\simeq 3$ MeV/cm, except that protons have larger ionization rates, $\sim 10$ MeV/cm. A good estimate of the distance a charged particle travels is $d \simeq E_k/(3\text{MeV/cm})$, where $E_k$ is the kinetic energy.

Occasionally, protons, neutrons, and pions could hadronically interact with an argon nucleus, break it up, and produce a few daughter particles. This happens at about (40, 10) and (140, -10) in Fig. 211. This process not only produces lower-energy particles, it is also an important channel for energy loss, i.e., some kinetic energy of the parent particle is spent to overcome the binding energy of the nucleus, typically tens of MeV per interaction. This interaction also has an almost particle-independent and energy-independent rate, with an interaction length of about 1 m.

Neutrons are difficult to detect. Occasionally, a neutron may have a hard interaction and knock out an energetic proton, which can be easily detected. But often, neutrons bounce around in liquid argon and only leave some de-excitation gammas (magenta points in Fig. 211). The detectability of these soft gammas is highly uncertain.

Though there are many subtleties in how well a detector can perform, basic physics considerations can offer a guideline. For example, a few MeV particles should travel a couple of cm, a few times the wire spacing of DUNE. Detection thresholds much higher than the scale are likely to be conservative, and thresholds even lower than the scale are likely to be aggressive, etc. Rapid progress is made towards fully understanding liquid argon technologies. Until then, it is important for everyone to be conscious about experimental inputs.

## 22 New Physics with DUNE Alternative Configurations - the Role of High Energy Beam Tunes (Mehta)

The proposed Deep Underground Neutrino Experiment (DUNE) utilizes a wide-band on-axis tunable muon-(anti)neutrino beam with a baseline of 1300 km to search for CP violation with high precision [23]. Given the long baseline, DUNE is also sensitive to effects due to matter induced non-standard neutrino interactions (NSI) as given by Eq. 1.1 which can interfere with the standard three-flavor oscillation paradigm.

In the presence of new physics effects, clean extraction of the CP violating phase becomes a formidable task [371–373]. In fact, a given measured value of CP phase could very well be a hint of new physics [351, 374]. In earlier works, it has been pointed out that there are degeneracies within the large parameter space in the presence of non-standard interactions (NSI) [12, 14, 30, 31, 252, 347–349, 353, 354, 375–381]. The need to devise ways to distinguish between the standard paradigm and new physics scenarios has been extensively discussed. Hence it is desirable to design strategies to disentangle effects due to NSI from standard oscillations.

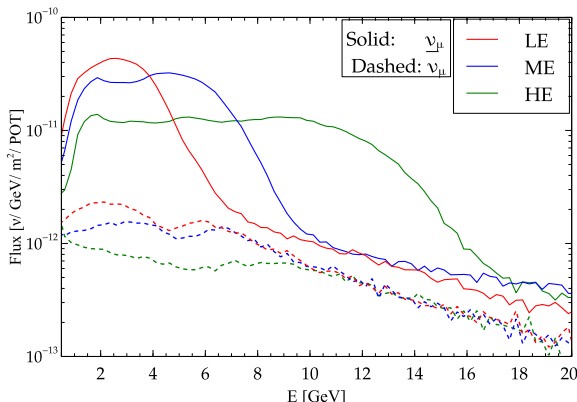

**Figure 221:** Comparison of the different flux tunes (LE, ME, HE) in the neutrino running mode. POT stands for protons on target. Taken from [370].

The standard neutrino flux (referred to as low energy (LE) beam) is peaked at energy values close to the first oscillation maximum ($E \sim 2.5$ GeV for DUNE). So, when the large CP asymmetry prominent at higher energies at the probability level is folded with the standard LE flux to generate the events, the difference between standard and new physics is masked because of the falling flux. It is therefore worthwhile and timely to ask if we can suitably tap the large signal of CP asymmetry at higher energies using higher energy beams. The beam tunes considered are: LE; medium energy (ME); and high energy (HE) as shown in Fig. 221.

## 22.1  Extricating physics scenarios with high energy beams

Here we exploit the tunability of the DUNE neutrino beam over a wide-range of energies to devise an experimental strategy for separating oscillation effects due to NSI from the standard three-flavor oscillation scenario. Using $\chi^2$ analysis, we obtain an optimal combination of beam tunes and distribution of run times in neutrino and anti-neutrino modes that would enable DUNE to isolate new physics scenarios from the standard. We can distinguish scenarios at $3\sigma$ ($5\sigma$) level for almost all ($\sim 50\%$) values of $\delta$. For more details, we refer the reader to [370].

We discuss the impact of using different beam tunes and run time combinations on the separability of physics scenarios.

(i) Impact of beam tunes on the event spectrum :- We show the variation in the $\nu_e$ event spectrum in Fig. 222 for the LE, ME and HE beam tunes under SI and NSI scenarios. For all the beam tunes, the red dashed line corresponds to $\delta = -\pi/2$ with NSI, green dashed line corresponds to $\delta = +\pi/2$ with NSI and the cyan band is for SI for $\delta \in [-\pi, \pi]$. The backgrounds are shown as grey shaded region. The black dashed lines (for $\delta = 0$ with NSI) lie farthest apart from the cyan band (SI) which means that one expects better separability between the two considered scenarios at values of $\delta \sim 0$ (or $\pm\pi$).

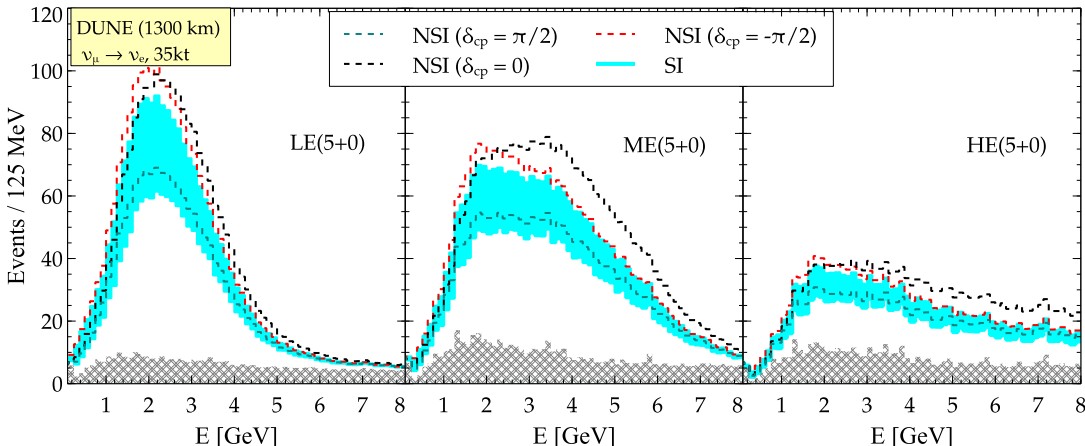

**Figure 222:** Separation between SI $\nu_\mu \to \nu_e$ events (cyan band, red and magenta dashed lines) and NSI $\nu_\mu \to \nu_e$ events (black dashed lines) at DUNE with LE (5+0), ME(5+0) and HE(5+0) beam tunes. The black dashed line is for a CP conserving NSI scenario. The cyan band corresponds to the SI case with the full variation of $\delta$. The dashed lines are for NSI case with different true values of $\delta$. The background events are similar for all the four cases and are shown as grey shaded region. Taken from [370].

(ii) Impact of beam tunes on extricating physics scenarios :- In Fig. 223, we show the ability of DUNE to separate SI from NSI using different combinations of beam tunes and running times at the $\chi^2$ level (as a function of true $\delta$). The left column is for an equal distribution of run time among neutrino and anti-neutrino modes while the right column corresponds to running in neutrino-only mode with the same total run time. A CP conserving NSI scenario is assumed in this plot (we assume $\phi_{e\mu} = \phi_{e\tau} = 0$). We have considered a combination of appearance ($\nu_\mu \to \nu_e$) and disappearance ($\nu_\mu \to \nu_\mu$) channels. The solid and dashed lines assume a beam power of 1.2 MW for both LE and ME beam tunes. The dotted black line corresponds to an ME option upgraded to 2.4 MW which is planned for later stages of DUNE. We note that the dominant channel contributing to the distinction of different physics scenarios is the $\nu_\mu \to \nu_e$ channel irrespective of our choice of the beam tune. The $\nu_\mu \to \nu_\mu$ channel adds somewhat ($\sim 1.5 - 2\sigma$ near the peak value at $\delta = 0$) to the total sensitivity but the $\nu_\mu \to \nu_\tau$ contribution is negligible.

(iii) Impact of beam tunes on extricating physics scenarios via the fraction plots :- Another important factor driving the sensitivity to SI-NSI separation is the fraction of values of CP phase for which the sensitivity is more than $3\sigma$ or $5\sigma$. In Fig. 224, the fraction of $\delta$ values for which the sensitivity lies above $3\sigma$ (magenta) and $5\sigma$ (blue) is plotted as a function of the run time for a combination of LE and ME (or HE) tuned beams. Both the panels are for a total run time of 10 years : the left one showing the case of 5 years of $\nu$ and 5 years of $\bar{\nu}$ run time while the right panel depicting the scenario of 10 years of $\nu$ run time alone. In the left panel, the 5+5 years of run time are distributed among the LE and ME (HE) beams for the solid (dashed) lines in the

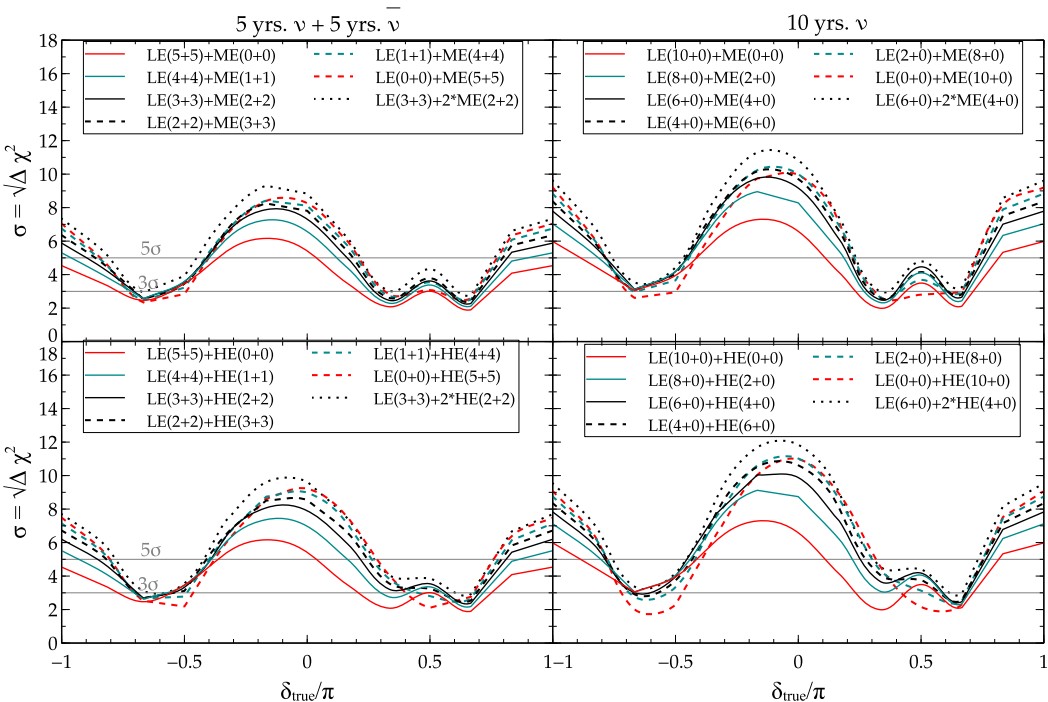

**Figure 223:** Separation between SI and NSI events at DUNE with different beam tunes at $\chi^2$ level. A CP conserving NSI scenario is assumed. The left column shows 5 years of neutrino and 5 years of anti-neutrino run times, while the right column depicts the case of 10 years of neutrino run time only. The plot is taken from [370].

following manner: ($x$ years of $\nu$ + $x$ years of $\bar{\nu}$) of LE beam $+\big((5-x)+(5-x)\big)$ years of ME or HE runtime. Similarly, the runtime in the right panel has been distributed as $(x+0)$ years of LE beam $+\big((10-x)+0\big)$ years of ME or HE run time.

We wish to stress that the fraction curves in Fig. 224 only show what portion of the sensitivity curve lies above $3\sigma$ (or $5\sigma$), and not necessarily the absolute value of the sensitivities. The estimate of the fraction of $\delta$ values thus depends on the points of intersection of the sensitivity curve with the $3\sigma$ (or $5\sigma$) horizontal lines in Fig. 223.

## 22.2 Probing the NSI parameter space at the level of $\chi^2$ with different beam tunes at DUNE

As another example of usefulness of high energy beams, we have explored the correlations and degeneracies among the NSI parameters at the level of $\chi^2$ using different beam tunes in [382]. We have considered the standard LE as well as ME beam tunes. Our main results are summarized in Fig. 225 where we depict contours at a confidence level (c.f.) of 99%. The solid cyan (black hatched) contours correspond to LE (LE + ME) beams. More specifically, the region enclosed by these contours depict the regions where there is SI-NSI degeneracy for those pair of parameters.

Below, we discuss some noteworthy features as can be observed from Fig. 225:

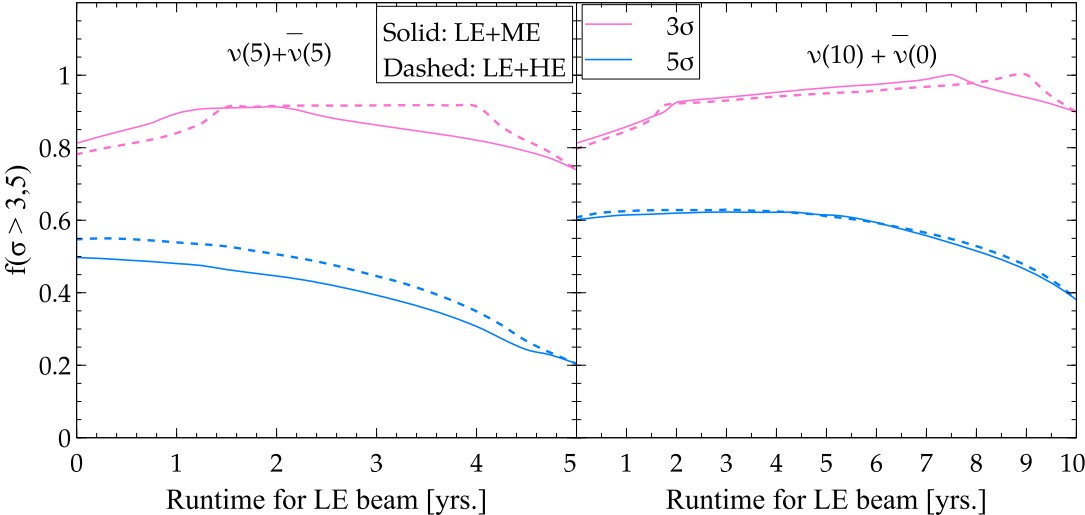

**Figure 224:** The fraction of the values of $\delta$ for which SI and NSI scenarios can be distinguished above $3\sigma$ (magenta) and $5\sigma$ (blue) using different combinations of beam tunes (LE+ME or LE+HE). The plot is taken from [370].

1. Let us first consider the panels with $\varepsilon_{e\mu}$ (either $|\varepsilon_{e\mu}|$ or $\varphi_{e\mu}$ or both) which are shown in light yellow colour. We note that use of different beam tunes (ME in conjunction with the LE beam) offers visible improvement of results (shrinking of contours) in these pairs of parameters. In order to explain the observed pattern, let us recollect that the presence of $|\varepsilon_{e\mu}|$ or $\varphi_{e\mu}$ leads to large difference between SI and NSI scenarios even at larger values of energies i.e., $E \gtrsim 4$ GeV. Thus, with the LE+ME option we are able to place tighter constraints on the parameter space corresponding to parameters $|\varepsilon_{e\mu}|$ and $\varphi_{e\mu}$.

2. We next consider the remaining panels in which we see that there is very little or no improvement of results after using the ME beam along with the LE beam. If we look at the pair of parameters, $|\varepsilon_{e\tau}| - \varepsilon_{ee}, \varphi_{e\tau} - \varepsilon_{ee}, \varepsilon_{\tau\tau} - \varepsilon_{ee}, \varphi_{e\tau} - |\varepsilon_{e\tau}|$ and $\varepsilon_{\tau\tau} - |\varepsilon_{e\tau}|$ in particular, we note that the degenerate regions get enlarged slightly. This is because of the fact that the presence of $\varepsilon_{e\tau}$, unlike $\varepsilon_{e\mu}$, actually adds to the SI-NSI degeneracy at higher energies.

3. For the panels with $|\varepsilon_{\mu\tau}|$ and $\varphi_{\mu\tau}$ as one of the parameters, there is very marginal improvement (except when $|\varepsilon_{e\mu}|$ or $\varphi_{e\mu}$ is present) in the degenerate contours using the LE+ME beam. Thus, even when $|\varepsilon_{\mu\tau}|$ is present, the $\Delta\chi^2$ receives dominant contribution from the $\nu_\mu \to \nu_e$ channel. This is more clear from the panels showing the parameter space associated to $\varphi_{\mu\tau}$ (i.e., where $\varphi_{\mu\tau}$ is not marginalised). The magnitude of $\Delta\chi^2$ in such panels is dominantly contributed by the $\nu_\mu \to \nu_\mu$ channel for all values of $\varphi_{\mu\tau} \not\approx \pm\pi/2$. But around $\varphi_{\mu\tau} \approx \pm\pi/2$, the contribution from the $\nu_\mu \to \nu_\mu$ becomes very small and $\nu_\mu \to \nu_e$ channel dominates, as we have also verified numerically. This explains the appearance of degenerate contours at $\varphi_{\mu\tau} \approx \pm\pi/2$ as

well.

4. All the parameter spaces showing $\varepsilon_{ee}$ (entire 2nd column and the top panel of the 1st column) have an additional degeneracy around $\varepsilon_{ee} \approx -2$, in addition to the true solution at $\varepsilon_{ee} \approx 0$. This extra solution comes due to the marginalisation over the opposite mass hierarchy. Similar degeneracy has also been observed in previous studies: in [16, 26, 30] (in the context of NSI) and also in [383] in the context of Lorentz violating parameters.

For details, we refer the reader to [382].

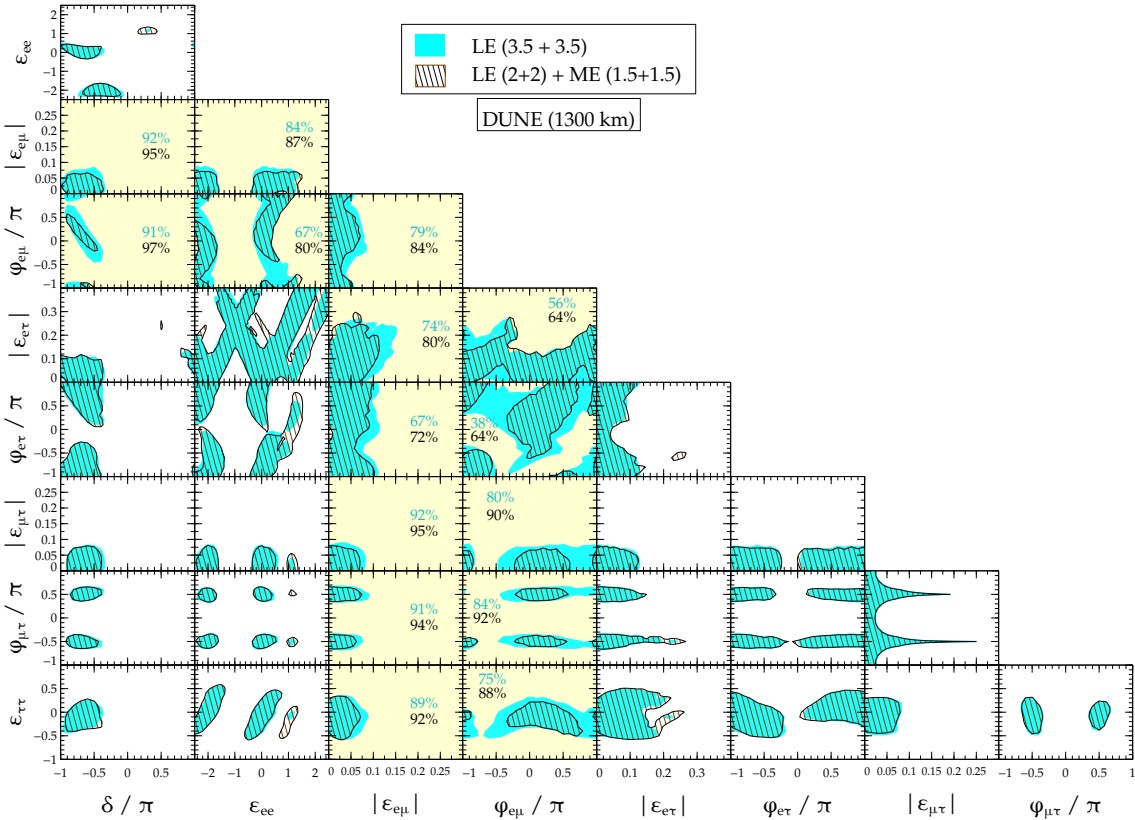

**Figure 225:** A comparison of the sensitivity of DUNE to probe the NSI parameters at 99% confidence level when a standard low energy (LE) beam tune is used (cyan region) and when a combination of low and medium energy (LE + ME) beam tune is used (black hatched region), keeping the total runtime same (3.5 years of $\nu$ + 3.5 years of $\bar{\nu}$ run) for both scenario. In the latter case, the total runtime is distributed between the LE beam (2 years of $\nu$ + 2 years of $\bar{\nu}$) and the medium energy beam (1.5 years of $\nu$ + 1.5 years of $\bar{\nu}$). The panels with a light yellow (white) background indicate significant improvement (no improvement) by using LE + ME beam over using LE only. The numbers in the light yellow shaded panels correspond to the area lying outside the contour for the two cases (cyan for LE and black for LE+ME) expressed as a percentage of the total parameter space plotted. These numbers quantify the improvement over the LE only option when the ME beam tune is used in conjunction with the LE beam tune in these panels. (taken from [382])

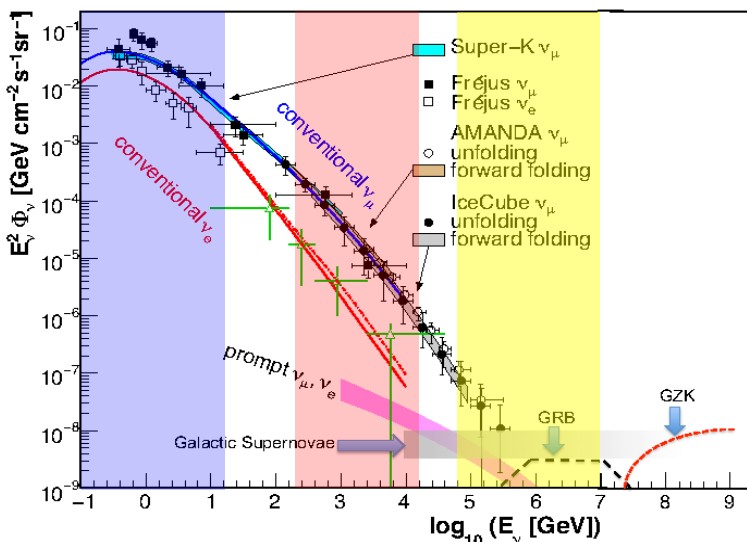

**Figure 231:** Atmospheric neutrino flux, with data from different experiments, the first two shaded regions(blue, red) show the approximate range of real neutrino energy for the low and high energy data samples used to perform oscillation analysis by IceCube. The energy range of the measured astrophysical neutrinos is shown in yellow.

## 23 NSI with IceCube (Salvado)

The last years the IceCube neutrino telescope detected for the first time astrophysical neutrinos. The large volumes required to measure the astrophysical flux make this detectors sensitive to atmospheric neutrinos, especially at high energies.

IceCube has been taking data for almost ten years in the final configuration and is able to measure neutrinos with energies from few GeV to more tens of PeV. This makes IceCube the perfect experiment to perform precision measurements of the high energy atmospheric neutrino flux.

Our current knowledge of neutrino oscillations can be well understood by a the coherent evolution of a three dimensional quantum system propagating macroscopic distances. This interference phenomena is very sensible to small parameters such the mass differences $7.39 \times 10^{-5} \text{eV}^2$ and $2.525 \times 10^{-3} \text{eV}^2$, tiny parameters compared with the kinetic energy of the measured neutrinos.

For neutrinos that travel in matter the so called matter potential due to the forward scattering should be added to the Hamiltonian. In this case the standard model of oscillations the Hamiltonian would be,

$$H = \frac{1}{2E} \begin{pmatrix} 0 & 0 & 0 \\ 0 & \Delta m_{21}{}^2 & 0 \\ 0 & 0 & \Delta m_{31}{}^2 \end{pmatrix} + U^{\dagger} \begin{pmatrix} V_{\text{CC}} + V_{\text{NC}} & 0 & 0 \\ 0 & V_{\text{NC}} & 0 \\ 0 & 0 & V_{\text{NC}} \end{pmatrix} U. \tag{23.1}$$

Standard matter effects for atmospheric neutrinos in the experimentally observed energy range are still marginal and difficult to measure. Nevertheless one may use atmo-

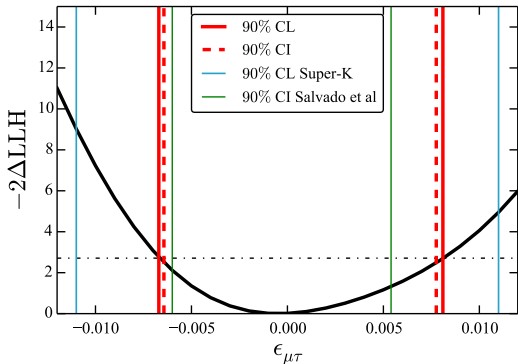

**Figure 232:** Bound on $\varepsilon_{\mu\tau}$ from different analyses are shown: in blue the analysis by SuperKamiokande, in red the low energy IceCube DeepCore analysis and in green the IceCube high energy NSI bound. Figure from [386]; see also Ref. [384].

spheric neutrino data to constrain non standard interactions of neutrinos with ordinary matter (NSI). In presence of NSI the Hamiltonian can be written as,

$$H = \frac{1}{2E} \begin{pmatrix} 0 & 0 & 0 \\ 0 & \Delta m_{21}{}^2 & 0 \\ 0 & 0 & \Delta m_{31}{}^2 \end{pmatrix} + \sqrt{2} G_F N_e(x) \begin{pmatrix} 1 + \varepsilon_{ee}(x) & \varepsilon_{e\mu}(x) & \varepsilon_{e\tau}(x) \\ \varepsilon_{e\mu}^*(x) & \varepsilon_{\mu\mu}(x) & \varepsilon_{\mu\tau}(x) \\ \varepsilon_{e\tau}^*(x) & \varepsilon_{\mu\tau}^*(x) & \varepsilon_{\tau\tau}(x) \end{pmatrix} \qquad (23.2)$$

where $\varepsilon_{ij}$ are the parameters that deviate from the SM case.

For atmospheric neutrinos the most sensitive sector is $\mu\tau$. The first result using Ice-Cube data with a preliminary study of the systematic errors was published by [384], giving compatible results with the latter published analysis with more accurate error treatment. The result for the current bounds of NSI performed in IceCube for the parameter $\varepsilon_{\mu\tau}$ are shown in Fig. 232. This include one year of data for high energy (green) [385], three years for the low energy using DeepCore(red) [386] and the analysis by Super-Kamiokande (blue) [387]. In general reaching high energies is good to prove new physics on the other hand the standard oscillations gets suppressed and the sensitivity to $\varepsilon' = \varepsilon_{\mu\mu} - \varepsilon_{\tau/tau}$ that comes due to the interference with the vacuum propagation gets lost. For this reason the high energy bound in Fig. 232 is using as a prior the measure of Super-Kamiokande.

In principle astrophysical neutrinos may be a good way to test NSI. The possibility of using the flavor ratio observable of the astrophysical neutrino flux was shown by [388]. This is still far from having sensitivity due to the low statistics and the need to measure the flavor content in different zenith directions.

Matter effects play an crucial roll to enhance the sensitivity of IceCube to $O(\text{eV})$ sterile neutrinos searches [389–392]. Therefore a modification of the matter potential such NSI may modify the result for the sterile searches relaxing the bound [214]. This apparent degeneracy can be broken since a large value for NSI is indeed moving the resonance in to the region not analyzed between the low energy and high energy in Fig. 231 [216]

In the limit where sterile neutrinos have large mass but they are still kinematically allowed to be produced, the oscillation effect is effectively averaged out. The interplay

between NSI and sterile neutrinos in this limit is well studied and shows that sterile neutrino is essentially equivalent to non-unitarity propagation or equivalently a modification of the matter potential [393]. The last one being for oscillations equivalent to a NSI. The effect of the sterile neutrinos can be parametrized by,

$$H = \frac{1}{2E} \begin{pmatrix} 0 & 0 & 0 \\ 0 & \Delta m_{21}{}^2 & 0 \\ 0 & 0 & \Delta m_{31}{}^2 \end{pmatrix} + + ((1-\alpha)U)^\dagger \begin{pmatrix} V_{\mathrm{CC}} + V_{\mathrm{NC}} & 0 & 0 \\ 0 & V_{\mathrm{NC}} & 0 \\ 0 & 0 & V_{\mathrm{NC}} \end{pmatrix} (1-\alpha)U, \quad (23.3)$$

where,

$$\alpha \simeq \begin{pmatrix} \frac{1}{2}\left(s_{14}^2 + s_{15}^2 + s_{16}^2\right) & 0 & 0 \\ \hat{s}_{14}\hat{s}_{24}^* + \hat{s}_{15}\hat{s}_{25}^* + \hat{s}_{16}\hat{s}_{26}^* & \frac{1}{2}\left(s_{24}^2 + s_{25}^2 + s_{26}^2\right) & 0 \\ \hat{s}_{14}\hat{s}_{34}^* + \hat{s}_{15}\hat{s}_{35}^* + \hat{s}_{16}\hat{s}_{36}^* & \hat{s}_{24}\hat{s}_{34}^* + \hat{s}_{25}\hat{s}_{35}^* + \hat{s}_{26}\hat{s}_{36}^* & \frac{1}{2}\left(s_{34}^2 + s_{35}^2 + s_{36}^2\right) \end{pmatrix}.$$

The matrix $\alpha$ is function of the mixing angles and complex phases between active and heavy states.

Examples of constraints for averaged out heavy neutrinos in this regime can be [392, 394, 395]. All of them are equivalent and can be re-parametrized as a NSI analysis. Is interesting to notice that this bounds both rely in the matter potential since this effects cancels in vacuum.

This equivalence between heavy sterile neutrinos and NSI may be broken if the sterile states are not kinematically allowed for some experiments or energy range. In the case of IceCube, with a large energy range to explore NSI, a tension between low and high energies may reveal the existence of a sterile state that becomes kinematically allowed in the un-analyzed energy region between the low and high energy data samples(Fig. 231). A full energy range analysis by IceCube to search for NSI or sterile neutrinos would be eventually needed to disentangle NSI from sterile neutrinos with different phenomenological implications in both low and high energies.

## 24 Inflation Meets Neutrinos (Denton)

The Standard Model (SM) of particle physics assumes neutrinos to be exactly massless. The discovery of neutrino oscillations has therefore been a clear hint to the existence of physics beyond the SM. Almost every attempt to account for neutrino masses necessitates the existence of yet unobserved particles or yet unobserved interactions in the neutrino sector.

It is interesting to consider the case in which $U(1)_{B-L}$ is spontaneously broken. In that case, we expect the existence of a new Goldstone particle – the Majoron – that couples to neutrinos via a Yukawa coupling,

$$\mathcal{L} = g_{\alpha\beta}\bar{\nu}_\alpha\nu_\beta\phi, \quad (24.1)$$

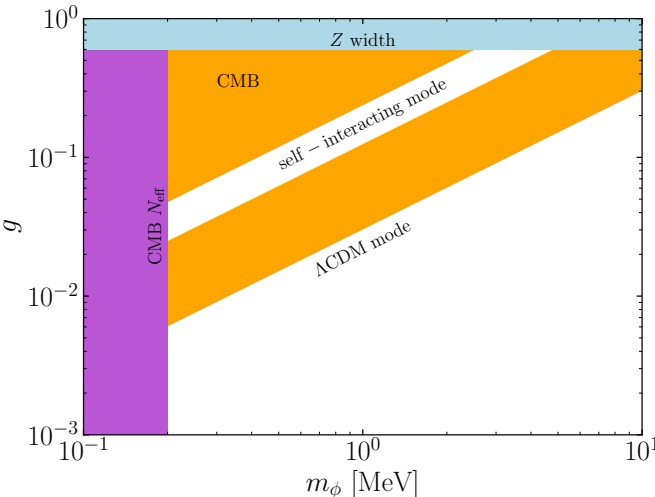

**Figure 241:** The various constraints on an interaction as described in Eq. (24.1). The new strongly interacting mode is identified by the diagonal white stripe through the orange CMB constrained region.

where $\alpha$ and $\beta$ stand for flavor or mass eigenstates, while $\phi$ is flavor blind. Therefore, we expect the appearance of non-standard neutrino interactions in those models.

In [396] we restricted our model to diagonal couplings, although there are reasons to consider couplings connected to the flavor sector. A comprehensive overview over the constraints as well as their ranges of validity in the $(g, m_\phi)$-plane can be found in [261]. Constraints on new physics of the form of eq. 24.1 are obtained from different observations: supernovae neutrinos [397, 398], big bang nucleosynthesis (BBN) [399] and the decay of the $Z$ boson [260, 400]. A relatively large parameter range of $(g, m_\phi)$ is however still allowed. Measurements of the cosmic microwave background (CMB) can tighten the bounds, but interestingly also point out a parameter range of $(g, m_\phi)$ that is in agreement with all observations given neutrinos that self-interact according to equation 24.1.

Such an interaction could also, in principle, be tested in the dense neutrino environment of a core-collapse supernovae. Ref. [401] showed that the neutrino flux from a CCSN is quite sensitive to such interactions.

By analyzing the effect of such a new interaction on CMB data from Planck [402], we found that while the data prefers no new interaction, the probability distribution is bimodal with a second peak at $G_{\text{eff}} = 10^{-1.7}$ MeV$^{-2}$ where $G_{\text{eff}} \equiv g^2/m_\phi^2$. While this is $\sim 10^9 G_F$ it is, in principle, allowed by all other constraints for $g \sim 0.1$ and $m_\phi \sim 1$ MeV, see fig. 241, under certain assumptions discussed below. Such a strong interaction modifies the allowed parameter spaces for various cosmological parameters, in particular enlarging the allowed space for $n_s$, the spectral index, and $r$, the tensor to scalar ratio. As these parameters are the key observables from models of inflation, it is necessary to consider this strongly interacting neutrino mode before ruling out models.

Two extremely well motivated models of inflation are Natural Inflation [403] which is related to a shift symmetry arising from an axion, and Coleman-Weinberg inflation [404,

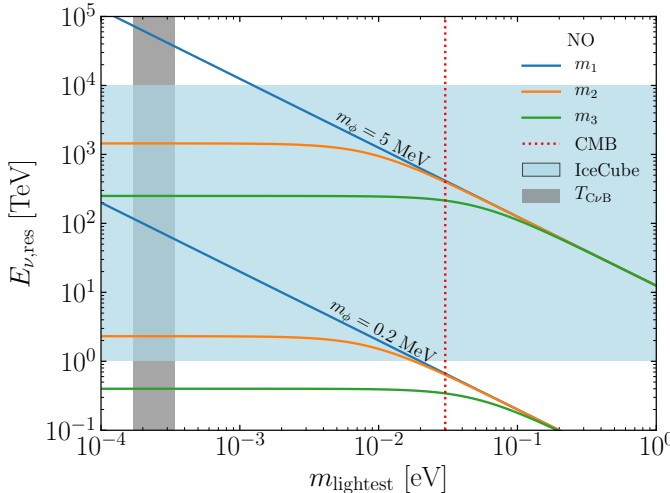

**Figure 242:** The resonant energy of each of the three CνB mass eigenstates as a function of the lightest neutrino mass in the normal mass ordering. The blue band is the broadest range of energies that IceCube could possibly measure the astrophysical neutrino flux. The red dashed line indicates the limit from cosmology [262].

405] which is a guaranteed 1-loop contribution. Natural inflation, in general, predicts non-zero and measurable tensor modes, while Coleman-Weinberg inflation predicts no tensor to scalar modes, but a somewhat low spectral index. Past constraints on $n_s$ and $r$ without this strongly interacting neutrino mode have put these models of inflation, along with others, in moderate tension with data [406]. Including this mode allows both of these models at $< 1\,\sigma$ again.

A separate similar analysis also included BAO data and the local $H_0$ inference [407]. These two additional data sets even further prefer the strongly interacting neutrino mode by somewhat alleviating the $H_0$ tension along with the $\sigma_8$ tension. They also very interestingly find $N_{\text{eff}} = 4.02 \pm 0.29$ which alleviates the sterile neutrino tension with CMB data, although the tension with BBN data still needs to be separately addressed.

The motivations for such a simple model are clear as they span a large number of seemingly unrelated open questions in particle physics and cosmology. It is then interesting to check if such a model can be experimentally tested. Mediators in the range $m_\phi \in [0.2, 5]$ MeV are exactly the region of interest that IceCube is sensitive to. IceCube has observed an unidentified high energy neutrino flux in the $10\,\text{TeV} \lesssim E_\nu < 1\,\text{PeV}$ energy range that is believed to be astrophysical [408] and extragalactic [409, 410]. High energy neutrinos propagating through the CνB are resonantly absorbed when $E_\nu \simeq m_\phi^2/2m_{\nu_i}$ where $m_{\nu_i}$ is the mass of the non-relativistic $\nu_i$. This results in a dip in the IceCube spectrum that would be expected to occur over energies $\sim 1$ TeV to $\sim 10$ PeV depending on the mass of the mediator, the absolute neutrino mass scale, and which neutrino mass state the resonance is occurring for. In light of the fact that there is currently a dip in the IceCube data at $\sim$500 TeV at low significance, this is a very interesting test to consider as IceCube progresses.

Confirming such a dip, however, is extremely challenging for several reasons. The

first is that the statistics at IceCube are poor and unlikely to improve enough without IceCube-Gen2, moreover, the energy resolution makes identifying features in the spectrum quite difficult. A deeper problem, however, is that distinguishing a dip feature from an astrophysical signal due to multiple sources would be extremely difficult. Finally, such a dip could also be due to resonant scattering off dark matter. It may still be possible, in principle, to identify such a model at IceCube, if multiple dips were identified at resonant energies corresponding to distinct neutrino masses. This, if measurable, would provide a nearly unambiguous signal of both the cosmic neutrino background, and a new mediator at the MeV scale.

While our assumptions of the previously existing constraints have tended towards the optimistic, a newer analysis took a more stringent view in interpreting strongly interacting neutrino models [411]. They find that such a strong interaction is only allowed if neutrinos are Majorana, the mediator is scalar, and the interaction is dominantly in the $\tau$ sector.

In conclusion, we find that there is a rich and complicated phenomenology involved with new neutrino interactions in the early universe. A strongly interacting mode, somewhat surprisingly, is allowed by the data. This mode is connected to many open questions in cosmology including inflation, the $H_0$ tension, the $\sigma_8$ tension, and the light sterile neutrino tension in the early universe making this an extremely attractive model to study. While numerous constraints exist from BBN and lab measurements of decays, some parameter space still exists. Finally, it may be possible to test nearly all of the parameter space in the future using IceCube.

## 25 Summary

To summarize, NSI is an active and interesting area of research, with connection to current and upcoming neutrino oscillation experiments, such as DUNE, as well as scattering experiments, including the LHC. There is also an interesting connection to dark matter and cosmology. We have reviewed in this document a representative collection of these topics. The field has been rapidly evolving and there are still many open questions (see Sec. 1.4). We hope the community addresses some of these issues.

### Acknowledgments

The workshop which inspired this document was supported in part by the US Neutrino Theory Network Program under Grant No. DE-AC02-07CH11359, as well as by the Department of Physics and the McDonnell Center for the Space Sciences at Washington University in St. Louis. KB, PBD, PSBD, SSC, MH, SJ and AT would like to thank the Fermilab Theory group for hospitality during a summer visit, where this report was completed. The work of PSBD is supported by the US Department of Energy under Grant No. DE-SC0017987. The work of KSB, SD and AT is supported in part by the US Department of Energy Grant No. de-sc0016013. CAA is supported by NSF grant PHY-1801996. JLB's work is supported by the U.S. Department of Energy, Office of Science, Office of Workforce Development for Teachers and Scientists, Office of Science Graduate Student Research

(SCGSR) program. The SCGSR program is administered by the Oak Ridge Institute for Science and Education for the DOE under contract number DESC0014664. JLB is also grateful to FNAL for their generous support of this and other work, and greatly appreciates the help of Saori Pastore (WUSTL), Steven Gardiner and Minerba Betancourt (FNAL) in these efforts. JLB's collaborative travel support was provided by the FNAL NTN proposal "Nuclear Theory for Accelerator Neutrino Experiments". SSC acknowledges the funding support from the European Union's Horizon 2020 research and innovation programme under the Marie Skłodowska-Curie grant agreement No 690575 while completing this write-up during his InvisiblesPlus secondment in Fermilab. The work of M-CC was supported, in part, by the U.S. National Science Foundation under Grant No. PHY-1620638. The work of AdG was supported in part by DOE grant #de-sc0010143. PBD's work is supported by the US Department of Energy under Grant Contract DE-SC0012704 and the Neutrino Physics Center. The work of BD is supported in part by the DOE Grant No. DE-SC0010813. Fermilab is operated by the Fermi Research Alliance, LLC under Contract No. DE-AC02-07CH11359 with the U.S. Department of Energy, Office of Science, Office of High Energy Physics. The work of DG and TH was supported in part by the Department of Energy under Grants No. DE-FG02-95ER40896 and DE-SC0015634, and in part by PITT PACC. DG is also supported by the U.S. National Science Foundation under the grant PHY-1519175. The work of SWL was supported by the Department of Energy under Contract No. DE-AC02-76SF00515. IM is supported by the U.S. Department of Energy (Grant No. DE-SC0013699). The work of JS is supported by MINECO grant FPA2016-76005-C2-1-P, Maria de Maetzu grant MDM-2014-0367, ELUSIVES (H2020-MSCA-ITN-2015-674896) and INVISIBLES-PLUS (H2020-MSCARISE-2015-690575). MT acknowledges support in part by the DOE grant de-sc0011784. MH acknowledges support from the Conselho Nacional de Ciência e Tecnologia (CNPq). The work of PM is supported by the Indian funding from University Grants Commission under the second phase of University with Potential of Excellence (UPE II) at JNU and Department of Science and Technology under DST-PURSE at JNU.

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
