# Peer review of "Neutrino Non-Standard Interactions: A Status Report"

_SciPost Physics Proceedings_

## Round 1 · Referee Report · Anonymous (Referee 1) · 2019-9-20

Strengths

This is a complete review of the interesting field of nonstandard interaction of neutrinos.

Weaknesses

No weakness as far as I could see.

Report

It is a very useful review and should be published.

---

## Round 1 · Referee Report · Anonymous (Referee 2) · 2019-11-3

Strengths

  1. State of the art report on the subject of non-standard interactions
  2. Covers many topics of current interest in the subject.
  3. Covers both phenomenological and theoretical aspects.

Weaknesses

  1. Since its a conference report contains mainly the works done by the authors and hence misses certain topics -- like source-detector nsi constraints, constraints from atmospheric neutrinos, possibility of breaking the degeneracy btween Dark-LMA and ordinary LMA by neutrino less double beta decay etc. So in that sense it is not a comprehensive review. But perhaps that was not the aim.

Report

It is a well written report covering fundamental aspects of NSI
and its implications. It will be very useful for the community as a reference manual.

Requested changes

  1. In page 2 the authors write 1 in 1+epslion_ee is due to Charged current interaction. Will the charged current interaction in presence of NSI not give any contribution to the potenetial ?

2.The degeneracy $\epsilon_{ee} \rightarrow - \epsilon_{ee} - 2$ was first mentioned in P.~Bakhti and Y.~Farzan, ``Shedding light on LMA-Dark solar neutrino solution by medium baseline reactor experiments: JUNO and RENO-50,'' JHEP {$\bf$ 1407}, 064 (2014). So perhaps this can be mentioned along with reference [15,16].

  1. For the sake of completeness the authors may think of including the references and few lines on the topics mentioned above, which were not covered. But this is an optional suggestion.

---

## Editorial Decision

resubmitted